# Enhanced dataset of global marine isoprene emissions from biogenic and photochemical processes for the period 2001–2020

**Lehui Cui**[1]**, Yunting Xiao**[1]**, Wei Hu**[1]**, Lei Song**[2]**, Yujue Wang**[3]**, Chao Zhang**[3]**, Pingqing Fu**[1]**, and Jialei Zhu**[1]

[1]Institute of Surface-Earth System Science, School of Earth System Science,
Tianjin University, Tianjin, 300072, China
[2]Center for Monsoon System Research, Institute of Atmospheric Physics,
Chinese Academy of Sciences, Beijing, 100029, China
[3]Frontiers Science Center for Deep Ocean Multispheres and Earth System, and Key Laboratory of Marine
Environment and Ecology, Ministry of Education, Ocean University of China, Qingdao 266100, China

**Correspondence:** Jialei Zhu (zhujialei@tju.edu.cn)

**Abstract.** Isoprene is a crucial non-methane biogenic volatile organic compound (BVOC) that exhibits the largest emissions globally. It is chemically reactive in the atmosphere and serves as the primary source of generating secondary organic aerosols (SOA) in terrestrial and remote marine regions. However, a comprehensive estimation of marine isoprene emissions is currently lacking. Here we built a module to present a 20-year (2001–2020) global hourly dataset for marine isoprene emissions, including phytoplankton-generated biological emissions (BIO emissions) and photochemistry-generated emissions in the sea surface microlayer (SML emissions) based on the latest advancements in biological, physical, and chemical processes, with high spatial resolutions. Our dataset suggests the annual global marine isoprene emissions amount to $1.097 \pm 0.009 \, \text{Tg yr}^{-1}$. Among these, the BIO emissions are $0.481 \pm 0.008 \, \text{Tg yr}^{-1}$ while SML emissions contribute $0.616 \pm 0.003 \, \text{Tg yr}^{-1}$. The ability of this module to estimate marine isoprene emissions was evaluated through comparison with a series of observations of marine isoprene concentrations and emission fluxes. The annual total isoprene emissions across the tropical ocean show a declining trend from 2001 to 2020. Most ocean regions exhibit a 1-year emission period, whereas a significant intraseasonal period is found in the tropical ocean. This dataset can be employed as input for the simulation of marine SOA formation in earth system models. This work provides the foundation for further studies into the impact of the air–sea system on marine SOA formation and its climate effect. The DOI link for the dataset is https://doi.org/10.11888/Atmos.tpdc.300521 (Cui and Zhu, 2023).

## 1 Introduction

Biogenic organic volatile compounds (BVOCs), one of the most important components in the marine boundary layer (MBL), play an important role in the formation of marine secondary organic aerosols (SOAs), particularly in pristine remote oceans (Yu and Li, 2021). By further generating SOAs, BVOCs have a potential impact on the radiation budget and cloud microphysical properties, thereby exerting a substantial influence on global climate change (Rosenfeld et al., 2014; Gantt et al., 2012). Among all the non-methane BVOC species, isoprene exhibits large emissions and demonstrates significant atmospheric chemical reactivity in the marine environment (Yokouchi et al., 1999; Guenther et al., 2012; Novak and Bertram, 2020). Isoprene has a lifetime of approximately 10–100 d in seawater (Booge et al., 2018). Once released into the atmosphere, it rapidly reacts with OH radicals, resulting in a short atmospheric lifetime of about 1 h

(Kameyama et al., 2014). Within the MBL, isoprene can undergo oxidation, leading to the formation of semi-volatile organic compounds (SVOCs) and low-volatility organic compounds (LVOCs) such as methacrolein and methacrylic acid. These compounds actively participate in the generation of marine SOAs (Claeys et al., 2004; Kim et al., 2017). Due to its significant emissions and capacity to contribute to SOA formation, marine isoprene plays a crucial role in aerosol generation and growth within the MBL. The estimation of marine isoprene emissions is essential and represents a fundamental aspect of future studies on marine SOAs and their climate effects (Carslaw et al., 2010). Isoprene can directly generate effects once it is emitted into the MBL, without requiring any transport processes.

Previous studies have estimated marine isoprene emissions using both bottom–up and top–down approaches. Bottom–up methods yielded emission estimates in the range of $0.11$–$1.36\,\mathrm{Tg\,yr^{-1}}$ (Gantt et al., 2009; Arnold et al., 2009; Booge et al., 2016; Conte et al., 2020; Myriokefalitakis et al., 2010; Palmer and Shaw, 2005; Sinha et al., 2007; Luo and Yu, 2010; Kim et al., 2017; Brüggemann et al., 2018; Shaw et al., 2010), while top–down methods yielded estimates in the range of $1.90$–$13.15\,\mathrm{Tg\,yr^{-1}}$ (Luo and Yu, 2010; Arnold et al., 2009). Over the past few decades, numerous studies have provided estimates of phytoplankton-generated biological emissions (BIO emissions) and photochemistry-generated emissions in the sea surface microlayer (SML emissions) over the global ocean. The estimation of BIO emissions is typically derived from an empirical linear relationship established between ocean chlorophyll concentration and isoprene emissions (Palmer and Shaw, 2005). This is because isoprene is a structural component and metabolic degradation product of various plant photosynthetic pigments such as chlorophyll and carotenoids (Hackenberg et al., 2017; Dani and Loreto, 2017; Booge et al., 2016). The empirical linear relationship can be further refined by taking into account different types of phytoplankton, which can vary in terms of their photosynthetic pigments and metabolic processes (Arnold et al., 2009; Gantt et al., 2009). Several enhancements and refinements have been incorporated into the calculation of BIO emissions. These updates include the diagnosis of the maximum depth of the euphotic zone each hour using the diffuse attenuation coefficient at 490 nm ($k_{490}$) and hourly downward surface solar radiation ($I_0$) (Gantt et al., 2009). The estimation of SML emissions is based on the surfactants present in the SML and their associated photochemical processes (Brüggemann et al., 2018; Conte et al., 2020). The SML acts as a flimsy interfacial layer between the marine atmosphere and the ocean. It is formed by natural surfactants produced through phytoplankton and other marine biological processes (Wurl et al., 2011). In previous studies, the quantification of surfactant enrichment in the SML was determined using net primary production (NPP), which serves as an indicator of phytoplankton productivity. Previous studies utilized experimentally based parameters to describe the photochemi-

cal processes within the SML, as well as a 10 m wind speed threshold indicating the point at which the SML starts to be torn apart (Ciuraru et al., 2015b; Brüggemann et al., 2017).

To date, estimates of global marine isoprene emissions have been derived by considering BIO and SML emission pathways (Conte et al., 2020; Zhang and Gu, 2022). However, few long-term datasets with high spatial resolutions are available for both types of emission as yet. Previous estimates also encountered challenges related to data availability and unclear emission mechanisms, leading to uncertainties in the estimated emissions (Palmer and Shaw, 2005, Gantt et al., 2009, Booge et al., 2016, Brüggemann et al., 2018, Conte et al., 2020). Estimations for high latitudes are particularly lacking due to limited satellite data coverage during the winter months. Moreover, previous estimations of vertical distributions of chlorophyll and isoprene concentrations did not entirely align with current observed vertical profiles in the subsurface ocean (Conte et al., 2020; Gantt et al., 2009; Zhang and Gu, 2022). The relationships between emissions and marine and meteorological factors, established on the basis of localized phytoplankton populations, are regionally constrained and may not be applicable in all situations. These limitations led to discrepancies between observed emissions and the estimations obtained using previous methods.

Here, we generated a $0.25° \times 0.25°$ grid dataset of global marine isoprene emissions covering a 20-year period from 2001 to 2020 with an updated method combining the latest emission features and state-of-the-art influencing factors. Two distinct types of emissions, BIO emissions and SML emissions, were calculated by satellite-derived monthly ocean chlorophyll concentration data from the MODIS and ERA5 hourly meteorological reanalysis separately. BIO emissions are derived by the correlations between isoprene production and marine chlorophyll concentration, while SML emissions are determined by the surfactant in the sea microlayer and wind speed. The availability and uncertainty of the dataset are discussed through comparisons with observed isoprene concentrations and a series of sensitivity tests. Our dataset can be used as input data for climate or atmospheric chemistry models. The module can also be coupled with the earth system model to calculate marine isoprene emissions online.

The next section elucidates the methods and factors employed in our estimation of marine isoprene emissions. Our results are compared with previous isoprene emission inventories and some field observations in Sect. 3. The characteristics of the marine isoprene emissions are analyzed in Sect. 4. Section 5 provides information on our dataset and data availability, and Sect. 6 presents the summary.

## 2 Methods

### 2.1 Input data

We obtained 20-year (2001–2020) monthly average chlorophyll concentration data at 9 km resolution and 490 nm downwelling radiative flux diffuse attenuation coefficient data with the same spatial resolution from the MODIS Level 3 product in the National Aeronautics and Space Administration (NASA) Ocean Color Web (NASA, 2022a, b) (https://oceancolor.gsfc.nasa.gov, last access: 20 November 2023). These two datasets were averaged into grids with a resolution of $0.25° \times 0.25°$ to fit the fifth-generation European Centre for Medium-Range Weather Forecasts (ECMWF) atmospheric reanalysis (ERA5) dataset used in this study (Hersbach et al., 2023). The ERA5 hourly average 10 m $u$-wind and $v$-wind component, 2 m temperature, sea surface temperature, and surface downwelling shortwave flux were applied in the module. Additionally, the monthly normalized water-leaving radiance at 410 nm for the period 2012–2020 from the National Oceanic and Atmospheric Administration (NOAA CoastWatch, 2017) (https://coastwatch.noaa.gov/cwn/index.html, last access: 20 November 2023) was utilized to determine the distribution of phytoplankton types together with chlorophyll concentration. These data are at 4 km resolution and are averaged into grids with a resolution of $0.25° \times 0.25°$. The most prevalent phytoplankton types on a monthly basis from 2012 to 2020 were determined for estimations of isoprene emissions over the 20-year period.

### 2.2 The BIO emission module

The phytoplankton-generated emission module was developed based on the assumption that the concentration of isoprene in the ocean remains static in each hour. This assumption implied that the net isoprene production is approximately equal to the isoprene flux from the ocean to the MBL in hourly calculation steps. Since isoprene is oxidized immediately once it enters the MBL because of its high chemical reactivity, the model assumes that the isoprene mixing ratio in the MBL is negligible. Typically, the lifetime of isoprene in the remote MBL is about 1 h, except coastal regions where there may be abundant terrestrial isoprene transport. Due to the small isoprene mixing ratio in the remote MBL ($\sim 20$ ppt, Yu and Li, 2021), it is reasonable to neglect the air-to-sea flux and focus on marine isoprene emissions into the MBL. The BIO model can be expressed by the following equation:

$$F_\mathrm{b} = (1 - \alpha) \cdot P \cdot S, \tag{1}$$

[TS1] where $F_\mathrm{b}$ (g grid$^{-1}$ h$^{-1}$) represents the isoprene emission flux from the air–sea interface to the MBL, and $P$ (g m$^{-2}$ h$^{-1}$) is the isoprene production rate generated by phytoplankton. $S$ (m$^2$ grid$^{-1}$) is the grid cell area and $\alpha$ is the chlorophyll-based rate constant to determine the biological and chemical consumption of isoprene per hour. Biological consumption is marine isoprene loss due to the degradation by isoprene-degrading bacteria and other microbials. Chemical consumption is caused by the photochemical processes in the surface ocean, which is calculated from the reaction rate constant. The value of $\alpha$ is calculated by the following equation based on a previous observational study (Simo et al., 2022):

$$\alpha = (0.0042 \times C_\mathrm{chl} + 0.0021)$$
$$(\text{When } C_\mathrm{chl} < 5.77 \,\mathrm{mg\,m^{-3}})$$
$$\alpha = (0.0042 \times 5.77 + 0.0021) = 0.026$$
$$(\text{When } C_\mathrm{chl} \geq 5.77 \,\mathrm{mg\,m^{-3}}). \tag{2}$$

The term $0.0042 \times C_\mathrm{chl}$ represents the degradation and utilization of isoprene by heterotrophic bacteria (Simo et al., 2022). It accounts for the observed correlation between bacterial activity and chlorophyll concentrations in the mixed layer. The second term, 0.0021, is the empirical rate of chemical consumption of isoprene per hour in the ocean (Palmer and Shaw, 2005; Booge et al., 2018). It is important to note that when the chlorophyll concentration in the seawater exceeds $5.77 \,\mathrm{mg\,m^{-3}}$, $\alpha$ is set to a constant value of 0.026 as a maximum stable biological and chemical consumption per hour. This approach was derived from observations when the chlorophyll concentration in the seawater was up to $5.77 \,\mathrm{mg\,m^{-3}}$ (Simo et al., 2022). Therefore, the specific value of 0.026 is determined to account for biological and chemical consumption in nutrient-rich environments.

The isoprene production rate, $P$, was determined by a linear relationship between chlorophyll concentration, radiation, and the diffuse attenuation coefficient at 490 nm, as well as the classification of phytoplankton types. The radiation was used to determine the term $I$, which was calculated as the total radiance in the euphotic layer. $T_c$ represents the ability of isoprene production for different phytoplankton types. Four distinct types of phytoplankton (i.e., haptophytes, *Prochlorococcus*, *Synechococcus*-like cyanobacteria, and diatoms) were involved, each with a different isoprene production rate defined below. These coefficients were determined in previous studies, which will be discussed in the next section. $C_\mathrm{chl}$ (mg m$^{-3}$) represents the sea surface chlorophyll concentration, which was considered as a parameter within the mixed layer of each grid cell. A comprehensive explanation of the methodology used to identify the phytoplankton types is provided in Sect. 2.3.

Here, Eq. (3) is for the isoprene production rate:

$$P = I \cdot C_\mathrm{chl} \cdot T_c. \tag{3}$$

$I$ (m) is the integrated result of radiation in the planktonic euphotic zone, where

$$I = 2\ln\left(\frac{2I_0}{3600}\right) H_\mathrm{max} - k_{490} \cdot H_\mathrm{max}^2. \tag{4}$$

$I$ is limited by the maximum depth $H_{max}$ (m) (Gantt et al., 2009; Shaw et al., 2003), which is calculated by

$$H_{max} = \left( -\ln\left( \frac{2.5}{I_0} \right) \cdot k_{490}^{-1} \right). \tag{5}$$

In Eqs. (4) and (5), $k_{490}$ ($m^{-1}$) is the diffuse attenuation coefficient of downwelling radiative flux at 490 nm, which characterizes the downwelling irradiance within the water column. Finally, $I_0$ ($J\,m^{-2}$) is downward surface solar radiation, for which we used hourly data here.

The aforementioned equations were utilized to estimate the hourly marine isoprene emissions originating from phytoplankton within each grid, with a spatial resolution of $0.25° \times 0.25°$. The diurnal variation of isoprene BIO emissions was estimated based on the hourly radiation data in this module. It should be noted that isoprene BIO emissions are negligible during nighttime hours due to the absence of radiation, as supported by relevant observational studies (Gantt et al., 2009; Sinha et al., 2007; Hackenberg et al., 2017).

## 2.3   Phytoplankton type distribution

Along with various oceanological conditions of different oceans on the global scale, various dominant phytoplankton types would produce isoprene in different rates through their photosynthesis and metabolic process (Booge et al., 2018; Dani and Loreto, 2017). For instance, cyanobacteria predominantly control the isoprene emission in tropical and subtropical oceans, while diatoms exhibit higher rates at high latitudes (Dani and Loreto, 2017). Moreover, it has been observed that the larger the size of a distinct type of phytoplankton, the less likely it is to thrive in the oligotrophic region of the ocean, due to the limited specific surface area of phytoplankton cells (Alvain et al., 2008). The coefficient $T_c$ ($\mu mol\ isoprene\,(g\,chl\ a)^{-1}\,h^{-1}$) in Eq. (3), which relates chlorophyll concentration to isoprene emissions, is determined by phytoplankton type. Four types of phytoplankton and their corresponding coefficients $T_c$ in this module are 0.028 for haptophytes, 0.029 for *Prochlorococcus*, 0.032 for *Synechococcus*-like cyanobacteria, and 0.042 for diatoms (Gantt et al., 2009).

The dominant phytoplankton type was determined using monthly satellite-observed normalized water-leaving radiance at 410 nm and seawater chlorophyll concentration. This classification method is based on the distinctive effects of pigments on the normalized water-leaving radiance for each phytoplankton type (Alvain et al., 2005, 2008), and the details are summarized in Table 1. A simplified scheme of normalized water-leaving radiance at 410 nm is used to determine phytoplankton types for the chlorophyll range 0.04–3 mg m$^{-3}$ (Alvain et al., 2005, 2008). The haptophyte is a widespread marine producer, which dominates chlorophyll-*a*-normalized phytoplankton standing stock in modern oceans (Liu et al., 2009). Haptophytes are dominant in the global ocean all year round, with contributions varying

from 45 % to 70 % depending on the season (Alvain et al., 2005). Because of its small cell volume with relatively large surface extent, this species is especially dominant in oligotrophic waters. Therefore, we decided to use the coefficient of 0.028 for haptophytes in the oligotrophic waters where chlorophyll-*a* concentration is lower than 0.04 mg m$^{-3}$ and area with missing values as suggested by Alvain et al. (2005). Conversely, the chlorophyll concentration is greater than 3 mg m$^{-3}$ in many coastal areas with sufficient nutrients. The normalized water-leaving radiance data are always missing due to turbid water bodies inshore in the coastal areas, which leads to underestimated isoprene BIO emissions there. Based on previous observational studies in the East China Sea, which is a typical coastal region, it was determined that the dominant phytoplankton type is a combination of 50 % diatoms and 50 % haptophytes in the grids with chlorophyll concentrations greater than 3 mg m$^{-3}$ (Guo et al., 2014; Li et al., 2018; Liu et al., 2016).

Figure 1 illustrates the monthly global distribution of marine phytoplankton types. Note that the large range of other types in the polar regions is caused by the limitations of satellite-derived data. In these polar regions, there are frequent missing values in satellite observations due to the low radiation levels during the winter months, which may lead to uncertainty regarding the phytoplankton types and BIO emissions in high-latitude regions. However, the impact of missing data in polar and subpolar regions is relatively limited, because previous studies indicated that isoprene is mostly emitted in the tropical and subtropical oceans in a trade-off relationship with dimethyl sulfide (DMS) (Dani and Loreto, 2017), which is also shown in our dataset. Therefore, despite the challenges posed by missing data in polar and subpolar regions, the overall estimation of global isoprene emissions is minimally affected when using other phytoplankton types in these areas. It is also found that the other types appear in the subtropical ocean, which is generally due to the low nutrient level there, resulting in chlorophyll concentrations lower than 0.04 mg m$^{-3}$. For the oligotrophic ocean, our module cannot determine the specific phytoplankton type, but the emissions in these areas were still included in our estimation with an emission factor of 0.028 according to the dominance of the haptophyte types in the global ocean (Alvain et al., 2005). Another noticeable ocean area of the Arabian Sea and Bay of Bengal, also with other types of phytoplankton, is affected by the weather conditions in the summer months. We applied the interpolation method for each grid cell in these regions for the boreal summer emissions. Details of the interpolation method and the improvement are discussed in Sect. 2.5.

## 2.4   The SML emission module

The radiation intensity within a specific radiation band (280–400 nm) has been found to be the factor determining the photochemistry-driven production and emission of isoprene according to the linear relationship between isoprene produc-

**Table 1.** Scheme of phytoplankton types and classification method.

| Chlorophyll concentration (mg m$^{-3}$) | Normalized water-leaving radiation | | Types | Factors ($T_c$) |
|---|---|---|---|---|
| < 0.04 | | | Other type | 0.028 |
| 0.04–3 | < 0.4 | | Other type | 0.028 |
| | 0.4–2.4 | 0.4–0.8 | Haptophytes | 0.028 |
| | | 0.8–1.0 | *Synechococcus*-like cyanobacteria | 0.032 |
| | | 1.0–1.3 | *Prochlorococcus* | 0.029 |
| | | 1.3–2.4 | Diatoms | 0.042 |
| | > 2.4 | | Other type | 0.028 |
| > 3 | | | 50 % haptophytes +50 % CE1 diatoms | 0.035 |

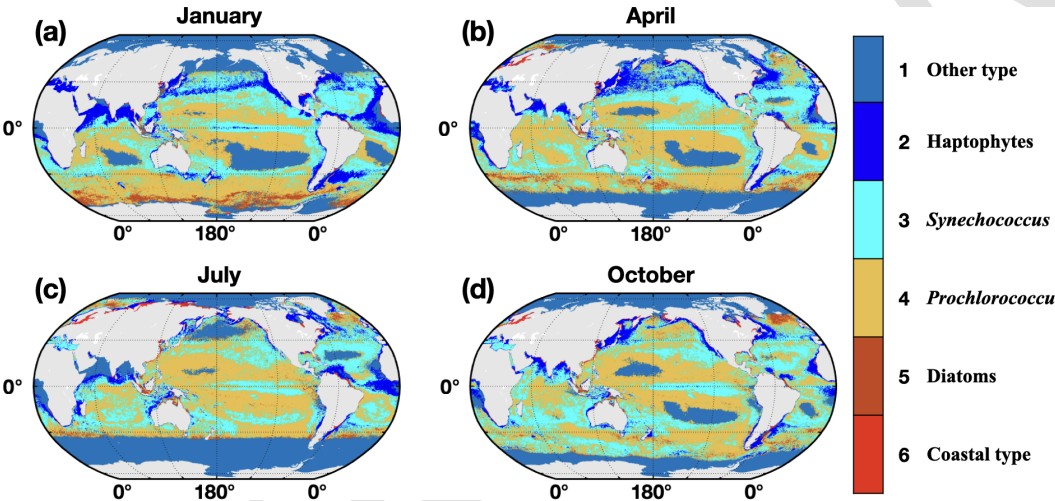

**Figure 1.** The spatial distribution of dominant phytoplankton types in January **(a)**, April **(b)**, July **(c)**, and October **(d)** during the period 2012–2020. Six phytoplankton types are used here: 1 for other type, 2 for haptophytes, 3 for *Synechococcus*-like cyanobacteria, 4 for *Prochlorococcus*, 5 for diatoms, and 6 for coastal type, which uses 50 % haptophytes +50 % diatoms.

tion and radiation intensity (Brüggemann et al., 2018). Following the parameterization of Brüggemann et al. (2018) and Conte et al. (2020), Eq. (6) is used to estimate the marine photochemical emission of isoprene:

$$F_s = F_{lab} \times \mu_{photo} \times S, \tag{6}$$

where $F_s$ (g grid$^{-1}$ s$^{-1}$) is the flux of isoprene emissions from the SML per second. $F_{lab}$ (molecules mW$^{-1}$ s$^{-1}$) is the flux of isoprene from marine SML and biofilm measured in previous laboratory studies (Ciuraru et al., 2015b, a; Brüggemann et al., 2017). $F_{lab} = 4.95 \times 10^7$ is used in this work, which represents the mean value within the range $(3.71 \times 10^7 - 6.19 \times 10^7)$ used by Conte, based on the data from Brüggmann and Ciuraru (Ciuraru et al., 2015a; Brüggemann et al., 2017; Conte et al., 2020). $S$ (m$^2$) is the grid cell area and $\mu_{photo}$ (mW m$^{-2}$) is the photochemical emission po-

tential. The calculation of $\mu_{photo}$ is determined by Eq. (7):

$$\mu_{photo} = E_{280-400} \times F_{surf} \times k_{SML}, \tag{7}$$

where $E_{280-400}$ (mW m$^{-2}$) is radiation intensity, which accounts for radiation between 280 and 400 nm reaching the surface of the ocean. It is determined to be 3.535 % of the downward surface solar radiation (Conte et al., 2020).

$F_{surf}$ represents the different surfactant concentrations in the SML defined as a ratio given by

$$F_{surf} = \frac{\ln(c_{surf})}{\ln(c_{max})}. \tag{8}$$

In Eq. (8), $F_{surf}$ accounts for a logarithmic decay of isoprene SML emissions with the decreasing surfactant concentration (Brüggemann et al., 2018). The two surfactant concentration terms, $c_{surf}$ and $c_{max}$, are determined with a simplified method based on previous research, using the con-

centration equivalents of Triton X as the surfactant concentration in SML (Wurl et al., 2011). Here the nutrient level of the ocean is determined by the concentration of chlorophyll $C_{chl}$ (mg m$^{-3}$). The surfactant concentration reaches its maximum at $c_{max} = 663\,\mu g\,Teq\,L^{-1}$, which is the mean concentration in the eutrophic waters ($C_{chl} \geq 0.4\,mg\,m^{-3}$) in the experiment by Wurl et al. (2011). A linear relationship was established to determine the surfactant concentration in the oligotrophic ocean with $C_{chl} < 0.4\,mg\,m^{-3}$, which is $c_{surf} = 857\,C_{chl} + 320\,\mu g\,Teq\,L^{-1}$. The $c_{surf}$ approaches $320\,\mu g\,Teq\,L^{-1}$ while chlorophyll concentration is at a low level.

The exchange velocity factor $k_{SML}$ in Eq. (7) is calculated with the following equation (Mcgillis et al., 2004):

$$k_{SML} = \frac{8.2 + [0.014 \times w^3]}{8.2 + [0.014 \times w_{lab}^3]}, \tag{9}$$

where the parameter $k_{SML}$ used in this study is normalized based on the work of Brüggemann et al. (2018) and Ciuraru et al. (2015a, b) with $w_{lab} = 5.31 \times 10^{-2}\,m\,s^{-1}$, which is derived from laboratory studies (Brüggemann et al., 2018; Ciuraru et al., 2015b, a); $w$ represents 10 m wind speed. In addition, the SML emissions are assumed to occur only when the 10 m wind speed is less than $13\,m\,s^{-1}$ according to field observations (Brüggemann et al., 2017, 2018; Sabbaghzadeh et al., 2017). The average annual SML emissions were calculated to be $0.616\,Tg\,yr^{-1}$ for the period 2001–2020, which is about 30 % larger than the BIO emissions.

## 2.5 Interpolation for missing values

Due to the influence of dust aerosols and clouds, there are regions with missing data for marine chlorophyll concentration, such as the north Arabian Sea and Gulf of Guinea (30° N–30° S, 0–120° E) and the North Pacific Subpolar Gyre (60–30° N, 150° E–150° W) (Alvain et al., 2005, 2008). Consequently, the calculation of isoprene emissions using the aforementioned methods is not possible in these regions, leading to the underestimation of global isoprene emissions. The missing-value regions primarily exist in the tropical and subtropical areas, where the seasonal variation of isoprene emissions is limited. An interpolation for hourly isoprene BIO and SML emissions is applied during the boreal summer (June, July, and August) and winter months (December, November, and January) in this study. This interpolation for the missing-value area is based on the emissions in the adjacent spring and fall months in the same grid. In the North Pacific region, missing values only occur in the summer months, with an extent comparable to interpolated regions in the tropical and subtropical areas. The same interpolation method is applied to fill the missing data and provide a basic emission status.

Figure 2 illustrates the interpolation process, which is an integral part of dataset establishment. This process entails utilizing the hourly isoprene emission data to calculate the monthly average diurnal variation for each grid that contains missing values. The cubic spline interpolation is then applied to determine the missing values in the summer and winter months using adjacent spring and fall emission data. The interpolated area accounts for approximately 3.1 % of the global ocean during the summer and 0.9 % during the winter. Overall, the interpolation increases global isoprene emissions by 7.0 % in the summer, 3.4 % in the winter, and 2.4 % for the entire year. For the comparison, the standard deviation of the 20-year annual marine isoprene total emissions is 0.0095 Tg, which is about 0.8 % of the annual total emissions. Compared to the result of the sensitivity tests, the change caused by the interpolation method is smaller than most of the other factors in their range of values.

## 3 Evaluation and comparison

### 3.1 Comparison with observations

The accuracy of our method for estimating isoprene emission flux was assessed by comparing our isoprene emission dataset with previous cruise and inshore observations. Most of these results provide information on the range of isoprene concentrations in the surface seawater of various regions, including the Atlantic, Northern Pacific, East China Sea, tropical Indian Ocean, and Southern Ocean, while several results were derived from one sampling site with only a single value such as for the tropical Pacific, Peninsular Malaysia, and the Mediterranean (Table 2).

Furthermore, our work collected observed marine isoprene emission flux results from previous research including four cruise studies and two inshore sites (Table 3). Most of these flux results were derived from calculations that involved the isoprene concentration in the seawater and the mixing ratio of isoprene in the marine boundary layer (method described below). Additionally, there was a floating flux chamber study conducted in the Peninsular Malaysia coastal region to measure the isoprene flux directly (Uning et al., 2021) (Table 3).

The comparison of estimated isoprene emission flux and isoprene concentration in the seawater with the corresponding observations was performed in the respective regions and months. The comparison of emission fluxes is summarized in Fig. 3. The absolute value deviations between our estimated results and the observations range from 42.3 % to 45.5 % in coastal regions and from 3.64 % to 54.6 % in remote oceans. Among the six comparisons, the largest deviation (54.6 %) was found in the North Atlantic region, as observed by Hackenberg et al. (2017) in boreal fall. However, our simulated emission flux showed a close agreement with another observation in the North Atlantic by Kim et al. (2017) with absolute value deviations of 35.4 %. It is important to note that various factors, such as occasional bloom events and the inherent variability of observations, may contribute to the differences observed in the same area. The mean deviation of

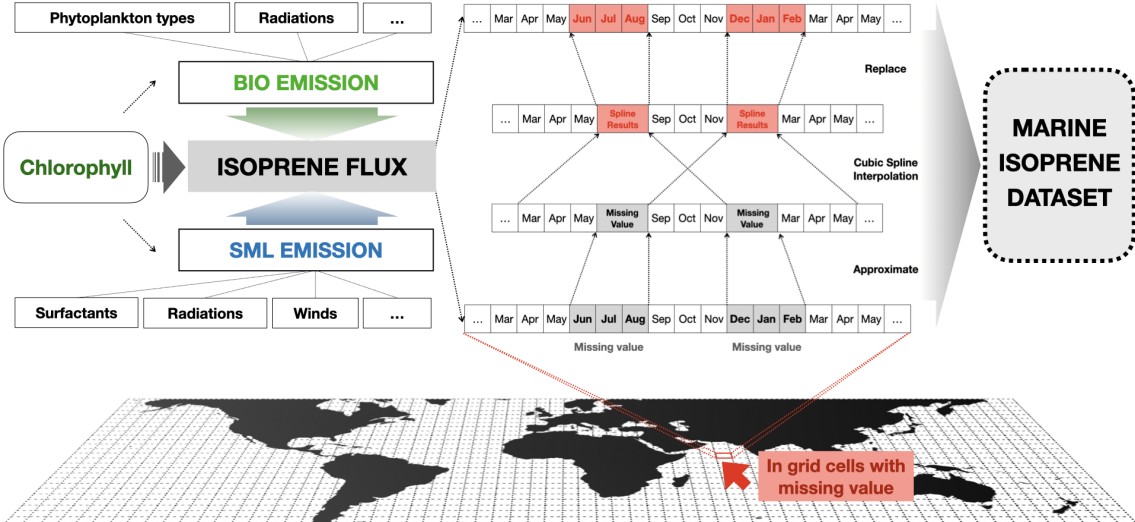

**Figure 2.** Calculation process of the estimation and interpolation method used in our dataset. Based on chlorophyll concentration and other meteorological factors, two types of isoprene emissions were included to determine the total marine isoprene emission flux. Cubic spline interpolation was used for grid cells with missing emission values during the period of boreal summer and winter.

33.7 %, which was derived from the isoprene emission flux comparison, is smaller than most of our sensitivity results.

In our method, the isoprene emission flux is directly derived assuming equivalence with isoprene production, and therefore the isoprene concentrations in seawater do not necessarily have to be explicitly calculated in the module described in Sect. 2. In order to make a comparison with the observed isoprene concentrations in the seawater, we calculated the seawater isoprene concentrations by simulated isoprene emission flux and exchange velocity using the following equation:

$$C_{\mathrm{iso}} = \frac{F_{\mathrm{b}} + F_{\mathrm{s}}}{k_{\mathrm{ex}}}, \tag{10}$$

where $F_{\mathrm{b}}$ is the BIO emissions flux, $F_{\mathrm{s}}$ is the SML emissions flux, and $k_{\mathrm{ex}}$ is the exchange velocity on the air–sea interface. Note that here both BIO and SML emissions are considered to have effects on the marine isoprene concentration with the assumption that all isoprene from SML emissions enters the underlying seawater. This may cause an overestimation of the isoprene concentration in the seawater compared to the actual situation. The exchange velocity $k_{\mathrm{ex}}$ (cm h$^{-1}$) is determined by wind and sea surface temperature (Wanninkhof, 2014):

$$k_{\mathrm{ex}} = \frac{0.31 w^2}{\sqrt{\frac{Sc}{660}}}, \tag{11}$$

where $w$ is 10 m wind speed. Note that Eq. (11) is valid with $w$ in the range of 4–15 m s$^{-1}$. $Sc$ is the Schmitt number determined by the sea surface temperature (Palmer and Shaw, 2005)

$$Sc = 3913 - 162.13t + 2.67t^2 - 0.012t^3, \tag{12}$$

where $t$ is the sea surface temperature in degrees Celsius. The hourly 10 m wind speed, sea surface temperature from reanalysis data, and hourly isoprene emission flux from our dataset were used to calculate sea water isoprene concentrations using Eq. (11). The comparisons between simulated isoprene concentrations and observations were conducted in six regions with different latitudes and various nutrient conditions (Fig. 4). The derived isoprene concentrations from our emission flux data have ranges overlapping the observations in the Southern Ocean, Atlantic and East China Sea, while the simulated isoprene concentrations in the North Pacific and tropical Indian Ocean were overestimated by 32.0 %–48.3 % compared to observations. The exchange velocity calculated using Eq. (11) may introduce uncertainty, which could partly explain the bias between simulations and observations. The uncertainty of the method for the sea-to-air exchange process will further affect the results of marine isoprene concentrations, which is about 20 % according to Wanninkhof (2014). In addition, the constant factor of 0.31 in Eq. (11) and the Schmitt number $Sc$ determined by Eq. (12) can vary depending on ocean conditions such as solute types and sea surface temperature, which may also contribute to the bias between simulations and observations.

## 3.2 Comparison with previous estimation results

The average annual isoprene emissions for the period 2001–2020 are estimated to be 1.097 Tg yr$^{-1}$, with a range of 1.075–1.112 Tg yr$^{-1}$ using our module. The annual global BIO emissions range is 0.464–0.493 Tg yr$^{-1}$, which corresponds to the total emissions from various types of phytoplankton. The annual global SML emissions are in the range of 0.611–0.621 Tg yr$^{-1}$, which is generated by photochem-

**Table 2.** Observed marine isoprene concentrations in previous studies.

| Time | Location | Range (pmol L$^{-1}$) | References |
|---|---|---|---|
| 1990 Apr | South Pacific | 6.69–99.1 | Bonsang et al. (1992) |
| 2010–2011 Dec–Jan | Southern Ocean | 0.2–348 | Kameyama et al. (2014) |
| 2012 Sep–Oct | Polar Northwest Pacific | 1.3–31 | Ooki et al. (2015) |
| | Subpolar Northwest Pacific | 2.2–60 | |
| | Transition water | 6.4–165 | |
| | Subtropical Indian Ocean | 5.4–50 | |
| | Tropical Indian Ocean | 29–75 | |
| 2008 Nov | East Atlantic | 2–157 | Booge et al. (2016) |
| 2013 Jul | East China Sea, south Yellow Sea | 32.46–173.52 | Li et al. (2017) |
| 2013 Oct–Nov | North Atlantic | 21 | Kim et al. (2017) |
| 2012 Oct–Nov | North Atlantic | 8.75–63.26 | Hackenberg et al. (2017) |
| 2013 Oct–Nov | North Atlantic | 1.12–38.20 | |
| 2013 Mar | Arctic | 1.96–10.57 | |
| 2013 Jul–Aug | Arctic | 3.86–66.38 | |
| 2014 Jul–Aug | Indian Ocean | 6.1–27.1 | Booge et al. (2018) |
| 2014 Aug–Oct | West Pacific | 15.9–33.1 | Li et al. (2019) |
| 2018 Jul | Zenibako coastal | 27.08–28 | Li et al. (2020) |
| | Bering Sea | 21.36–67.73 | |
| 2017 Jul–Sep | Peninsular Malaysia | 8.3–34.3 | Uning et al. (2021) |
| 2018 Apr–May | Southwest UK coast | 80–100 | Phillips et al. (2021) |
| 2017 Jul | Davis Strait | 59 | Wohl et al. (2022) |
| 2019 Jul–Aug | Southern Ocean | < 54.00 | Zhou et al. (2022) |
| 2018 Apr | Tropical Pacific | 17.5 | Simo et al. (2022) |
| 2014 Apr–May | Mediterranean | 25.1–39.0 | |
| 2014 Oct–Nov | Atlantic | 4.5–104.1 | |
| 2015 Jan–Feb | Southern Ocean | 6.3-64.2 | |

ical processes in the SML. The standard deviation of the 20-year annual marine isoprene total emission is 0.0095 Tg, which is about 0.8 % of the annual total emissions.

In previous studies, several model-based estimations of marine isoprene emissions were conducted, as summarized in Table 4. Most of these studies utilized a bottom–up approach, while a few employed a top–down approach. There is a significant difference in the estimated isoprene emissions between these two methods. Top–down estimations generally yield larger values compared to bottom–up estimations. This difference can be attributed, in part, to the exclusion of high-emission events and hotspots in bottom–up methods (Yu and Li, 2021). The missing values of the source data and the unclear mechanisms of marine isoprene production, consumption, and sea–air exchange all lead to uncertainty using the bottom–up method (Conte et al., 2020; Gantt et al., 2009; Hackenberg et al., 2017; Palmer and Shaw, 2005; Yu and Li, 2021). On the other hand, the limited observation datasets and insufficient spatial resolutions of input data decrease the accuracy of current top–down results (Arnold et al., 2009; Luo and Yu, 2010). Additionally, the air–sea exchange flux of marine isoprene, which is used in top–down methods, cannot be directly observed, further contributing to the uncertainty in these approaches. Furthermore, most of the available isoprene flux observations are conducted at inshore sites, which may not be suitable for estimating emissions in remote ocean areas (Simo et al., 2022). Based on the previous estimate method, our work has applied several improvements to our bottom–up method in order to address the existing gaps and discrepancies between top–down and bottom–up results. These improvements are discussed in detail in the next section.

**Table 3.** Observed marine isoprene emission flux in previous studies.

| Time | Location | Range molecules cm$^{-2}$ s$^{-1}$ | Range (Daily) µg m$^{-2}$ | Methods | References |
|---|---|---|---|---|---|
| 2013 Oct–Nov | North Atlantic | $5.0 \times 10^7$ | 4.84 | Eddy covariance method | Kim et al. (2017) |
| 2017 Jul–Sep | Peninsular Malaysia | $19.4 \times 10^7$ | 18.71 | Floating flux chamber TD-GC-MS | Uning et al. (2021) |
| 2017 Apr–May | Arabian Sea | $1.5–12 \times 10^7$ | 1.45–11.61 | Seawater isoprene concentration | Tripathi et al. (2020) |
| 2012&2013 Oct–Nov | Atlantic Ocean | $0.005–34 \times 10^7$ | 0.006–32.58 | Exchange velocity | Hackenberg et al. (2017) |
| Time | Location | Range nmol m$^{-2}$ d$^{-1}$ | Range (Daily) µg m$^{-2}$ | Methods | References |
| 2001 May | Western North Pacific | 161.5 (22.17–537.2) | 10.98 (1.57–37.67) | Average isoprene mixing ratio | Li et al. (2017) |
| 2010–2011 Dec–Jan | Southern Ocean | 181–313 | 12.26–21.29 | Seawater isoprene concentration Exchange velocity | Kameyama et al. (2014) |

**Table 4.** Marine isoprene emission estimations in previous studies.

| Compounds | Emissions Tg yr$^{-1}$ | | Reference |
|---|---|---|---|
| Isoprene | 0.11 | (BIO emissions) | Palmer and Shaw (2005) |
| | 1.36 | (BIO emissions) | Sinha et al. (2007) |
| | 0.79 | (BIO emissions) | Gantt et al. (2009) |
| | 0.31 | (BIO emissions) | Arnold et al. (2009) |
| | 1.90 | (Top–down) | Arnold et al. (2009) |
| | 0.99 | (BIO emissions) | Myriokefalitakis et al. (2010) |
| | 0.36 | (BIO emissions) | Luo and Yu (2010) |
| | 13.15 | (Top–down) | Luo and Yu (2010) |
| | 0.24 | (BIO emissions) | Booge et al. (2016) |
| | 0.65 | (BIO emissions) | Kim et al. (2017) |
| | 1.11 | (SML emissions) | Brüggemann et al. (2018) |
| | 0.75 | (Total emissions) | Conte et al. (2020) |
| | 0.96 | (BIO emissions) | Li et al. (2020) |
| | 1.10 | (Total emissions) | This study |

## 3.3 Model improvements and comparisons

In our model, we implemented several ways to improve the estimation of global BIO and SML emissions compared to previous datasets. These improvements include updates to the methods and an increase in temporal and spatial resolution. The temporal resolution of the dataset was enhanced to 1 h, allowing for a more detailed examination of the diurnal and seasonal variations of isoprene emissions to capture short-term changes and events that may influence emissions, which probably provides a better representation of emission dynamics. The hourly wind speed data performed better in the calculation of SML emissions. The SML emissions directly corresponded to the cube of wind speed (Eqs. 6, 7, 9),

so that the high wind speed made large contributions. High wind speed can be captured hourly, while monthly averaging eliminates high wind speed, which results in a relative underestimation of SML emissions using monthly wind speed data as input. The spatial resolution was set to $0.25° \times 0.25°$, which is consistent with the spatial resolution of ERA5 reanalysis data. This fine spatial resolution allows for a more precise representation of the spatial distribution of isoprene emissions, particularly in coastal regions where emission patterns vary significantly. The phytoplankton type distribution scheme used in the calculation of BIO emissions has been updated and simplified based on the normalized water-leaving radiation at 410 nm and chlorophyll concentration data, according to previous work (Alvain et al., 2005, 2008). This up-

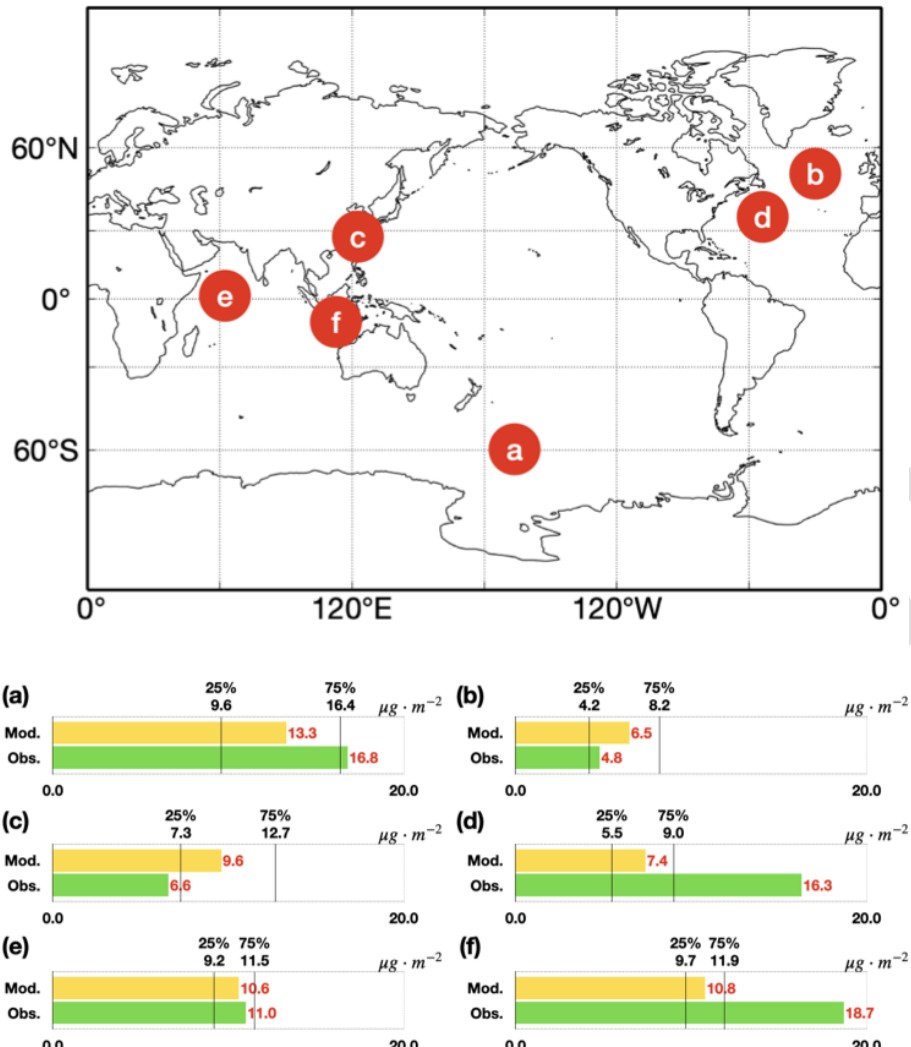

**Figure 3.** Comparisons between simulated isoprene emission daily fluxes (unit in µg m$^{-2}$) and observations. Yellow bar is daily mean isoprene emission flux in corresponding ocean regions. Two solid lines represent quartiles of the range for simulations. Green bar is the daily mean of observed emission flux. Six regions including the Southern Ocean **(a)** (Kameyama et al., 2014), North Atlantic **(b)** (Kim et al., 2017) and **(d)** (Hackenberg et al., 2017), East China Sea and south Yellow Sea **(c)** (Li et al., 2017), Arabian Sea **(e)** (Tripathi et al., 2020) and Peninsular Malaysia **(f)** (Uning et al., 2021).

date helps to avoid the issue of missing phytoplankton types within a number of grid cells in coastal regions, leading to a substantial improvement in the accuracy of emission estimation in these specific areas. Moreover, the latest parameterization (in Eq. (2)) was developed to estimate the biological and chemical consumption based on observations by Simo et al. (2022) with an upper limit of 0.373 when the chlorophyll concentration was larger than 5.77 mg m$^{-3}$. These improvements help to reduce the uncertainty of BIO emission estimation and enable us to examine the characteristics of BIO emissions in high spatial resolutions.

The estimation of SML emissions was based on the radiation, wind speed, and surfactants in the SML. Here we used chlorophyll concentration to determine the quantity of surfactants based on field measurement by Wurl et al. (2011), instead of the net primary production used by Brüggemann et al. (2018). This simplification of the model eliminates potential inconsistencies that may arise from using different datasets (chlorophyll concentration and net primary production) to describe the nutrient levels of the ocean.

## 3.4 Data uncertainties

The uncertainties in our model primarily are present in the parameterizations of various physical and biological and chemical processes. Since the linear relationship between isoprene emission and phytoplankton biomass is not universally applicable in all situations (Kameyama et al., 2014),

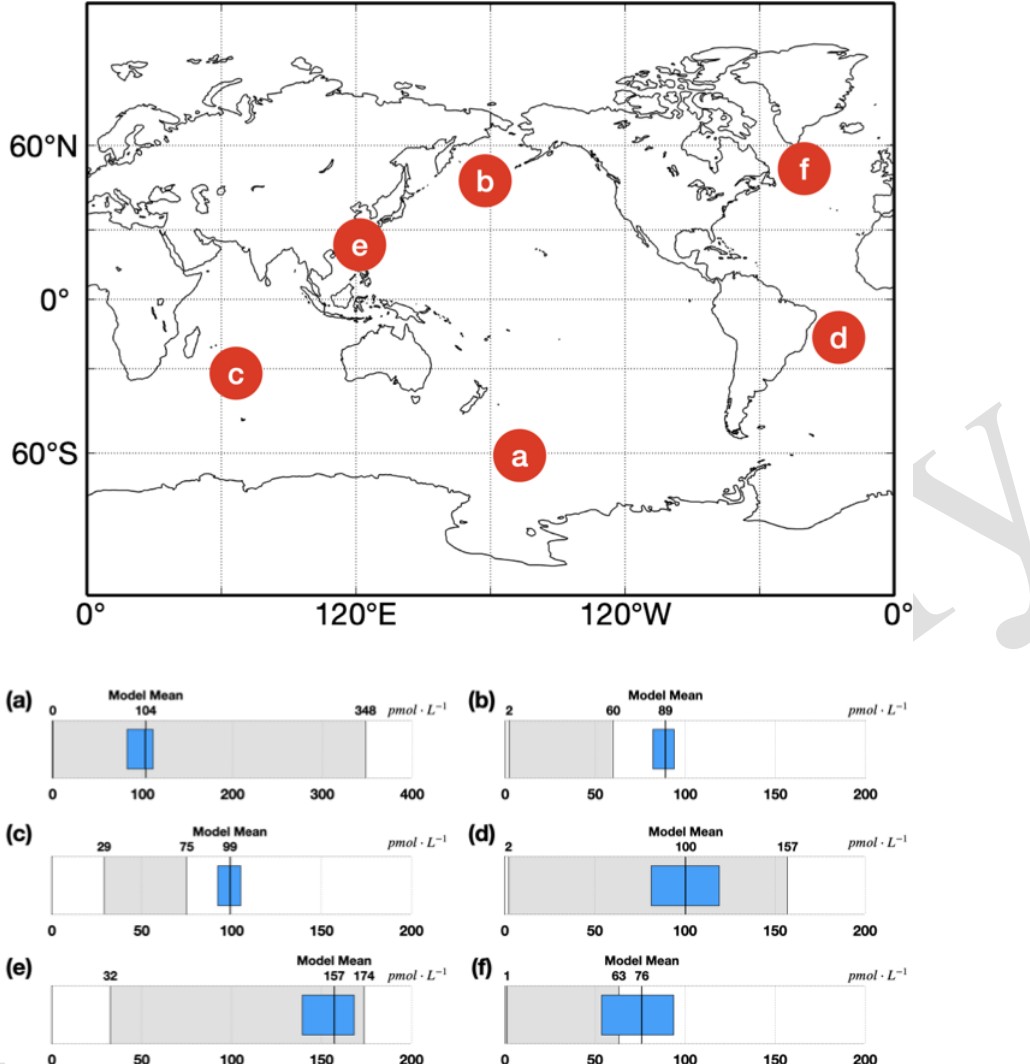

**Figure 4.** Comparisons between simulated isoprene concentrations (unit in $\mathrm{pmol\,L^{-1}}$) and observations. Blue bar is the range (25–75 percentile) of simulated isoprene concentrations in the corresponding ocean region. The solid black line within the blue bar represents the mean of simulated isoprene concentrations. The gray bar is the range of observed isoprene concentrations. Six regions including the Southern Ocean **(a)** (Kameyama et al., 2014), subpolar Pacific Ocean **(b)** (Ooki et al., 2015), tropical Indian Ocean **(c)** (Ooki et al., 2015), east Atlantic Ocean **(d)** (Booge et al., 2016), East China Sea and south Yellow Sea **(e)** (Li et al., 2017), and North Atlantic **(f)** (Hackenberg et al., 2017).

a large-size measurement is required at higher spatial and temporal resolution to improve the parameterizations. Additionally, the column concentration of chlorophyll was derived from satellite observations in our module with the assumption that chlorophyll is well mixed in the euphotic layer, although satellite observations are only able to detect the chlorophyll concentration on the surface of the ocean. The isoprene productions in our model are determined by integrating other depth CE2, taking into account the radiation levels that control the isoprene emission rate at different depths. However, previous studies indicated that the highest isoprene concentrations may occur below the surface, often coinciding with the maximum chlorophyll concentrations (Conte et al., 2020; Wohl et al., 2022). As a result, uncertainty in the

vertical distributions of chlorophyll and isoprene concentration under the SML may lead to the uncertainty in the estimation of marine isoprene emission. Furthermore, previous observations detected notable VOC emissions in the Arctic region and high-latitude Southern Ocean during winter (Abbatt et al., 2019; Wohl et al., 2023). These emissions may be underestimated in our model due to the limitations of satellite data. Moreover, observations have indicated that isoprene production in the ocean occurs even when phytoplankton are covered by sea ice. As a result, high marine isoprene concentrations were measured in ice-edge waters and melted ponds (Wohl et al., 2022, 2023; Abbatt et al., 2019). The accumulated isoprene under sea ice is emitted once the ice melts, but this process was not included in our module.

Here we design several tests to determine and describe the uncertainties of this dataset. It is essential to testify and quantify the possible affect factors by means of sensitivity testing and to provide descriptions for the further use of the dataset and corresponding calculation modules. There are several possible sources of the dataset uncertainties, including the input reanalysis dataset, satellite data, and empirical parameterization.

From a series of sensitivity tests, the range of annual global BIO emissions is 0.443–0.664 and 0.583-0.655 Tg yr$^{-1}$ for SML emissions. The uncertainty of BIO emissions is mainly caused by the phytoplankton types with their specific corresponding correlation. These types are determined from our simplified method, with the maximum parameter used in our module for diatoms and the minimum parameter used for haptophytes. We determined the BIO emission uncertainty range using diatoms or haptophytes as the only input type. The uncertainty of SML emissions is also related to marine productivity, as the parameter of surfactant concentration is determined by chlorophyll-$a$ concentrations in our module. We split the surfactant concentration into three bins, according to the chlorophyll-$a$ concentration. In our test for the uncertainty of SML emissions, the maximum and the minimum concentration are used to determine the uncertainty range.

Our module used the dominant phytoplankton type for each month instead of higher temporal resolution due to the restriction of the temporal resolution of chlorophyll-$a$ and water-leaving radiance data. We simply diagnosed the monthly phytoplankton types during the period 2012–2020. The phytoplankton types in 55 % of the global grid cells were the same for July in the 9-year period, while the types in 89 % of the grid cells were the same for over 5 months of July in the 9-year period. The other months also have similar results. The phytoplankton types in 51 % of the grid cells were the same for January, and 90 % were the same for over 5 months of January in the 9-year period. For the mean percentage of the 12 months, 51 % of the grid cells were of the same phytoplankton type, and 89 % were the same for over 5 of the 9 years. As a result, we believe it is reliable to apply the monthly dominant phytoplankton type in each grid during 2012–2020 for the estimation for all 20 years (2001–2020).

A monthly marine isoprene emission dataset is made using the same module but with monthly input reanalysis, which is also from the ERA5 product. These relatively low temporal resolution emission data were compared with our hourly dataset. For the global annual total emissions, the monthly data result in 1.050 Tg yr$^{-1}$, which was underestimated by 4 % compared to the estimation using hourly radiation. Among this, the annual SML emissions are 0.499 Tg yr$^{-1}$, which was underestimated by 19 % compared to the hourly result of 0.616 Tg yr$^{-1}$. The annual BIO emissions were 0.551 Tg yr$^{-1}$, overestimated by 15 % compared to the hourly result of 0.481 Tg yr$^{-1}$. The deviation of BIO emissions was mainly accounted for by the accordance of the radiation data and their temporal resolution, which caused

a fixed-depth euphotic layer for every month. Besides, the monthly averaged radiation ignored the influence of weather conditions on radiation. The deviation of SML emissions was mainly from the monthly mean wind speed data. High wind speed is eliminated by the monthly average, while the SML emissions correspond directly with the wind speed cubed. The hourly wind speed data perform better in the calculation of SML emissions. The SML emissions correspond directly with the cube of wind speed (Eqs. 6, 7, 9), so that the high wind speed made large contributions. High wind speed can be captured hourly, while monthly averaging eliminates high wind speed, which results in a relative underestimation of SML emissions using monthly wind speed data as input.

Another input meteorological dataset is used in our module to validate the robustness of our module. We used the data from the National Center for Environmental Prediction (NCEP, 2015) (https://rda.ucar.edu/datasets/ds083.3, last access: 20 November 2023) Global Data Assimilation System (GDAS)/FNL (final) 0.25 Degree Global Tropospheric Analyses and Forecast Grids. We derived the radiation on the ground and water surface level and wind speed at 10 m for a monthly average of 2020 as input data for monthly calculations. This result (referred to as "TEST result") is compared with the monthly emission data calculated from monthly ERA5 reanalysis, which was discussed in the previous paragraph. The TEST result shows the global total isoprene emissions are 1.132 Tg for 2020, with BIO emissions of 0.588 Tg and SML emissions of 0.544 Tg. The total emissions of the TEST result are 7.8 % higher than the former monthly results from the ERA5 reanalysis, which is 1.050 Tg yr$^{-1}$. The BIO emissions and SML emissions in the TEST result are both higher than the former monthly estimations by 6.7 % and 9.0 %, respectively. This deviation between these two reanalysis products is obviously smaller than the deviation between our dataset and the observation data, as well as the deviations in the results of the sensitivity tests. Therefore, we think our module is valid enough and applicable to data from multiple sources.

A series of sensitivity tests were conducted for input meteorological data, input parameters, and assumptions used in our module. These sensitivity tests focused on several critical input factors and parameters which may have effects on the uncertainty of the dataset. Detailed information and the results of the sensitivity test are presented in Tables 5 and 6.

The sensitivity tests are based on the monthly result of our module. For the input data, we chose radiation, 10 m wind speed, and chlorophyll-$a$ concentration and set a 50 % deviation of each factor. The results show that radiation is the most important factor for the total emissions, which caused up to 35.0 % deviation. The chlorophyll-$a$ concentration also had a considerable influence on the total emissions and contributed about 27 % deviation. The test results suggest there is a different influence for BIO emissions and SML emissions. Radiation is dominant in SML emissions with about 50 % deviation, while its influence on BIO emissions is only

**Table 5.** Sensitivity test of input reanalysis data.

| Emission | ERA5 reanalysis (Tg yr$^{-1}$) | NCAR reanalysis | Wind | | Radiation | | Chlorophyll-$a$ concentration | |
|---|---|---|---|---|---|---|---|---|
| | | | +50% | −50% | +50% | −50% | +50% | −50% |
| BIO | 0.551 | +6.7% | – | – | +13.6% | −21.4% | +49.9% | −49.9% |
| SML | 0.499 | +9.0% | +38.9% | −21.2% | +49.5% | −50.1% | +1.6% | −2.2% |
| Total | 1.050 | +7.8% | +18.5% | −10.1% | +31.0% | −35.0% | +26.9% | −27.2% |

**Table 6.** Sensitivity test of assumptions and parameters.

| Emission | ERA5 reanalysis (Tg yr$^{-1}$) | Phytoplankton types | | Surfactant | | C_air | | F_lab | |
|---|---|---|---|---|---|---|---|---|---|
| | | All diatoms: 0.042 | All other: 0.028 | Min: 320 | Max: 663 | 1 ppt (Global) | 1 ppt (Remote) + 20 ppt (Coastal) | Max: $6.19 \times 10^7$ | Min: $3.71 \times 10^7$ |
| BIO | 0.551 | +38.1% | −8.0% | – | – | −11.1% | −12.9% | – | – |
| SML | 0.499 | – | – | −5.4% | +6.4% | – | – | +25.1% | −25.1% |
| Total | 1.050 | +20.0% | −4.2% | −2.1% | +3.0% | −5.8% | −6.8% | +11.9% | −11.9% |

up to 21.4 %. On the contrary, the chlorophyll-$a$ concentration contributes half of the deviation for BIO emissions, but only about 2 % for SML emissions. This result suggests the chlorophyll-$a$ concentration in centered on the large value and small value. Note that the wind speed only affects SML emissions, while the greater wind speed contributes approximately twice the deviation that the smaller wind does. This reflects the non-linear relationship between wind speed and SML emissions.

Besides, we design several tests for the assumption and parameters used in our module, including the phytoplankton types, surfactant concentration in the sea microlayer, fixed euphoric zone depth, and the assumption for the zero isoprene mixing ratio in the marine boundary layer (MBL). First, we set the phytoplankton type into an "all diatom" scenario and "all other" scenario. The global total emissions increase by 20.0 % and BIO emissions increase by 38.1 % in the "all diatom" scenario. On the other hand, the total emissions decrease by only 4.2 %, while BIO emissions decrease by 8.0 % using the "all other" scenario. The "all other" test results in a more stable change than using diatoms as the dominant phytoplankton type. This result is similar to the former conclusion that the haptophytes, which have the same emission parameter as the other type, are dominant in a large extent of the global ocean. The surfactant concentration test shows an even smaller influence on the total emissions (−2.1 %–3.0 %) and SML emissions (−5.4 %–6.4 %). It suggests that SML emissions are dominated by meteorological factors rather than marine productivity. Finally, we investigate the influence of isoprene in the MBL with various mixing ratios. The BIO module is based on the assumption that isoprene in the MBL has a very short lifetime, as well as on its low mixing ratio in most remote ocean areas. The presence of isoprene in the MBL will inhibit the emission of marine isoprene to the MBL. Considering the atmospheric concentration of isoprene in the MBL ($C_{air}$), an emission suppression term is added to Eq. (1):

$$F_b = (1 - \alpha) \cdot P \cdot S - k_{ex} \cdot H \cdot C_{air}. \tag{13}$$

In Eq. (13), the air–sea exchange velocity $k_{ex}$ (m h$^{-1}$) is determined by Eq. (11). $H$ is a dimensionless Henry's law constant, which is calculated by Mochalski et al. (2011):

$$H = \exp\left(-17.85 + \frac{4130}{T + 273.16}\right). \tag{14}$$

Here, $T$ is water temperature in degrees Celsius. An observation-based coastal isoprene mixing ratio of 400 ppt is used and applied to the global ocean (Warneke et al., 2004). This produces a 51.0 % decrease in the total emissions and nearly all BIO emissions are suppressed. Isoprene mixing ratios under the remote ocean condition are collected from Yu's previous work (Yu et al., 2021). Here we used the mixing ratio of 20 ppt for the coastal region and 1 ppt as input data and calculated the total global emissions. For the mixing ratio of 20 ppt in the coastal region, the total global emissions decrease by 6.8 %, while BIO emissions decrease by 12.9 %. For the mixing ratio of 1 ppt, the total global emissions decrease by 5.8 %, while BIO emissions decrease by 11.1 %. The isoprene mixing ratio in the MBL shows a strong effect on global isoprene emissions. However, previous studies suggest that the high mixing ratio in the coastal area is seriously affected by the terrestrial source, especially under the specific condition that the lifetime of isoprene is equal to or even greater than the terrestrial-source isoprene transportation temporal scale (Warneke et al., 2004, Booge et al.,

2016). Furthermore, several observations suggest a minimum isoprene mixing ratio is below the detection limit range, usually smaller than 2 ppt. We believe that in the most remote oceans with adequate oxidation radicals, isoprene is consumed very fast with a lifetime of hours (Palmer et al., 2005, Booge et al., 2016, Conte et al., 2020). The very short lifetime of isoprene in the MBL still confirms our former assumption of a zero mixing ratio of isoprene in the MBL. Besides, even though the possible isoprene mixing ratio exists in the MBL, which is measured to be several parts per trillion, it only affects a small amount of the total isoprene emissions.

## 4 Results

### 4.1 Spatial and temporal distribution of marine isoprene emissions

Generally, our dataset suggests annual global marine isoprene emissions ranging from 1.075 to 1.112 $\text{Tg yr}^{-1}$ for the period 2001–2020, with an average of 1.097 $\text{Tg yr}^{-1}$ over the 20 years. Annual average global BIO emissions for the 20-year period were 0.481 $\text{Tg yr}^{-1}$, ranging from 0.464 to 0.493 $\text{Tg yr}^{-1}$, while annual average global SML emissions were 0.616 $\text{Tg yr}^{-1}$, ranging from 0.611 to 0.621 $\text{Tg yr}^{-1}$. In the 20-year period, the average annual emissions in the Northern Hemisphere (NH) amounted to approximately 44.9 %, whereas the Southern Hemisphere (SH) accounted for 55.1 % of the total emissions. However, the emissions per unit area in NH (3.3 $\text{mg m}^{-2}\text{yr}^{-1}$) is 6.5 % are greater than those in the SH (3.1 $\text{mg m}^{-2}\text{yr}^{-1}$) due to the larger and better nutritional status of coastal ocean areas in the NH. The difference in the total emissions between the two hemispheres is largest in boreal winter (Fig. 5). The emissions in the boreal winter of the SH contributed 17.7 % of annual global emissions on average, while the emissions in the same season of the NH accounted for only 8.7 %. Meanwhile, the emissions per unit area in the NH (0.70 $\text{mg m}^{-2}$) were still lower than those in the SH (0.85 $\text{mg m}^{-2}$) in boreal winter. Radiation and duration of day directly dominate the seasonal variations of total emissions and they also affect chlorophyll concentration, thereby indirectly influencing the emissions. This highlights the non-negligible importance and dominance of marine isoprene emissions in the SH compared to the NH, suggesting potential environmental impacts and climate modifications associated with these emissions.

Based on the datasets, we can find distinct spatial characteristics in marine isoprene emissions at a global scale, revealing specific patterns in annual emissions (Fig. 6). The BIO emissions are closely linked to chlorophyll concentration, exhibiting a similar spatial pattern to marine chlorophyll (Fig. 6a, c). Regions such as coastal areas, convergence zones, and upwelling areas (e.g., East China Sea, tropical Pacific, offshore Peru) exhibit high BIO emissions due to the presence of elevated chlorophyll concentrations and abundant nutrients. These conditions may arise from anthropogenic eutrophication in coastal areas or the natural flow of ocean current systems (Dai et al., 2023). The emission rates in coastal areas are significantly larger than the remote ocean areas by several orders of magnitude. In the 20-year period, the mean isoprene BIO emissions per unit area in the coastal ocean areas (East Asia, 110–130° E, 40–20° N) are 0.273 $\mu\text{g m}^{-2}\text{h}^{-1}$, while the average emissions are 0.076 $\mu\text{g m}^{-2}\text{h}^{-1}$ in remote ocean areas (subtropical Pacific, 180–120° W, 20–30° S). The global average BIO emissions per unit area are 0.141 $\mu\text{g m}^{-2}\text{h}^{-1}$. However, the emissions from remote oceans still dominate global marine isoprene emissions due to the vast surface area of remote ocean regions. Additionally, there is evidence of an increased frequency of potential phytoplankton bloom events, particularly in coastal regions and the Southern Ocean, over the past two decades (Dai et al., 2023). The spatial distribution of SML emissions is more uniform than that of BIO emissions and is limited in range. Indirect use of chlorophyll data contributed to this characterization, in which the surfactant concentrations were determined from chlorophyll and divided into three bins. Therefore, SML emissions are insensitive to chlorophyll concentration, which results in a different spatial pattern of SML emissions and chlorophyll. SML emissions contribute relatively larger isoprene emissions in the subtropical remote ocean regions. In these regions, SML emissions are dominated by radiation and wind speed. This relationship is further discussed in Sect. 4.2.

The annual average global marine isoprene BIO and SML emissions exhibit a slight decreasing trend over the last 20 years (Fig. 7b). However, the emission trends vary significantly among different ocean regions. The annual emissions from the Pacific (49.5 %) and Indian Ocean (22.2 %) contribute 71.7 % to global isoprene emissions, and emissions in both regions were decreasing in the last 20 years (Fig. 7a). By contrast, the Arctic Ocean shows an increasing trend in annual emissions, although its contribution to global marine isoprene emissions is only 1.2 % (Fig. 7a, c). This increasing trend in the Arctic Ocean was further analyzed using sea ice concentration and chlorophyll concentration data. We find that shrinkage of the sea ice extent and reduction of the sea ice concentration in recent decades led to an increase in both emission area and period in boreal summer. Additionally, recent research suggests that along with the ice-free area lasting longer, new phytoplankton blooms in fall are more likely to happen (Ardyna et al., 2014). The bloom events may potentially contribute to the increasing isoprene emissions. The emissions in the low-latitude ocean are most important over the global marine isoprene emissions attributed to intense radiation, and high concentrations of chlorophyll relative to the subtropical remote ocean areas, which account for 36.7 % of global marine isoprene emissions. This trend is controlled by the tropical air–sea system potentially. Our former investigation suggests that the El Niño–Southern Oscillation (ENSO) influences the tropical Pacific isoprene emissions significantly when the ENSO is at its strong positive or

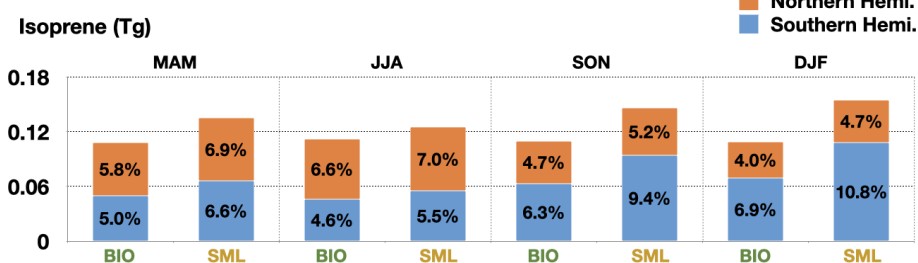

**Figure 5.** Seasonal variation of the contribution of BIO and SML emissions from the two hemispheres to annual global emissions for the period 2001–2020.

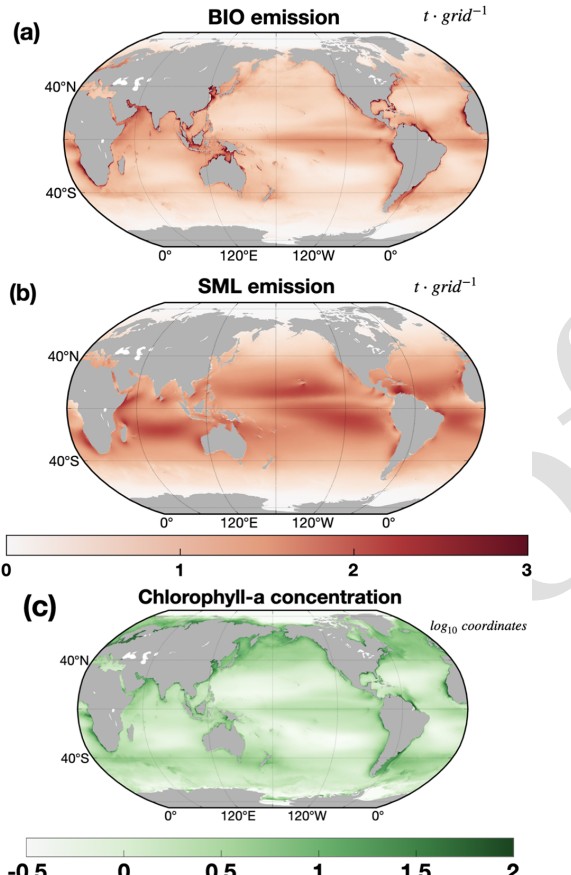

**Figure 6.** Mean annual BIO emissions **(a)**, annual SML emissions **(b)** (unit in t), and annual chlorophyll-*a* concentration **(c)** (in log10 coordinates) for 20-year period.

negative phase. In the strong positive phase, the tropical west wind is strengthened, which leads to warm water accumulating in the tropical Pacific. This process leads to an increase in the sea surface temperature in the tropical Pacific, which further weakens the isoprene emissions in this area. The SML emissions in low latitudes decreased by 5.6 % yr$^{-1}$ while the BIO emissions decreased by 3.0 % yr$^{-1}$ over the 20-year period (Fig. 7a, b). In addition, the SML emissions in the Atlantic also had a decreasing trend, while the BIO emissions had no specific trend in the 20-year period.

## 4.2 Influence of marine and meteorological factors

The variations in isoprene emissions are primarily influenced by marine and meteorological conditions, both directly and indirectly. The effects of four dominant factors including 10 m wind speed, downward surface solar radiation, sea surface temperature and marine chlorophyll concentration, were examined by correlating them with BIO and SML emissions (Fig. 8). Chlorophyll concentration is considered a factor that quantifies nutrient levels and phytoplankton activities, which was also used to determine the surfactant content in the SML. Globally, chlorophyll concentration has a significant positive correlation ($r = 0.67$, $p \leq 0.05$) with BIO emissions (Fig. 8g). However, chlorophyll concentration has a positive correlation with SML emissions in the polar, subpolar, and tropical regions, but a negative correlation in the subtropical region, suggesting that other critical factors control SML emissions in these areas (Fig. 8h). Downward surface solar radiation is another important factor influencing both BIO emissions and SML emissions. There is a globally positive correlation between downward surface solar radiation and SML emissions (Fig. 8d), with a significant coefficient of $r = 0.62$ ($p \leq 0.05$) in the global average, while positive correlations with BIO emissions are only found at mid and high latitudes beyond 40° N and 40° S (Fig. 8c). Sea surface temperature and 10 m wind speed have less impact on BIO and SML emissions compared to the other two factors in most open ocean areas. A large number of grid cells with weak correlations ($|r| \leq 0.4$) and correlations that did not pass our significance test ($p > 0.05$) for sea surface temperature and 10 m winds peed with BIO and SML emissions are shown in Fig. 8 (Fig. 8a, b, e, f). These two physical factors affect marine isoprene emissions indirectly by altering the air–sea exchange processes, and show contrasting correlations, especially in the tropical ocean (Fig. 8a, b, e, f), where BVOCs emissions are determined by local atmosphere and ocean conditions (Xu et al., 2016). The wind mainly contributed to the SML emissions. First, it determined the surfactant coverage on the ocean surface. A wind threshold of

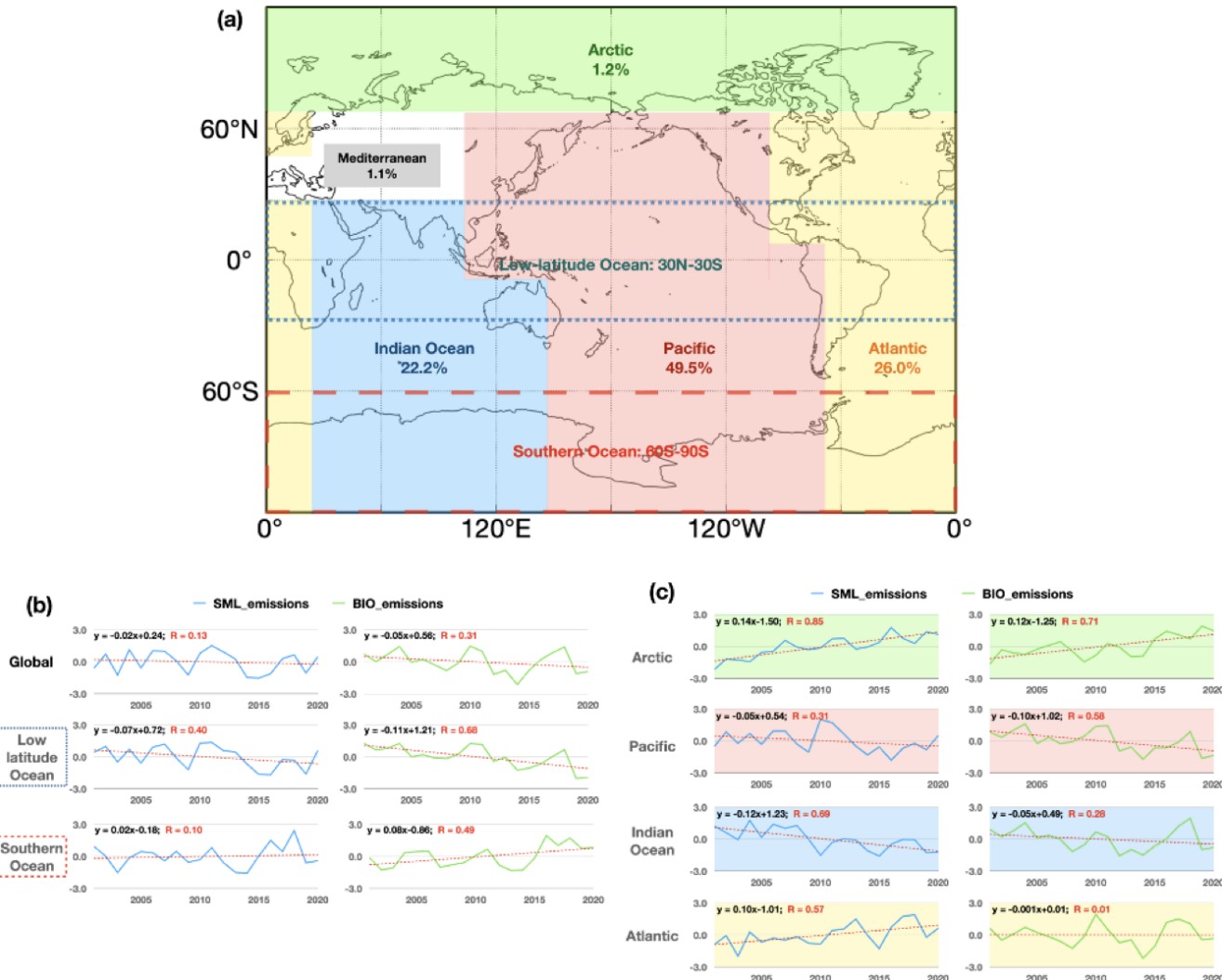

**Figure 7.** The 20-year mean contributions of annual isoprene emissions from different ocean regions to global annual emissions **(a)** and standardized trends of two types of annual isoprene emissions in different ocean regions **(b, c)**.

$13\,\mathrm{m\,s^{-1}}$ is used to restrict the extent of the sea microlayer. Besides, the wind is used as input data for the exchange velocity in the sea microlayer, which directly corresponds to the cube of wind speed. This cubic relationship results in a positive correlation between SML emissions and wind speed. In Fig. 8b, wind speed shows positive correlations with SML emissions in the low-latitude regions and several coastal regions, while negative correlations appear in high-latitude regions. We believe this spatial difference is caused by the wind threshold. In the low-latitude and coastal regions, the wind keeps a relative low-level threshold CE3. Therefore, when the wind increases in these areas, the SML emissions will increase accordingly. On the contrary, wind increases in high-latitude regions lead to more grid cells with the wind speed beyond the limit, resulting in no emissions in these areas. Finally, it yields a negative relationship.

The sea surface temperature was not directly used in the calculations of BIO and SML emissions. In fact, the sea surface temperature (SST) affects the marine productivity by modifying the biological activity of phytoplankton. However, a previous study proved that the SST is only dominant in phytoplankton productivity when the nutrient conditions are not limited. This conclusion can be derived from Fig. 8c and d, in which the subtropical remote ocean region is shown with no correlations. On the other hand, phytoplankton has a suitable temperature range for its growth and metabolic processes. It explains why a positive relationship appears in the tropical ocean with higher SST, while a negative correlation is found in the high-latitude ocean which is colder. The variations in marine and meteorological factors are the result of variations in the air–sea system, suggesting that the variability in large-scale air–sea systems may contribute to the variability in marine isoprene emissions (Abbatt et al., 2019; Hacken-

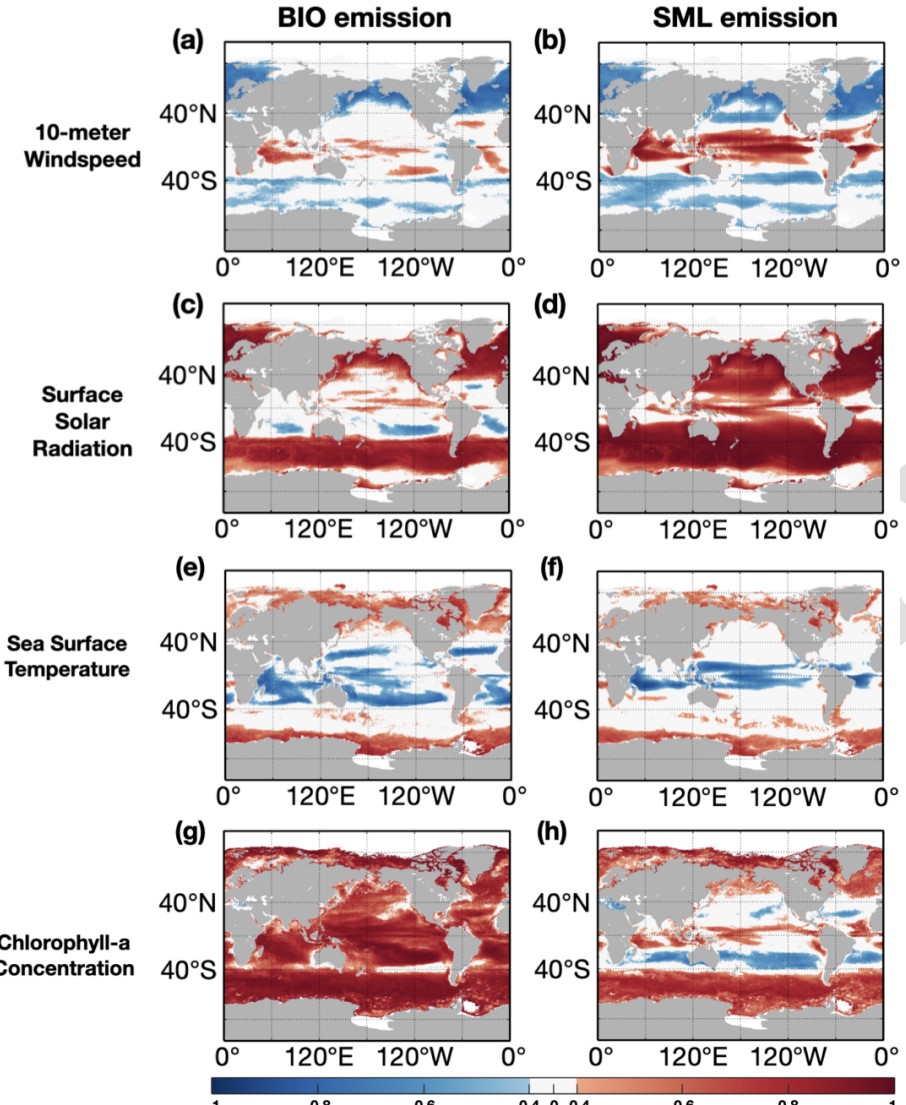

**Figure 8.** Correlation coefficients of monthly factors including 10 m wind speed **(a, b)**, downward surface solar radiation **(c, d)**, sea surface temperature **(e, f)**, and chlorophyll concentration **(g, h)** with monthly BIO emissions **(a, c, e, g)** and with monthly SML emissions **(b, d, f, h)**. Note that the grids with absolute values of correlation coefficients over 0.4 ($P \leq 0.05$) are shown in different colors.

berg et al., 2017; Xu et al., 2016; Zhang and Gu, 2022). The air–sea system plays a leading role in marine isoprene emissions. The air–sea system such as the Madden–Julian oscillation (MJO), ENSO, and Indian Ocean Dipole (IOD) may have potential influence on the marine isoprene emissions. A large-scale air–sea system is a combination of atmospheric and oceanic systems with their characteristics, mechanisms, and interactions in a large spatial range. These systems dominate the dynamic processes as well as marine and meteorological factors with their specific patterns on the global scale, especially in the tropical and subtropical areas (e.g., ENSO, MJO), where important isoprene emissions with explicit variations and spatial patterns are found. With adequate understanding of these air–sea systems, we can better com- prehend the mechanisms and characteristics of marine iso- prene emissions.

## 4.3 Potential effects of the air–sea system

In order to locate and investigate the potential impact of air– sea systems on isoprene emissions, the multiple variables empirical orthogonal function (MVEOF) was employed to examine the spatial pattern of temporal variation in BIO and SML isoprene emissions (Fig. 9). From a global per- spective, the leading MVEOF principal component (39.88 % explained variance) reveals a seasonal periodicity in both types of emissions, with a symmetrical pattern between the two hemispheres (Fig. 9a–d). In addition, the other principal components do not exhibit any distinct or meaningful spa-

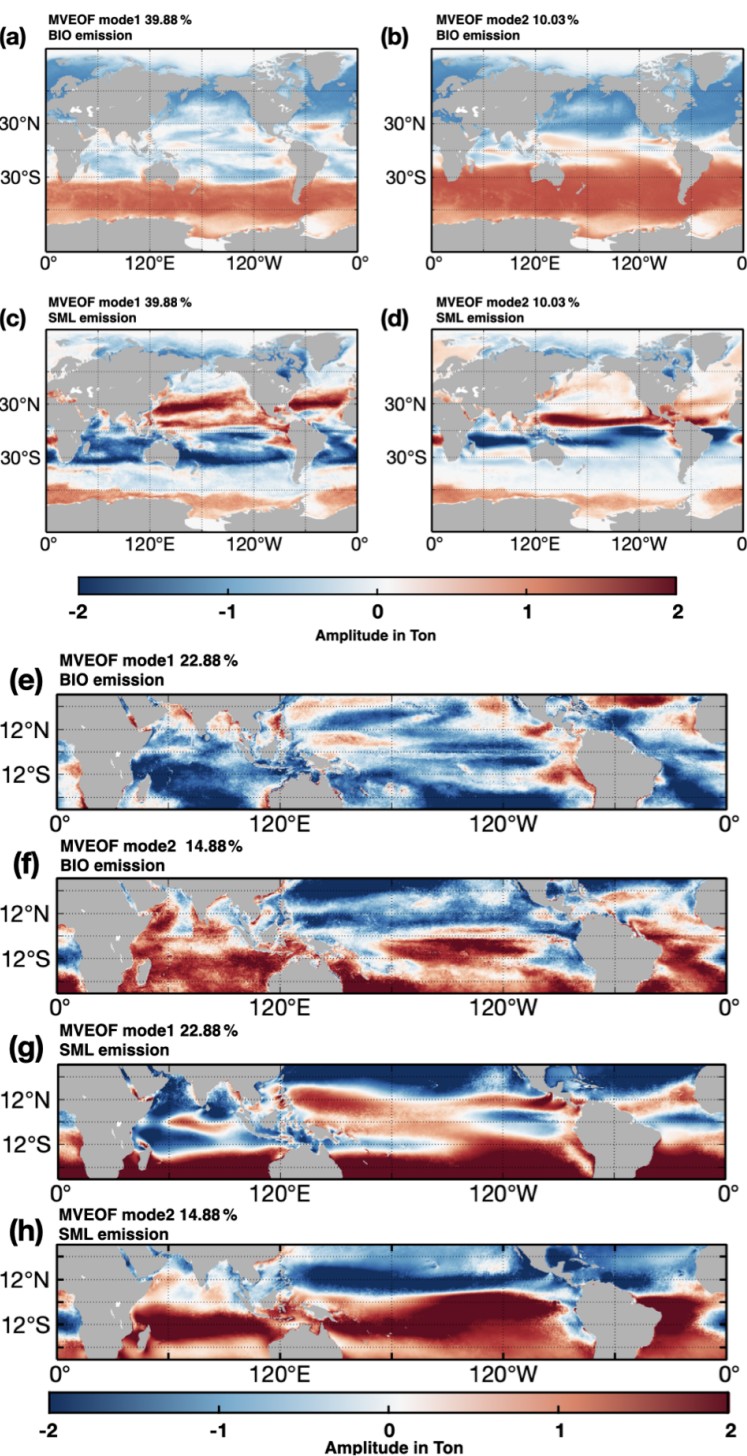

**Figure 9.** MVEOF results for global emissions **(a–d)** and low-latitude (30° N–30° S) emission **(e–h)**. **(a)**–**(b)** and **(e)**–**(f)** are the first two modes of BIO emissions, **(c–d)** and **(g–h)** are the first two modes of SML emissions. At the global scale, the explained variances for the two leading modes are 39.88 % and 10.03 %. In the low latitudes, the explained variances for the two leading modes are 22.88 % and 14.88 %. The amplitudes are in tons.

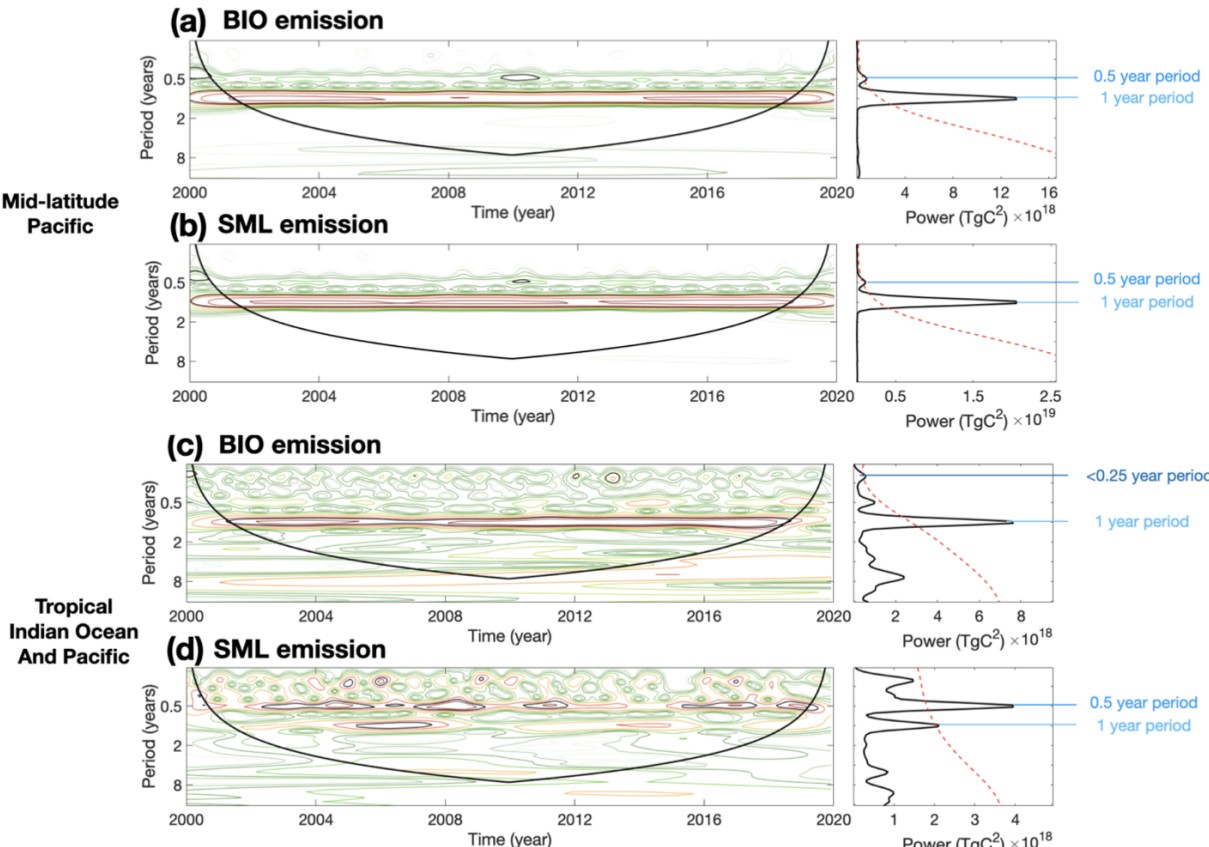

**Figure 10.** Wavelet power spectrum and time-averaged wavelet spectrum of BIO emissions **(a, c)** and SML emissions **(b, d)** of mid-latitude Pacific **(a–b)** and tropical Indian Ocean and Pacific **(c–d)**. The irregular black closed contours in the left column represent periods for which the significance level is greater than 95 %. The symmetrical solid black curve in the left column is the cone of influence. Periodic signals appear above this curve. Dashed red lines in the right column represent the 95 % significance level. The peaks of the black curves in the right row over the dashed red lines show the 95 % significance level in the 20-year period average.

tial patterns. The same analysis method was used to identify the leading potential pattern for the tropical and subtropical regions (30° N–30° S) (Fig. 9e–f). In this case, two leading EOF modes are presented, with the sum of explained variances of 37.76 %. The first mode suggests that BIO and SML emissions have different spatial patterns, in which the BIO emissions show potential opposite patterns between the coastal and remote ocean regions. The second leading mode reveals a distinct signal in the Indian Ocean, characterized by a symmetrical pattern resembling the IOD, which is a dominant quasi-periodic variation in sea surface temperature in the Indian Ocean. For SML emissions, the first mode shows an ENSO-like spatial pattern in the tropical Pacific. This suggests a connection between annual and seasonal variations in isoprene emissions and the large-scale air–sea system variability. It is likely that marine isoprene emissions are influenced by air–sea interactions, including the ENSO and other climate patterns at various scales. Previous studies have also found increased marine DMS emissions in the tropical Pacific during La Niña events due to anomalies in sea surface winds (Xu et al., 2016).

To further investigate the periodic changes in isoprene emissions and identify corresponding air–sea systems with similar cycles, wavelet analysis was applied to the monthly data. This analysis allowed us to identify significant periods in different regions. At the global scale, interannual variability is the most common and prominent for both BIO emissions and SML emissions (Fig. 10). This annual signal is largely influenced by the solar radiation cycle. Besides, a half-year period was derived from the mid-latitude and tropical SML emissions. The same period was also observed for BIO emissions in mid-latitude ocean. Furthermore, a significant intraseasonal period was found in the tropical Indian Ocean and the tropical Pacific (Fig. 10c). This period, shorter than a season (0.25 years), occurs almost every year and is believed to be associated with the MJO. The MJO is a dominant component of tropical intraseasonal variability and is associated with the large-scale signal of deep convection, which strongly affects precipitation and radiation in the trop-

ical ocean area. The periodic information is a potential indicator to find and link emission variations and driver changes. These identified periods demonstrate the potential relationships between marine isoprene emissions and variations in the air–sea system.

## 5   Data availability

The hourly global marine isoprene BIO and SML emission dataset at a spatial resolution of $0.25° \times 0.25°$ from 2001 to 2020 can be accessed directly through: https://doi.org/10.11888/Atmos.tpdc.300521 (Cui and Zhu., 2023).

## 6   Summary

In this work, a new marine isoprene emission module was built to generate a dataset of marine isoprene emissions with improved spatial and temporal resolution. This was achieved by incorporating comprehensive parameterized solutions based on remote sensing data on ocean chlorophyll concentration and reanalysis of climate data. The module considers separate parameterizations for BIO emissions and SML emissions, taking into account different physical processes. Our module estimates the total global marine isoprene emissions to be $1.097 \, \mathrm{Tg \, yr^{-1}}$ on average over a 20-year period, with $0.481 \, \mathrm{Tg \, yr^{-1}}$ attributed to BIO emissions and $0.616 \, \mathrm{Tg \, yr^{-1}}$ to SML emissions. To validate our results, several observations of marine isoprene concentrations and emission fluxes were collected for comparison with our results. These comparisons demonstrate the reasonableness and consistency of our data.

Using the hourly data, we conducted a detailed analysis of the spatial and temporal distributions of marine isoprene emissions, including their trends and periodic characteristics. On a global scale, significant disparities and variations in emissions between the SH and the NH have been observed, displaying distinct seasonal patterns. The emissions from the SH play a crucial role, particularly during the boreal winter, while the emissions in the NH amount to only half of those in the SH. Isoprene emissions are unevenly distributed across various ocean regions. Eutrophic ocean areas, such as coastal regions and eastern boundary current systems, consistently demonstrate higher marine isoprene emissions compared to remote oligotrophic ocean areas, often by orders of magnitude. We identified a slight decreasing trend in global annual isoprene emissions over the 2001–2020 period, which is dominated by a significantly decreased trend at low latitudes. Through wavelet analysis, multiple significant periods of isoprene emissions are found, including annual, semi-annual, and intraseasonal periods in different ocean regions. Several periodic and quasi-periodic signals appear in the tropical and subtropical Indian Ocean and Pacific. These findings indicate that air–sea systems drive isoprene emissions, particu-

larly in the tropical and subtropical Indian Ocean and Pacific regions. These quasi-periodic patterns and their relationships with emissions provide valuable insights for refining existing methods and improving the reliability of isoprene emission estimations. They also help bridge the gap and lessen discrepancies between observations and model calculations.

**Author contributions.** JZ conceived the research; LC and JZ designed the module, performed emission module runs, created the emission dataset and analyzed the data. YX contributed to the preparation of the module input and data processing; YX, WH, LS, YW, CZ and PF joined the discussion of the research and offered advice; LC wrote the first draft of the manuscript; JZ and LC revised the manuscript before submission with contributions from all co-authors.

**Competing interests.** The contact author has declared that none of the authors has any competing interests.

**Disclaimer.** Publisher's note: Copernicus Publications remains neutral with regard to jurisdictional claims made in the text, published maps, institutional affiliations, or any other geographical representation in this paper. While Copernicus Publications makes every effort to include appropriate place names, the final responsibility lies with the authors.

**Acknowledgements.** The monthly normalized water-leaving radiance data at 410 nm for the period 2012–2020 were provided by the NOAA Center for Satellite Applications and Research (STAR) and the CoastWatch program.

**Financial support.** This research has been supported by the National Key R&D Plan (Grant No.2022YFF0803000) and the National Natural Science Foundation of China (Grant No. 42177082).

**Review statement.** This paper was edited by Bo Zheng and reviewed by two anonymous referees.

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

## Remarks from the language copy-editor

## Remarks from the typesetter