# Peer review of "Enhanced dataset of global marine isoprene emission from biogenic and photochemical processes for the period 2001-2020"

_Earth System Science Data, 2023_

## Referee Comment (RC2)

**Review for ESSD-2023-237**

**General comments:**

This article presents a comprehensive global marine isoprene emission dataset at high spatial and temporal resolution for the period of 2001 - 2020. The authors separate marine isoprene emissions into two distinct sources (biogenic and surface microlayer). Emissions are calculated using a combination of satellite chlorophyll and radiance measurements and meteorological reanalysis data (e.g., windspeed from ECMWF's ERA-5 product) with empirical parameterizations. The estimated emissions are compared with a variety of observational records, and correlations with meteorological driving variables as well as climate modes of variability (e.g., El Niño - Southern Oscillation) are explored.

Overall, I think this is a useful and interesting dataset. These high-resolution emissions could be included in a global atmospheric chemistry model to explore the impacts of marine isoprene emissions on aerosol formation and tropospheric oxidation chemistry over the remote ocean. I think the atmospheric chemistry and climate research communities would both benefit from these data.

The article itself is reasonably clear, and the methodology is presented in a straightforward and comprehensive way. However, I have some issues with the lack of uncertainty analysis as well as the lack of justification / explanation for a few assumptions. I recommend publishing this manuscript once these concerns are addressed, because I think it would be very valuable to the global atmospheric science community.

1) I am concerned by the lack of uncertainty analysis presented in this paper. The calculation of both the biogenic ("BIO") and surface microlayer ("SML") isoprene emissions depends on satellite observations from MODIS and VIIRS, meteorological reanalysis data (ERA-5 in this case), and numerous empirical parameters derived from oceanographic or laboratory measurements. Each of these quantities has some uncertainty associated with it, and these will propagate into your emission estimate. While I appreciate that putting precise error bars on global emission estimates is not trivial, some kind of error analysis or sensitivity test seems essential in order to make proper use of your data and methods. Even something as simple as calculating the emissions with a different reanalysis product or changing the values of some of the empirical parameters would give a strong indication of how sensitive the emission estimate is to errors in the model inputs and parameters. This would make the comparison with observations and previous emission estimates more meaningful, and it would make it easier to apply your methodology in different modelling frameworks (perhaps using different meteorological reanalysis data or satellite observations).

2) There are a few assumptions and methods that need more justification / explanation. These include the assumption that isoprene concentrations in seawater are constant, that isoprene is immediately oxidized in the marine boundary layer, and the use of an 8-year plankton type distribution over a twenty-year period. Please see my specific comments for more details.

3) The plots are generally clear and of high quality, but you should include labels for the different subplots and colour bars. I found myself frequently jumping back and forth between the main text, the figures, and the figure captions in order to make sense of everything. I have included a few specific comments about this below.

4) I was able to download and explore your dataset, and I could not find any problems with the files or data structure. However, I could not find any source code for your modules on the FTP server. Perhaps this is intentional; however, your introduction made it seem like your module could be easily embedded in an Earth System Model, so I was under the impression that I would be able to download the source code and play around with it. This is not necessarily a problem, but if you don't intend to release any source code you should consider rephrasing your introduction to avoid giving the impression that people can download your model.

5) The order of the paper is odd at times, and some sections are mislabeled in the introduction (see my specific comment below). I found it strange that the BIO emissions are presented before you explain how the plankton types were calculated. I was also surprised to see the comparison with observations (Section 3) presented before you discussed the spatial and temporal characteristics of your modelled emissions (Section 4). This made it a bit harder for me to follow your overall arguments. But I acknowledge this may just be my personal preference.

**Specific Comments:**

**# # # #    *1    Introduction*    # # # #**
*Lines 40 - 42:*
There has been some work (e.g., Palmer et al 2022:
https://www.science.org/doi/10.1126/science.abg4506)
suggesting that terrestrial BVOC emissions have a large impact on downwind VOC and aerosol concentrations over the remote ocean, particularly in the South Atlantic due to the relatively long lifetime of BVOCs coming from the Amazon basin. Is it fair to say that terrestrial BVOC emissions do not exert significant influence over the remote ocean?

*Line 52:*
Here you use the terms "BIO" and "SML" without first defining them in the main text; they are explained in the abstract, but you did not explain them in the introduction.

*Line 60:*
I am not sure what this sentence means. I interpret this to mean that biochemical losses of BVOCs are parameterized based on laboratory or field observations. But it is unclear what "dynamic euphotic zone" means in this context. Are you referring to Equation (5), where you calculate the depth of the plankton euphotic zone based on surface downwelling radiation?

*Line 70:*
Which two emission pathways are you referring to here? Are you talking about the BIO and SML sources, or are you talking about photochemical and windspeed-driven processes in the

surface microlayer? I assume you mean BIO and SML based on the rest of the article, but you could avoid the ambiguity by clearly stating which pathways you're referring to.

*Lines 80-88:*
This is good background information, but it feels out of place in this paragraph. Consider moving this to the first or second paragraph instead. This information helps motivate why marine isoprene emissions are important, so I feel it should be introduced before you start describing emission estimate methodologies and uncertainties.

*Line 90:*
This is unclear to me. It would be easier to understand if you succinctly explained how you calculated the BIO and SML sources and how your approach addresses some of the uncertainties you outlined in the previous paragraph (data availability, unclear mechanisms, lack of satellite observations at high latitudes during winter, estimates of chlorophyll vertical distribution, and relations between isoprene and marine/meteorological factors). The way it is currently written, I have no way of knowing how your study plans to approach these issues until I have finished reading the paper.

*Line 91:*
The sentence "Two distinct types of emissions are separately calculated..." doesn't really make it clear what you're doing, or how you're using the MODIS chlorophyll and ECMWF reanalysis data. More broadly, I find that your introduction provides motivation for studying marine isoprene emissions and addresses some uncertainties in previous approaches, but it does not clearly explain how your new dataset was developed or how it addresses the uncertainties you mentioned. I understand that the introduction needs to be brief, and you describe these methods in detail later. But right now your introduction does not give enough information for me to tell what you actually did. After reading your introduction I should know what to expect from the rest of the paper, and right now that isn't the case.

*Line 95 - 98:*
This paragraph seems to be incorrect. I think you have swapped the descriptions of Sect 4 and Sect 5. You said "Sect. 4 provides information on our dataset and data availability", but Sect. 4 in the text is simply titled "Results" and is focused on spatio-temporal variability of emissions and correlations with climate modes of variability. Similarly for Sect 5, you said it describes the "characteristics of marine isoprene emission", but in the text Sect 5 is just a data availability statement.

**2 Methods**
*Line 101:*
What is the spatial resolution of the downwelling radiative flux diffuse attenuation coefficient data? Is it also at 9km?

*Line 106:*
These meteorological variables (u-wind and v-wind, T2M, SST, and surface downwelling shortwave flux) are all from ERA-5, right?

*Lines 107 - 110:*
Is the monthly normalized water-leaving radiance at 410 nm also at 0.25x0.25 degree spatial resolution?

*Line 110:*
Please clarify how you can apply this plankton distribution dataset over the entire twenty-year period. I understand that you use MODIS chlorophyll and NOAA water-leaving radiance at 410nm to obtain a plankton type distribution from 2012 - 2020. But it is unclear how you can use an 8-year plankton type distribution to estimate emissions over a twenty-year period.

*Lines 113 - 115:*
You assumed the concentration of isoprene in the ocean is static. Is this steady state assumption valid? What is the justification? Is it based on observations of marine isoprene concentrations, or is it based on theoretical considerations (e.g., ocean chemistry modelling)? And over what time period could we expect this assumption to be valid (Days? Weeks? Months?)? I appreciate that this assumption is very useful so that isoprene flux is equal to net isoprene production, but some more explanation / justification should be included.

*Line 116:*
How do we know isoprene will be oxidized immediately once it enters the marine boundary layer? While isoprene typically has a very short lifetime against OH oxidation, non-negligible isoprene and other BVOC mixing ratios have been measured in the marine boundary layer (e.g., Warneke et al., 2004), particularly around day-to-night transitions when OH concentrations are lower. If I understand correctly, you are assuming MBL isoprene concentrations are negligible so that you can neglect isoprene fluxes from air-to-sea, and instead focus exclusively on fluxes from sea-to-air. Can you provide some more context (i.e., why are you assuming it's negligible?) and justification (i.e., how do we know that isoprene is oxidized immediately in the MBL? What is its typical lifetime in the marine atmosphere?)?

*Line 121:*
Could you please clarify what you mean by "biochemical costs of isoprene is seawater"? Does this refer to the consumption of isoprene by biological processes, or are you talking about something else?

*Line 122:*
What kinds of observations did Simo et al 2022 use to calculate alpha? Was this relationship observed in different regions of the ocean, or did they use a small set of observations? In other words, do we think this relationship is robust enough to be applied to the global ocean?

*Lines 133 - 135:*
Can you clearly state that radiation is given by I and the plankton type coefficient is given by Tc in this sentence? Otherwise Equation (3) is unclear until after the next paragraph.

*Lines 156 - 158:*
I understand that 0.433 Tg yr-1 is only accounting for the BIO source, but you include your total estimate (BIO + SML) in Table 1. Are the other emissions estimates in Table 1 total emissions?

This is what I assumed, but the Brüggemenn et al study is listed as "Sea Surface Microlayer" so now I am not sure. Please specify in the table whether these are TOTAL, BIO, or SML emission estimates so that it is easier to compare the different studies.

Also, what is the uncertainty on your BIO estimate? Do you have an idea of how this estimate might change based on errors in the input data (e.g., MODIS chlorophyll or ERA-5 meteorological data) or model parameters?

*Line 161:*
You say these factors (temp, salinity, etc.) will lead to different isoprene production rates, but you only accounted for radiation and photic zone depth (Hmax) in equations 1 - 5. How did you account for the impact of temperature, salinity, and nutrients? Are these impacts small / negligible, or are they implicitly accounted for by the chlorophyll concentration term in Equation 2 and Equation 3?

*Line 176:*
Why was the value of 0.028 chosen? Do we expect haptophytes to dominate in oligotrophic regions of the ocean, or is there another reason you chose this value?

*Lines 180 - 183:*
Is this for coastal regions everywhere, or did these studies focus on specific regions?

*Lines 195 - 197:*
What about the large areas of undefined types in tropical and subtropical regions? In particular, Figure 1 a), b), and d) show large "undefined" areas in the southern Subtropical Pacific and western tropical pacific. Another hotspot seems to be the Arabian Sea and Bay of Bengal in Figure 1 c). Is the use of an undefined plankton type a large source of uncertainty in these regions?

*Lines 219 - 227:*
I understand that you are using chlorophyll observations as a proxy for nutrient levels and surfactant concentrations, but I do not understand where the expressions for Csurf and Cmax come from. Can you explain this? Perhaps a brief explanation of the methodology of Wurl et al 2011, or at least state the rationale behind Equation (8) and the expressions for Csurf and Cmax. Right now this method seems very opaque without having read Wurl et al 2011.

*Lines 232 - 234:*
Is this due to the destruction of the surface micro-layer at high windspeeds? Why is your wind speed threshold (13ms-1) different from the one mentioned in the introduction (10ms-1)?

What is the mean SML emission rate? At the end of Section 2.2 you gave a mean BIO estimate, so it would be nice to see the same for SML here. And just like with Section 2.2, you should address the uncertainties in this estimate (or at least explain if you will address them in a later section). Your estimate of SML emissions relies on several empirical parameters (e.g., *Flab*) and meteorological input variables. All of these quantities have uncertainties, which will propagate into your emission estimate. Some sort of error analysis or sensitivity test would be extremely valuable.

_Lines 251 - 253:_
How do these changes compare to the uncertainties on BIO and SML emissions? Is your interpolation a major source of uncertainty in your emission estimate, or are these changes significantly smaller than the other sources of error?

**3 _Evaluation and comparison_**

_Lines 272 - 277:_
How do these differences compare to the uncertainty of your estimate? For the comparison with observations to be meaningful, we need to know what kinds of errors are present in your emission estimate.

_Line 292:_
I mentioned this in an earlier section, but please explain the assumption that marine isoprene concentrations are constant. It seems to me that you are assuming concentrations are constant to estimate the fluxes, then using those fluxes to estimate the concentrations. The logic seems circular. We know the concentrations are not constant because you show large variability in both observed and modelled concentrations in Figure 4. I don't doubt that you can use some sort of steady-state approximation in order to relate isoprene production to fluxes, but this needs to be clearly explained.

_Lines 295 - 301:_
Is sea surface temperature also coming from ERA-5? Would you get very different results for Equations 10 - 12 if you used a different reanalysis product?

_Lines 309 - 313:_
Similar to my previous comments on BIO and SML estimates, can you do a sensitivity test to at least get some idea about the uncertainties? You mention that Equation (11) may introduce uncertainties which could partly explain model-observation discrepancies, but you don't quantify how big these biases might be. Even if getting a precise error estimate is difficult, it should at least be easy to figure out the impact of errors in sea surface temperature and 10-metre wind speed.

You also mention that Eq. 11 is only valid in the range of $w = 4 - 15ms-1$, so ideally you should eliminate this source of error by excluding locations and times where w falls outside of this range. Do you filter for wind speed in your simulation, or are you using all data points even if they fall outside the range of 4-15ms-1? If you are using all data points even though Eq. 11 is not valid, how big of an error might this introduce?

_Lines 325 - 328:_
These ranges are small, so I would expect that the uncertainties on these estimates are probably much larger than the reported ranges. So it would be very useful to include an error estimate here.

_Lines 345 - 360:_
If I understand correctly, the three main benefits of your approach to estimating BIO emissions are: 1) increased temporal resolution which better captures emission dynamics, 2) increased spatial

resolution which better captures the spatial heterogeneity of emissions in coastal environments, and 3) and improved plankton type distribution which resolves the issue of missing phytoplankton types in coastal regions. What is the benefit of using the new parameterization in Eq. 2, and is this significantly better than the previous approach you described in the introduction where a linear relationship between chlorophyll concentration and isoprene emission was used?

In general, I agree that there are benefits to using a higher spatial and temporal resolution, but I am not completely convinced that the updates used to calculate BIO reduce uncertainties. Your estimate depends on various satellite and ERA-5 input variables as well as laboratory-derived empirical parameters. All these quantities have their own uncertainties which will affect your BIO estimate. I think it is essential that you try to quantify this uncertainty.

*Line 364:*
Is this the main benefit of your SML approach compared to previous methods?

*Lines 365 - 367:*
I don't think this needs to be re-stated here.

**####     4     Results     ####**

*Lines 375 - 380:*
Do these statistics account for the difference in ocean surface area between the two hemispheres? (i.e., does an "average" Southern Hemisphere grid cell emit more isoprene than an "average" Northern Hemisphere grid cell, or can the difference partly be explained by the fact that there is less ocean in the Northern Hemisphere?). It's not easy to tell using the maps in Figure 6.

*Line 440:*
This vocabulary is a bit unclear to me. Are you saying that BVOC emissions in the tropical ocean are primarily governed by local / small scale atmosphere-ocean interactions (e.g., small scale weather systems)? Or am I misinterpreting something? What do you mean by "local air-sea system"?

*Line 441:*
This is why I am confused about "air-sea system". In the previous sentence you say emissions are determined by local air-sea system (local weather conditions?), and here you say large scale variability is important. Please clarify what you mean by air-sea system and what you mean by "local" versus "large-scale" air-sea system. Are we talking about weather systems or large-scale climate variability?

*Line 445:*
General comment that applies to all multi-panel plots, especially the world maps: Please include labels for the different subplots and for the colour bars. The plots themselves are clear, but it is tedious to keep going back and forth between the figure caption and the plot to make sense of what I am looking at.

*Line 449:*
Again, it is unclear what you mean by air-sea system. In this section it seems that you are describing modes of climate variability like ENSO, but in the previous section it sounded like you were describing local weather systems.

*Line 449:*
Please clarify what "identify the target area" means. Is it the region of the ocean that is affected by a particular mode of climate variability (e.g., ENSO)?

*Line 482:*
I appreciate that this is an exploratory analysis. It would be interesting if you could briefly speculate on how to verify these relationships. Are there other analyses you could do with your dataset? If not, do you know what other kinds of data / observations would help determine whether these relationships are robust? The correlations you have shown between your emission dataset and different modes of climate variability are interesting, and I think it would be beneficial if you could provide some ideas for how to use this information in follow-up studies.

*Line 484 (Figure 10):*
I have a couple problems with this figure. You didn't include a legend, so it is unclear what the different colours contours represent, what the solid black lines represent, or what the dashed red lines represent.
Also, the subplots need to be labelled so that it is obvious which emissions (BIO or SML) and which region (Mid-latitude Pacific or Tropical Indian Ocean & Pacific) we are looking at. I found myself frequently jumping back and forth between the figure, the body text, and the figure caption to make sense of the results.
The patterns you described in the text are reasonably clear (e.g., the 0.25 year signal in panel C) when you know what to look for, but it would be much easier to interpret the figure if you included a legend and labels for the subplots.

*Lines 487 - 505:*
This section needs to be expanded. The discussion here is good, but you also need to address uncertainties due to the parameterizations as well as the ERA-5 and satellite input data. Some effort needs to be made to quantify these uncertainties for other researchers to make use of these data. My concern is that if other researchers try to apply your method and get wildly different results, they won't know whether it's due to an error in their methodology or if it's an expected error due to uncertainties in the model parameters and inputs.

*# # # #        5        Data Availability        # # # #*

*Line 506:*
I see the hourly dataset at 0.25x0.25 degrees is available online. I was able to connect to the FTP server and download the files. I only downloaded a small subset due to the large size of the dataset (2.65 TB), but the files I looked at were formatter properly and I was able to make some plots with them using Python's netCDF libraries.

However, I don't see any way to access your emission module. You state on lines 93-94 that the module can be used to calculate emissions online in an Earth System Model. Are you planning to release the code for your module, or is it expected that other researchers wishing to use your method would implement it themselves based on the equations you provided? Please clarify, because the introduction made it seem like it would be possible to download the source code.

**6 Conclusions and Perspective**

*Line 522:*
You are only talking about Winter, right? In Section 4.1 you said NH emissions were 44% and SH emissions were 56%. Here it sounds like you are claiming SH emissions are twice as large as NH emissions.

*Lines 525 - 527:*
Can you explicitly connect the observed trends in Section 4.1 with the correlations observed in Section 4.2 and air-sea systems observed in Section 4.3? It would be useful to provide some more context for the reported trends.

*Lines 531 - 533:*
It's not entirely clear how these relationships will improve the accuracy of isoprene emission estimates. The connections are certainly interesting, but it is not clear what you can do with this information to improve emissions or reconcile discrepancies between observations and models.

---

## Author Comment (AC1)

**Response to the referees for comments on "Enhanced dataset of global marine isoprene emission from biogenic and photochemical processes for the period 2001-2020"**

Lehui Cui[1], Yunting Xiao[1], Wei Hu[1], Lei Song[2], Yujue Wang[3], Chao Zhang[3], Pingqing Fu[1], Jialei Zhu[1]

[1]Institute of Surface-Earth System Science, School of Earth System Science, Tianjin University, Tianjin, 300072, China

[2]Center for Monsoon System Research, Institute of Atmospheric Physics, Chinese Academy of Sciences, Beijing, 100029, China

[3]Frontiers Science Center for Deep Ocean Multispheres and Earth System, and Key Laboratory of Marine Environment and Ecology, Ministry of Education, Ocean University of China, Qingdao 266100, China

*Correspondence to*: J. Zhu (zhujialei@tju.edu.cn)

Dear Editors and Referees

Thank you for your valuable comments and suggestions. Based on these comments, we carefully reviewed all suggestions and requests and revised the manuscript accordingly. All the additions and corrections in the main text of our revised manuscript using red letters, while these updated paragraphs are quoted in *blue italics* in this response letter. The general revisions in the paper and replies to each comment are as following.

According to the common concerns of two referees, we deployed a series of sensitivity experiments for the input data, factors, assumptions and parameters used in our method and added detail analysis of the uncertainties in our method and dataset in the Sect. 3.4. In addition, the Sect. 3.4 is attached in the end of this Response Letter. The uncertainty discussion part in Sect. 4.4 of former manuscript is also merged into Sect. 3.4. Besides, we revised the improper and vague statements and descriptions in the manuscript. The structure of the manuscript is also adjusted. Furthermore, the figures and tables have been updated with Table 5 and Table 6 added for sensitivity experiments.

**Response to the comments from Referee #1**

**General comments:**

Understanding the mechanisms of isoprene production and consumption in the world oceans is crucial to finally investigate the influence of marine isoprene on climate. This work summarizes the current knowledge of production and loss processes in the surface ocean as well as the photochemical production in the SML and uses satellite and ERA5 data in order to calculate isoprene emissions with a high temporal and spatial resolution, which is of absolute importance in order to evaluate the impact of global marine isoprene emissions.

However, the authors like to stress the high temporal resolution, but they do not show any hourly or daily data. Results are shown mainly as global annual mean maps. Moreover, Chla is used as a proxy for both, the BIO as well as for the SML emission calculations, but input Chla data is on monthly basis. Apart from hourly ERA5 data which definitely is a key to describe diurnal cycles of isoprene emissions (which are also not shown) using monthly mean Chla data, is somewhat contrary to "a high temporal resolution". In a revised version of the manuscript I suggest to stress the "high temporal resolution" a bit less and concentrate on a proper discussion of the findings (in comparison to other models and/or observations). As a validation of the model, especially a proper comparison with observations is of fundamental importance.

The manuscript is very long in the method part, which is not a problem in general, but could be shortened, as some parts of the results and discussion in the method section should be moved. Comparisons to other model results or to observations are partly mentioned but more like in small bits here and there which makes it difficult to follow the flow. The authors should think about a few main scientific questions in which context they want to present, but also discuss (important!) their findings. This will help the reader to follow the "red line".

On the other hand, in some parts information is missing which is needed to understand the reasoning of the authors (see specific comments).

The manuscript additionally needs an english language revision by a native speaker.

Therefore, I suggest to revise this manuscript related to content and structure before publication.

Response to **General comments:**

We are grateful to receive your endorsement for the significance of our present work, as well as your helpful general comments.

First, thanks for your comments on the temporal resolution of our dataset. The Chla data here are used together with monthly phytoplankton type distributions to describe the monthly steady productivity in each grid cell. The Chla concentration and phytoplankton type dominate the spatial distribution of isoprene emission while the diurnal variation of radiation dominated the hourly temporal distributions. The monthly Chla concentration data was used in our module is mainly because of the property of MODIS satellite. The high temporal resolution (daily or hourly) data from MODIS can not cover all the world and show a comprehensive global distribution of Chla as we showed in the paper because of its fixed scanning period and the interference from clouds, dust and so on. In addition, the variation of marine Chla concentration is almost stable in a month. Previous studies have shown that the spring phytoplankton blooms, which account for the dramatic increasing of Chla in the regional scale, typically persist for more than a month (Yamada and Ishizaka, 2006, Friedland et al., 2015, Groetsch et al., 2016). Even these dramatic changes in Chla caused by bloom events can be captured by the monthly average, so we believe the monthly Chla data is able to describe the variation of basic ocean productivity status while the variation of the Chla within a month is reasonably ignored. However, we do realize that "high temporal resolution" is not a convincing description of the characteristic of our dataset as you commented. We have removed these descriptions of "high temporal resolution".

Secondly, thank you for your suggestion for the discussing contents of the manuscript. We enhanced the validation part of the dataset, including validation of phytoplankton types and monthly results, sensitivity experiments, and the uncertainty discussions. These contents were added in Sect. 3.4. In addition, the Sect. 3.4 is attached in the end of this Response Letter.

Finally, we appreciate your advices on the structure and content expression of the manuscript. We moved the discussion on the estimations from method section (2.2 and 2.4) to Section 3.2 with some update of previous studies comparison and corrected characteristic of the emissions. Besides, we merged the Section 4.4 (Data uncertainties) into Section 3.4 with extension by conducting a series of sensitivity experiments.

Responses to **Specific comments:**

**1. Introduction: Please streamline the introduction dependent on the main questions which will be discussed in the manuscript. The first paragraph about BVOCs is very general and could potentially be used to concentrate on isoprene (which is in similar context discussed in ll.80-88).**

Response:

Thank you for the suggestion. We have reduced description for BVOCs in the Introduction and rearranged the paragraphs to address your concerns and hope that it is now clearer.

We removed the paragraph discussed BVOCs following Line 31 and replaced with the introduction part of isoprene: *"Among all the non-methane BVOCs species, isoprene exhibits a large emission and demonstrates significant atmospheric chemical reactivity in the marine environment (Yokouchi et al., 1999; Guenther et al., 2012; Novak and Bertram, 2020). Isoprene has a lifetime of approximately 10-100 days in seawater (Booge et al., 2018). Once released into the atmosphere, it rapidly reacts with OH radicals, resulting in a short atmospheric lifetime of about one hour (Kameyama et al., 2014). Within the marine boundary layer (MBL), isoprene can undergo oxidation, leading to the formation of semi-volatile organic compounds (SVOCs) and low-volatility organic compounds (LVOCs) such as methacrolein and methacrylic acid. These compounds actively participate in the generation of marine secondary organic aerosols (SOAs) (Claeys et al., 2004; Kim et al., 2017) and plays a crucial role in aerosol growth within the MBL. The estimation of marine isoprene emission is fundamental for the future studies on marine SOAs and their climate effects (Carslaw et al., 2010)."* For the detail, please check the Introduction of the revised manuscript.

**2. Line 69: "Previous estimates…" Please be more specific and/or give references.**

Response:

We added references in Line 70: *"Previous estimates also encountered challenges related to data availability and unclear emission mechanisms, leading to uncertainties in the*

*estimated emissions (Palmer and Shaw, 2005, Gantt et al., 2009, Booge et al., 2016, Brüggemann et al., 2018, Conte et al., 2020)."*

3. **Line 88: "The subsequent section…" This paragraph is unnecessary.**

   Response:

   We finally decided to keep this paragraph but made some revision. This paragraph was change to: *"The subsequent section (Sect. 2) elucidates the methods and factors employed in our estimation of marine isoprene emissions. Our results are compared with previous isoprene emission inventories and some field observations in Sect. 3. The characteristics of the marine isoprene emission are analysed in the Sect. 4. Sect. 5 provides information on our dataset and data availability. Sect. 6 is the conclusions and discussions."* It will possibly help readers to get a comprehensive overview of this manuscript, as well as they can quickly find the section which they care about.

4. **Line 95: "...downwelling radiative flux diffuse attenuation coefficient..." Please be more specific. Is it 490nm? Somewhat later 490nm is mentioned but should be shifted to this section.**

   Response:

   This coefficient is for 490nm. We added related information in the sentences in Line 94:*"Twenty years (2001-2020) monthly average chlorophyll concentration data at 9 km resolution and 490 nm downwelling radiative flux diffuse attenuation coefficient data with the same spatial resolution were obtained from…"*

5. **Line 104: It is not clear to me how the use of the water leaving radiance at 410nm from the period 2012-2020 is matched with the general period starting in 2001. I assume that the authors used average monthly values from the period 2012-2020 in order to determine the prevalent phytoplankton types, which would be a monthly climatology. Please revise. Additionally, if somehow**

**monthly averages over time period 2012-2020 are used, this issue should be discussed later in the manuscript as all other input parameters are not averaged over 20 years.**

Response:

Thank you for the question about the input data. We used the water leaving radiance data from the period 2012-2020 and Chla concentration data of the same period to determine the phytoplankton types. Based on these data, we determined the monthly phytoplankton types for the night-year period first. Then we counted the most frequent monthly type in each grid cell for the 2012-2020 period, as the input type in our module. The variation in the global spatial distribution of phytoplankton type is dominated by the seasonal variation of radiation, temperature and the ocean trophic level, while the interannual variations of phytoplankton type in each month is of less importance. (Dandonneau et al., 2004, Uitz et al., 2010, Brewin et al., 2012). We did have considered the issue of the mismatch of the time periods between the water leaving radiance data and all other input data. The discussion on this issue is added in Sect. 3.4 (Line 423): *"Our module used the dominant phytoplankton type for each month without hourly and daily variations due to the restriction of temporal resolution of measured chlorophyll-a and water leaving radiance data. We simply diagnosed the monthly phytoplankton types during period of 2012-2020. The phytoplankton types in 51 % of global grid cells are same in the all nine-year period, while the types in 89% of the grid cells are same for more than five years. As a result, we believe it is reliable to apply the monthly dominant phytoplankton type in each grid during 2012-2020 in the estimation during all twenty years (2001-2020)."*

6. **Line 119: "..biological costs...". This wording sounds wrong to me and appears a few times in the manuscript.**

   **BIO emission module: It is not clear to me how the authors use equation 2 (from eq. 3 in Simó et al., (2022)) by phrasing that the resulting value α is dimensionless. In the correlation from Simó et al. (2022) the chl-a dependent loss term is a rate constant in 1/day. Please elaborate on this. Also, how is the**

**maximum value α=0.373 (when chl-a conc. is higher than 5.77 mg/m3) calculated?.**

Response:

The words: "biochemical cost" are replaced with the clearer statement: "biological and chemical consumption" in Line 119.

Thank you for the question. We gratefully thank the referee for pointing this mistake out. First, the equation 2 in the manuscript is an incorrect citation. We actually used the equation 1 in Simó et al., (2022) to calculate biological and chemical consumption term in our BIO emission module, which is $\alpha = 0.1 \times C_{chl} + 0.05$. This linear equation was regressed using the data in Fig. 3a of Simó et al., (2022), with a convincible coefficient of determination ($R^2 = 0.96$). We got the α=0.373 when substituted 5.77 mg·m$^{-3}$ into the equation.

In addition, we applied the Eq. 1 in Simó et al., (2022) instead of the Eq. 3 to determine the loss term because we tried to decrease the uncertainties in the transition from fluorometric result to satellite Chla concentrations. The coefficient of determination of Eq. 2 ($R^2 = 0.66$) (Eq. 2 gives the transition from fluorometric result to satellite result) is much lower than that of Eq. 1. In the former manuscript, we incorrectly used the consumption rate constant (α) with the daily consumption in the hourly calculation by mistake. We have revised this issue in the updated module with the correct rate constant for hourly consumption.

Therefore, the BIO emission in the dataset and related conclusions are revised. The main characteristics of the revised BIO emission were updated in the manuscript and listed here (in Line 508 (Sect. 4.1)):

*"Generally, our dataset suggests annual global marine isoprene emissions ranging from 1.075 to 1.112 Tg·yr$^{-1}$ for the period 2001-2020, with an average of 1.097 Tg·yr$^{-1}$ over the twenty years. Annual average global BIO emissions for the twenty-year period were 0.481 Tg·yr$^{-1}$, ranging from 0.464 to 0.493 Tg·yr$^{-1}$, while annual average global SML emissions was 0.616 Tg·yr$^{-1}$, ranging from 0.611 to 0.621 Tg·yr$^{-1}$."*

The revised estimation and conclusion of BIO emission have the similar spatial pattern and temporal variations, while the global annual mean BIO emission increase about 11%. We have updated all data and figures through the paper accordingly.

We apologize for our mistake. Related correction of the dataset has been submitted to the data center. Thank you again for pointing out this essential and crucial incorrectness.

7. **Line 158: "The mean annual..." This is a result and should be moved to the result section.**

   Response:

   We have moved these discussions of the results together with the Table 1 to Section 3.2.

8. **Line 179: I absolutely agree to fill grid cells with numbers if there are missing values in order to avoid underestimation of isoprene emissions. However, the authors should justify why a coefficient of 0.028 is used in areas with chl-a conc lower 0.04 mg/m3 or areas with missing values.**

   Response:

   Thank you for your suggestions on adding the justification of the usage for coefficient of 0.028 for haptophyte. The haptophyte is a wide-spread marine producer, which dominates Chla-normalized phytoplankton standing stock in modern oceans (Liu et al., 2009). This phytoplankton type was replaced by "nanoeukaryotes" in a previous method (Alvain et al., 2005) therefore used by Gantt (2009) in their marine isoprene estimate method.

   Haptophytes dominant the global ocean all year-long, with contribution varies from 45% to 70% depending on the seasons (Alvain et al., 2005). Because of its small cell volume with relatively large surface extent, this species dominant the oligotrophic waters. Therefore, we decided to use the coefficient of 0.028 for haptophyte in the oligotrophic waters where Chla concentration lower 0.04 mg·m$^{-3}$ and area with missing value as suggested in Alvain (2005).

We added sentences to explain this assumption in Line 175: *"The haptophyte is a wide-spread marine producer, which dominates Chla-normalized phytoplankton standing stock in modern oceans (Liu et al., 2009). Haptophytes dominant the global ocean all year-long, with contribution varies from 45% to 70% depending on the seasons (Alvain et al., 2005). Because of its small cell volume with relatively large surface extent, this species dominant the oligotrophic waters. Therefore, we decided to use the coefficient of 0.028 for haptophyte in the oligotrophic waters where Chla concentration lower 0.04 mg·m$^{-3}$ and area with missing value as suggested in Alvain (2005)."*

9. **Line 185: "…a combination of 50% diatoms and 50% haptophytes in the grids…". In the adjacent Table 2 it says "50% other types + 50% diatoms". Please change accordingly.**

   Response:

   We revised the word in Table 1: "50% other types + 50% diatoms" to "50% diatoms + 50% haptophytes" accordingly.

10. **Line 195: until end of paragraph: This section belongs to the discussion section.**

    Response:

    We removed the sentence: "Development of inversion technique of remote data and marine observation are required to improve the BIO emission dataset in the high latitudes in the future."

11. **Line 221: The citation of Flab is incorrect. It reads that the authors calculated a mean Flab value from the published values by Ciuraru et al. (2015a) and Conte et al. (2020). However, the authors use an average Flab value which already was calculated by Conte et al. (2020) and is dependent on the data from Brüggemann et al. (2017), Ciurau et al. (2015a) and Ciuraru et al. (2015b).**

    Response:

We revised the description and citation of $F_{lab}$ in Line 221: *"$F_{lab}$ = 4.95×10⁷ is used in this work, which represents the mean value within the range (3.71×10⁷-6.19×10⁷) used by Conte, depending on the data from Brüggmann and Ciuraru (Ciuraru et al., 2015a; Brüggemann et al., 2017; Conte et al., 2020)."*

**12. Line 223: μ(photo) is not the radiation intensity in mW/m2. According to Brüggemann et al. (2018) it is the photochemical emission potential. Please revise wording and units. Table 4 and Figure 3: Both, table and figure, present isoprene emission values. However, those values are hardly comparable as they are noted in three different units. Please just use consistent units throughout the manuscript.**

Response:

Thank you for your correction of the citation content. $\mu_{photo}$ is the photochemical emission potential, while the $E_{280-400}$ is radiation intensity in the Eq. 7 of the manuscript. The unit of the $\mu_{photo}$ we used is mW·m$^{-2}$, accounting for the hourly emission potential, which has a little difference with the unit in Bruggemann et al.'s work. Because we used hourly radiation data, the length of the day does not need to be considered in our work. We have revised the description of $\mu_{photo}$ in the manuscript in Line 223: *"S (m²) is the grid cell area and $\mu_{photo}$ (mW·m$^{-2}$) is photochemical emission potential. The calculation of $\mu_{photo}$ is determined by Eq. (7):*

$$\mu_{photo} = E_{280-400} \times F_{surf} \times k_{SML} \tag{7}$$

*Where $E_{280-400}$ (mW·m$^{-2}$) is radiation intensity, which accounts for radiation between 280 and 400 nm reaching the surface of the ocean. It is determined to be 3.535 % of the surface downward solar radiation (Conte et al., 2020)."*

Besides, we have revised the units in Table 4 and Figure 3 to use unit of μg·m$^{-2}$·d$^{-1}$ in both.

**13. Line 316: The citation for the Schmidt number is incorrect. The correct reference is Palmer and Shaw (2005).**

Response:

The citation of Schmidt number has been corrected in Line 316: *"Notes that Eq. (11) is valid with $w$ in the range of 4-15 m·s⁻¹. $Sc$ is Schmitt number determined by sea surface temperature (Palmer and Shaw, 2005)"*.

**14. Line 367: The authors highlight their use of hourly data "...which probably provides a more accurate representation of emission dynamics." This sounds very vague. On the other hand, the authors could actually proof this using their dataset and compare to results using a lower temporal or spatial resolution and explicitly show the improvements (also in relation to one of the general comments).**

Response:

Thank you for your suggestion and concern, as well as the suggestion in the general comments. In order to test the sensitivity of temporal resolution of input data, we tried to calculate the marine isoprene emission using the same method described in our manuscript but applied monthly data instead of hourly data of radiation and wind speed. We found the annual isoprene emission is 1.050 Tg·yr⁻¹ when use monthly average radiation and wind speed, which is underestimated by 4% compared to the estimation using hourly radiation in the manuscript. Among this, the annual SML emission is underestimated by 19% while the annual BIO emission overestimated by 15%. The deviation of BIO emission is mainly accounted by the accordance of the radiation data and its temporal resolution, which caused a fixed depth of euphotic layer for every month. The deviation of SML emission is from the monthly mean windspeed data. High windspeed is eliminated by the monthly average, while the SML emission is directly corresponded with the windspeed cubed. This discussion is added in the revised manuscript (Line 369): *"The hourly windspeed data perform better in the calculation of SML emission. The SML emission directly correspond to the cube of windspeed (Eq. 6, 7, 9), so that the high windspeed is of large contributions. High windspeed can be captured in hourly data, while monthly average eliminates high*

*windspeed, which results in a relative underestimation of SML emission using monthly windspeed data as input."*

15. **Line 511 and Figure 5: The authors provide results of contribution (in percent) of different areas to the total isoprene emissions. Please provide information if these numbers are area normalized or if those emissions are absolute numbers. In the ladder, it is not surprising that the Southern Hemisphere contributes more to the total isoprene flux than the Northern Hemisphere does, just because of a larger oceanic surface area.**

Response:

Thank you for the professional suggestion. We compared absolute emission in the Northern Hemisphere and Southern Hemisphere in the previous version. We agree with you that the larger oceanic surface area plays a considerable role in this difference. The emission per unit area was added in this revised version. We investigated the area normalized emission contribution, and revised the paragraph in Line 511: *"In the twenty-year period, the average annual emissions in the Northern Hemisphere amounted to approximately 44.9 %, whereas the Southern Hemisphere accounted for 55.1 % of the total emissions. However, the emission per unit area in NH (3.3 mg·m$^{-2}$·yr$^{-1}$) is 6.5% larger than that in SH (3.1 mg·m$^{-2}$·yr$^{-1}$) due to the larger and better nutritional status of coastal ocean areas in NH. The difference in the total emissions between two hemispheres is largest in boreal winter (Fig. 5). The emission in the boreal winter of the Southern Hemisphere contributed 17.7 % of annual global emissions in average, while the emission in the same season of the Northern Hemisphere accounted for only 8.7 %."* Meanwhile, the emission per unit area in NH (0.70 mg·m$^{-2}$) was still smaller than that in SH (0.85 mg·m$^{-2}$) in the boreal winter. Radiation and duration of day dominate the seasonal variations of total emissions directly and indirectly through their influences on the distribution of chlorophyll concentration."*

16. **Line 533: "The emission rates in coastal areas...larger...by several orders of magnitude." Please provide numbers, as this are results from your work. Also,**

**I do not see this difference in the described Figure 6**

Response:

Thank you for your comment. The comparison between isoprene emissions in coastal areas and remote ocean is added in the manuscript (Line 534): *"In the twenty-year period, the mean isoprene BIO emission per unit area in the coastal ocean (e.g. East Asia, 110E-130E, 40N-20N) is 0.273 $\mu g \cdot m^{-2} \cdot h^{-1}$, while the average emission is 0.076 $\mu g \cdot m^{-2} \cdot h^{-1}$ in remote ocean area (e.g. Subtropic Pacific, 180W-120W, 20S-30S). The global average BIO emission per unit area is 0.141 $\mu g \cdot m^{-2} \cdot h^{-1}$."* For your comment on Figure 6, we have updated coordinates and colorbars to show the difference clearer.

**17. Line 540: Description of SML emission results is missing.**

Response:

Thanks for the comment. We have added *"The spatial distribution of SML emissions are more uniform than that of BIO emissions and limited in range. Indirect use of chlorophyll data contributed to this characterization, in which the surfactant concentrations were determined from chlorophyll and divided into three bins. Therefore, SML emissions are insensitive to chlorophyll concentration, which results in a different spatial pattern between SML emissions and chlorophyll. SML emissions contribute relatively large isoprene emission in the subtropic remote ocean. In these regions, SML emissions are dominated by radiation and windspeed, This relationship is further discussed in Sect. 4.2."* in Line 540.

**18. Line 554: The Arctic Ocean shows an increasing trend of isoprene emissions which is different to the Pacific or Indian Ocean. Perhaps the authors could discussion this within the context of sea ice retreat? Could that be a reason?**

Response:

Thank you for the insightful question. The increasing trend of isoprene emissions in the Arctic Ocean does exist according to our dataset. We have discussed possible reasons for this trend in Line 556 as: *"This increasing trend in the Arctic ocean was probably*

*attributed to the shrinkage of the sea ice extent and reduction of the sea ice concentration in recent decades lead to the increase in both emission area and period in boreal summer based on sea ice concentration in recent decades lead to increase in both emission area and period in boreal summer. Additionally, recent research suggests that along with the ice-free area lasting longer, the novel fall phytoplankton blooms are more likely to happen (Ardyna et al., 2014). The bloom events may contribute to the increasing of isoprene emission potentially.".* With the retreat of Arctic sea ice to the higher latitude in the boreal summer in recent decades, the Chla was detected by MODIS in increasing area in the Arctic, which cause the increase in the marine isoprene emission.

**19. Line 564 and Figure 7: The Atlantic trends are shown in Figure 7c. However, this subfigure is not discussed in the text.**

Response:

Thank you for the suggestion. We added the related discuss for the trend in Atlantic in Line 564: *"In addition, the SML emission in Atlantic is also shows a decreasing trend, while the BIO emission there has no specific trend in the twenty-year period."*

**20. Line 616: correlation plots. What temporal resolution of datasets were used to perform the correlation calculations per each grid cell? Hourly, daily, monthly data? This information is missing in the text.**

Response:

The correlations in Figure 7 are calculated based on the monthly data of emission and emission factors. This information is added in the figure legend in Line 616: *"Correlation coefficients of monthly factors including 10-meter windspeed (a, b) , surface solar radiation downward (c, d), sea surface temperature (e, f) and chlorophyll concentration (g, h) with monthly BIO emissions (a, c, e, g) and with monthly SML emissions (b, d, f, h)."*

**21. Line 587: "These two physical factors...show contrasting correlations". The is**

**an issue which should be discussed in the manuscript. How do these factors (SST and wind) influence the different emission modules (BIO and SML)?**

Response:

Thank you for the suggestion. We added the following paragraph in Line 590: *"The wind mainly contributed to the SML emission. First, it determined the surfactant coverage on the ocean surface. A wind threshold of 13 $m \cdot s^{-1}$ is used to restrict the extent of sea micro-layer. Besides, the wind is input data for exchange velocity in sea micro-layer, which is directly correspond to the cube of wind. This cubic relationship makes the SML emissions positive correlated to the wind. In Fig. 8b, wind speed shows positive correlations with SML emissions in the low-latitude and several coastal regions, while negative correlations appear in high-latitude. This spatial difference is probably caused by the distribution of wind speed. In the low-latitude and coastal regions, the wind speed is low compared to the wind threshold, so the SML emission increases with the increase in the wind speed. On the country, the wind in high-latitude is always close to or beyond the wind threshold. As a result, the surfactant layer may destruct with increase in the wind speed leading to the sharp decrease in the SML emission.*

*The sea surface temperature is not directly used in both BIO and SML emissions calculations. In fact, the SST affects the marine productivity by modifying the biological activity of phytoplankton. However, previous study proved that the SST dominates the phytoplankton productivity when the nutrient conditions is not limited. Both BIO and SML emissions show no correlations with SST in the subtropical remote ocean and Southern ocean according to Fig. 8e and 8f. On the other hand, there is a suitable temperature range for the growth and metabolic processes of phytoplankton, so the negative relationship occurs in the tropical ocean where SST is high while negative correlation occurs in the high latitude ocean where SST is low."*

**22. Line 609: "It is not clear to me what the authors mean with "large-scale air-sea system" within this context. Perhaps they can be a more specific.**

Response:

Thank you for your question. The "large-scale air-sea system" here leads the further discussion on El niño-Southern Oscillation (ENSO) and Indian Ocean Dipole (IOD), which may affect the isoprene emission through the correspond meteorological factors. According to the analysis of the correlation between emissions and meteorological factors, these air-sea systems have the abilities to affect the marine isoprene emission potentially. We will start a further study to discuss these mechanisms and processes in detail. Here we updated with a more specific description of the "large-scale air-sea system" in Line 609:

*"Large-scale air-sea system is a combination of atmospheric and oceanic systems with their characteristics, mechanisms and interactions in a large spatial range. These systems dominate the dynamic processes as well as oceanographical and meteorological factors with specific patterns on the global scale, especially in tropical and subtropical regions (e.g. ENSO, MJO), where large isoprene emissions with distinct temporal and spatial variations are found. The mechanisms and characteristics of marine isoprene emissions can be further investigated with the variation in these air-sea systems."*

23. **The paragraph "Conclusions and Perspective" should be changed to "Summary" if the content stays as is.**

Response:

We have revised it into "Summary".

**Special thanks to you for the very constructive comments!**

**Section 3.4 Data uncertainty**

[revised manuscript text omitted]

Bonsang, B., Gros, V., Peeken, I., Yassaa, N., Bluhm, K., Zoellner, E., Sarda-Esteve, R., & Williams, J. (2010). Isoprene emission from phytoplankton monocultures: the relationship with chlorophyll-a, cell volume and carbon content. Environmental Chemistry, 7(6), 554-563. https://doi.org/10.1071/En09156

Bonsang, B., Polle, C., & Lambert, G. (1992). Evidence for marine production of isoprene. Geophysical Research Letters, 19(11), 1129-1132. https://doi.org/https://doi.org/10.1029/92GL00083

Booge, D., Marandino, C. A., Schlundt, C., Palmer, P. I., Schlundt, M., Atlas, E. L., Bracher, A., Saltzman, E. S., & Wallace, D. W. R. (2016). Can simple models predict large-scale surface ocean isoprene concentrations? Atmospheric Chemistry and Physics, 16(18), 11807-11821. https://doi.org/10.5194/acp-16-11807-2016

Booge, D., Schlundt, C., Bracher, A., Endres, S., Zancker, B., & Marandino, C. A. (2018). Marine isoprene production and consumption in the mixed layer of the surface ocean - a field study over two oceanic regions. Biogeosciences, 15(2), 649-667. https://doi.org/10.5194/bg-15-649-2018

Brewin, R. J. W., Hirata, T., Hardman-Mountford, N. J., Lavender, S. J., Sathyendranath, S., & Barlow, R. (2012). The influence of the Indian Ocean Dipole on interannual variations in phytoplankton size structure as revealed by Earth Observation. Deep-Sea Research Part Ii-Topical Studies in Oceanography, 77-80, 117-127. https://doi.org/10.1016/j.dsr2.2012.04.009

Bruggemann, M., Hayeck, N., Bonnineau, C., Pesce, S., Alpert, P. A., Perrier, S., Zuth, C., Hoffmann, T., Chen, J. M., & George, C. (2017). Interfacial photochemistry of biogenic surfactants: a major source of abiotic volatile organic compounds. Faraday Discussions, 200, 59-74. https://doi.org/10.1039/c7fd00022g

Bruggemann, M., Hayeck, N., & George, C. (2018). Interfacial photochemistry at the ocean surface is a global source of organic vapors and aerosols. Nature Communications, 9. https://doi.org/ARTN 2101

10.1038/s41467-018-04528-7

Carslaw, K. S., Boucher, O., Spracklen, D. V., Mann, G. W., Rae, J. G. L., Woodward, S., & Kulmala, M. (2010). A review of natural aerosol interactions and feedbacks within the Earth system. Atmospheric Chemistry and Physics, 10(4), 1701-1737. <Go to ISI>://WOS:000274851500015

Ciuraru, R., Fine, L., van Pinxteren, M., D'Anna, B., Herrmann, H., & George, C. (2015). Unravelling New Processes at Interfaces: Photochemical Isoprene Production at the Sea Surface. Environmental Science & Technology, 49(22), 13199-13205. https://doi.org/10.1021/acs.est.5b02388

Ciuraru, R., Fine, L., van Pinxteren, M., D'Anna, B., Herrmann, H., & George, C. (2015). Photosensitized production of functionalized and unsaturated organic compounds at the air-sea interface. Scientific Reports, 5. https://doi.org/ARTN 12741

10.1038/srep12741

Claeys, M., Wang, W., Ion, A. C., Kourtchev, I., Gelencser, A., & Maenhaut, W. (2004). Formation of secondary organic aerosols from isoprene and its gas-phase oxidation products through reaction with hydrogen peroxide. Atmospheric Environment, 38(25), 4093-4098. <Go to ISI>://WOS:000222706000001

Conte, L., Szopa, S., Aumont, O., Gros, V., & Bopp, L. (2020). Sources and Sinks of Isoprene in the Global Open Ocean: Simulated Patterns and Emissions to the Atmosphere. Journal of Geophysical Research-Oceans, 125(9). https://doi.org/ARTN e2019JC015946 10.1029/2019JC015946

Dandonneau, Y., Deschamps, P. Y., Nicolas, J. M., Loisel, H., Blanchot, J., Montel, Y., Thieuleux, F., & Becu, G. (2004). Seasonal and interannual variability of ocean color and composition of phytoplankton communities in the North Atlantic, equatorial Pacific and South Pacific. Deep-Sea Research Part Ii-Topical Studies in Oceanography, 51(1-3), 303-318. https://doi.org/10.1016/j.dsr2.2003.07.018

Dani, K. G. S., & Loreto, F. (2017). Trade-Off Between Dimethyl Sulfide and Isoprene Emissions from Marine Phytoplankton. Trends in Plant Science, 22(5), 361-372. https://doi.org/10.1016/j.tplants.2017.01.006

Friedland, K. D., Leaf, R. T., Kane, J., Tommasi, D., Asch, R. G., Rebuck, N., Ji, R., Large, S. I., Stock, C., & Saba, V. S. (2015). Spring bloom dynamics and zooplankton biomass response on the US Northeast Continental Shelf. Continental Shelf Research, 102, 47-61. https://doi.org/10.1016/j.csr.2015.04.005

Gantt, B., Meskhidze, N., & Kamykowski, D. (2009). A new physically-based quantification of marine isoprene and primary organic aerosol emissions. Atmospheric Chemistry and Physics, 9(14), 4915-4927. https://doi.org/DOI 10.5194/acp-9-4915-2009

Groetsch, P. M. M., Simis, S. G. H., Eleveld, M. A., & Peters, S. W. M. (2016). Spring blooms in the Baltic Sea have weakened but lengthened from 2000 to 2014. Biogeosciences, 13(17), 4959-4973. https://doi.org/10.5194/bg-13-4959-2016

Guo, S. J., Feng, Y. Y., Wang, L., Dai, M. H., Liu, Z. L., Bai, Y., & Sun, J. (2014). Seasonal variation in the phytoplankton community of a continental-shelf sea: the East China Sea. Marine Ecology Progress Series, 516, 103-126. https://doi.org/10.3354/meps10952

Kameyama, S., Yoshida, S., Tanimoto, H., Inomata, S., Suzuki, K., & Yoshikawa-Inoue, H. (2014). High-resolution observations of dissolved isoprene in surface seawater in the Southern Ocean during austral summer 2010-2011. Journal of Oceanography, 70(3), 225-239. https://doi.org/10.1007/s10872-014-0226-8

Kim, M. J., Novak, G. A., Zoerb, M. C., Yang, M. X., Blomquist, B. W., Huebert, B. J., Cappa, C. D., & Bertram, T. H. (2017). Air-Sea exchange of biogenic volatile organic compounds and the impact on aerosol particle size distributions. Geophysical Research Letters, 44(8), 3887-3896. https://doi.org/10.1002/2017gl072975

Liu, H., Probert, I., Uitz, J., Claustre, H., Aris-Brosou, S., Frada, M., Not, F., & de Vargas, C. (2009). Extreme diversity in noncalcifying haptophytes explains a major pigment paradox in open oceans. Proceedings of the National Academy of Sciences of the United States of America, 106(31), 12803-12808. https://doi.org/10.1073/pnas.0905841106

Liu, X., Xiao, W. P., Landry, M. R., Chiang, K. P., Wang, L., & Huang, B. Q. (2016). Responses of Phytoplankton Communities to Environmental Variability in the East China Sea. Ecosystems, 19(5), 832-849. https://doi.org/10.1007/s10021-016-9970-5

Milne, P. J., Riemer, D. D., Zika, R. G., & Brand, L. E. (1995). Measurement of vertical distribution of isoprene in surface seawater, its chemical fate, and its emission from several phytoplankton monocultures. Marine Chemistry, 48(3), 237-244. https://doi.org/https://doi.org/10.1016/0304-4203(94)00059-M

Moore, R. M., & Wang, L. (2006). The influence of iron fertilization on the fluxes of methyl halides and isoprene from ocean to atmosphere in the SERIES experiment. Deep-Sea Research Part Ii-Topical Studies in Oceanography, 53(20-22), 2398-2409. https://doi.org/10.1016/j.dsr2.2006.05.025

Palmer, P. I., Marvin, M. R., Siddans, R., Kerridge, B. J., & Moore, D. P. (2022). Nocturnal survival of isoprene linked to formation of upper tropospheric organic aerosol. Science, 375(6580), 562-+. https://doi.org/10.1126/science.abg4506

Palmer, P. I., & Shaw, S. L. (2005). Quantifying global marine isoprene fluxes using MODIS chlorophyll observations. Geophysical Research Letters, 32(9). https://doi.org/Artn L09805 10.1029/2005gl022592

Shaw, S. L., Gantt, B., & Meskhidze, N. (2010). Production and Emissions of Marine Isoprene and Monoterpenes: A Review. Advances in Meteorology, 2010. https://doi.org/Artn 408696 10.1155/2010/408696

Simo, R., Cortes-Greus, P., Rodriguez-Ros, P., & Masdeu-Navarro, M. (2022). Substantial loss of isoprene in the surface ocean due to chemical and biological consumption. Communications Earth & Environment, 3(1). https://doi.org/ARTN 20

10.1038/s43247-022-00352-6

Sinha, V., Williams, J., Meyerhofer, M., Riebesell, U., Paulino, A. I., & Larsen, A. (2007). Air-sea fluxes of methanol, acetone, acetaldehyde, isoprene and DMS from a Norwegian fjord following a phytoplankton bloom in a mesocosm experiment. Atmospheric Chemistry and Physics, 7, 739-755. https://doi.org/DOI 10.5194/acp-7-739-2007

Uitz, J., Claustre, H., Gentili, B., & Stramski, D. (2010). Phytoplankton class-specific primary production in the world's oceans: Seasonal and interannual variability from satellite observations. Global Biogeochemical Cycles, 24. https://doi.org/Artn Gb3016

10.1029/2009gb003680

Wanninkhof, R. (2014). Relationship between wind speed and gas exchange over the ocean revisited. Limnology and Oceanography-Methods, 12, 351-362. https://doi.org/10.4319/lom.2014.12.351

Warneke, C., de Gouw, J. A., Goldan, P. D., Kuster, W. C., Williams, E. J., Lerner, B. M., Jakoubek, R., Brown, S. S., Stark, H., Aldener, M., Ravishankara, A. R., Roberts, J. M., Marchewka, M., Bertman, S., Sueper, D. T., McKeen, S. A., Meagher, J. F., & Fehsenfeld, F. C. (2004). Comparison of daytime and nighttime oxidation of biogenic and anthropogenic VOCs along the New England coast in summer during New England Air Quality Study 2002. Journal of Geophysical Research-Atmospheres, 109(D10). https://doi.org/Artn D10309

10.1029/2003jd004424

Wohl, C., Jones, A. E., Sturges, W. T., Nightingale, P. D., Else, B., Butterworth, B. J., & Yang, M. X. (2022). Sea ice concentration impacts dissolved organic gases in the Canadian Arctic. Biogeosciences, 19(4), 1021-1045. https://doi.org/10.5194/bg-19-1021-2022

Wohl, C., Li, Q. Y., Cuevas, C. A., Fernandez, R. P., Yang, M. X., Saiz-Lopez, A., & Simo, R. (2023). Marine biogenic emissions of benzene and toluene and their contribution to secondary organic aerosols over the polar oceans. Science Advances, 9(4). https://doi.org/ARTN eadd9031

10.1126/sciadv.add9031

Wurl, O., Wurl, E., Miller, L., Johnson, K., & Vagle, S. (2011). Formation and global distribution of sea-surface microlayers. Biogeosciences, 8(1), 121-135. https://doi.org/10.5194/bg-8-121-2011

Xu, L., Cameron-Smith, P., Russell, L. M., Ghan, S. J., Liu, Y., Elliott, S., Yang, Y., Lou, S., Lamjiri, M. A., & Manizza, M. (2016). DMS role in ENSO cycle in the tropics. Journal of Geophysical Research-Atmospheres, 121(22), 13537-13558. https://doi.org/10.1002/2016jd025333

Yamada, K., & Ishizaka, J. (2006). Estimation of interdecadal change of spring bloom timing, in the case of the Japan Sea. Geophysical Research Letters, 33(2). https://doi.org/Artn L02608 10.1029/2005gl024792

Yu, Z. J., & Li, Y. (2021). Marine volatile organic compounds and their impacts on marine aerosol-A review. Science of the Total Environment, 768. https://doi.org/ARTN 145054 10.1016/j.scitotenv.2021.145054

---

## Author Comment (AC2)

**Response to the referees for comments on "Enhanced dataset of global marine isoprene emission from biogenic and photochemical processes for the period 2001-2020"**

Lehui Cui[1], Yunting Xiao[1], Wei Hu[1], Lei Song[2], Yujue Wang[3], Chao Zhang[3], Pingqing Fu[1], Jialei Zhu[1]

[1]Institute of Surface-Earth System Science, School of Earth System Science, Tianjin University, Tianjin, 300072, China

[2]Center for Monsoon System Research, Institute of Atmospheric Physics, Chinese Academy of Sciences, Beijing, 100029, China

[3]Frontiers Science Center for Deep Ocean Multispheres and Earth System, and Key Laboratory of Marine Environment and Ecology, Ministry of Education, Ocean University of China, Qingdao 266100, China

*Correspondence to*: J. Zhu (zhujialei@tju.edu.cn)

Dear Editors and Referees

Thank you for your valuable comments and suggestions. Based on these comments, we carefully reviewed all suggestions and requests and revised the manuscript accordingly. All the additions and corrections in the main text of our revised manuscript using red letters, while these updated paragraphs are quoted in *blue italics* in this response letter. The general revisions in the paper and replies to each comment are as following.

According to the common concerns of two referees, we deployed a series of sensitivity experiments for the input data, factors, assumptions and parameters used in our method and added detail analysis of the uncertainties in our method and dataset in the Sect. 3.4. In addition, the Sect. 3.4 is attached in the end of this Response Letter. The uncertainty discussion part in Sect. 4.4 of former manuscript is also merged into Sect. 3.4. Besides, we revised the improper and vague statements and descriptions in the manuscript. The structure of the manuscript is also adjusted. Furthermore, the figures and tables have been updated with Table 5 and Table 6 added for sensitivity experiments.

**Response to the comments from Referee #2**

**General comments:**

This article presents a comprehensive global marine isoprene emission dataset at high spatial and temporal resolution for the period of 2001 - 2020. The authors separate marine isoprene emissions into two distinct sources (biogenic and surface microlayer). Emissions are calculated using a combination of satellite chlorophyll and radiance measurements and meteorological reanalysis data (e.g., windspeed from ECMWF's ERA-5 product) with empirical parameterizations. The estimated emissions are compared with a variety of observational records, and correlations with meteorological driving variables as well as climate modes of variability (e.g., El Niño - Southern Oscillation) are explored.

Overall, I think this is a useful and interesting dataset. These high-resolution emissions could be included in a global atmospheric chemistry model to explore the impacts of marine isoprene emissions on aerosol formation and tropospheric oxidation chemistry over the remote ocean. I think the atmospheric chemistry and climate research communities would both benefit from these data.

The article itself is reasonably clear, and the methodology is presented in a straightforward and comprehensive way. However, I have some issues with the lack of uncertainty analysis as well as the lack of justification / explanation for a few assumptions. I recommend publishing this manuscript once these concerns are addressed, because I think it would be very valuable to the global atmospheric science community.

1) I am concerned by the lack of uncertainty analysis presented in this paper. The calculation of both the biogenic ("BIO") and surface microlayer ("SML") isoprene emissions depends on satellite observations from MODIS and VIIRS, meteorological reanalysis data (ERA-5 in this case), and numerous empirical parameters derived from oceanographic or laboratory

measurements. Each of these quantities has some uncertainty associated with it, and these will propagate into your emission estimate. While I appreciate that putting precise error bars on global emission estimates is not trivial, some kind of error analysis or sensitivity experiment seems essential in order to make proper use of your data and methods. Even something as simple as calculating the emissions with a different reanalysis product or changing the values of some of the empirical parameters would give a strong indication of how sensitive the emission estimate is to errors in the model inputs and parameters. This would make the comparison with observations and previous emission estimates more meaningful, and it would make it easier to apply your methodology in different modelling frameworks (perhaps using different meteorological reanalysis data or satellite observations).

2) There are a few assumptions and methods that need more justification / explanation. These include the assumption that isoprene concentrations in seawater are constant, that isoprene is immediately oxidized in the marine boundary layer, and the use of an 8-year plankton type distribution over a twenty-year period. Please see my specific comments for more details.

3) The plots are generally clear and of high quality, but you should include labels for the different subplots and colour bars. I found myself frequently jumping back and forth between the main text, the figures, and the figure captions in order to make sense of everything. I have included a few specific comments about this below.

4) I was able to download and explore your dataset, and I could not find any problems with the files or data structure. However, I could not find any source code for your modules on the FTP server. Perhaps this is intentional; however, your introduction made it seem like your module could be easily embedded in an Earth System Model, so I was under the impression that I would be able to download the source code and play around with it. This is not necessarily a problem, but if you don't intend to release any source code you should consider rephrasing your introduction to avoid giving the impression that people can download your model.

5) The order of the paper is odd at times, and some sections are mislabelled in the introduction (see my specific comment below). I found it strange that the BIO emissions are presented before you explain how the plankton types were calculated. I was also surprised to see the comparison with observations (Section 3) presented before you discussed the spatial and temporal characteristics of your modelled emissions (Section 4). This made it a bit harder for me to follow your overall arguments. But I acknowledge this may just be my personal preference.

Response to **General comments:**

We greatly appreciate your acknowledgment of the importance of our current work, as well as your helpful general comments.

As a description paper of our newly presented dataset, this manuscript is asked and needed to state the basic information of the dataset for the potential users. The questions raised by the referee is of a great importance, which include the uncertainty of the dataset and justification and explanation for the parameterizations and assumptions used in our module. We updated these discussions according to the referee's comments carefully and precisely. We added related sensitivity experiments and explanation for the parameterizations and assumptions we used in our module.

We updated the section of  uncertainty discussion with a series of sensitivity experiments. We also updated some  justification and explanation for the parameterization and assumptions used in our marine isoprene estimation module as well as some tables and figures according to referee's comments. The detail revision can be found in the following response to specific comments.

Additionally, the code is available together with the dataset online.

Finally, we adjust the structure of the manuscript. We moved the discussion on the estimations from method section (2.2 and 2.4) to Section 3.2 with some update of previous studies comparison and corrected characteristic of the emissions. Besides, we merged the Section 4.4 (Data uncertainties) into Section 3.4 with extension by conducting a series of sensitivity experiments. In addition, the Sect. 3.4 is attached in the end of this Response Letter.

Response to **Specific comments:**

1. **Lines 40 - 42: There has been some work (e.g., Palmer et al 2022: https://www.science.org/doi/10.1126/science.abg4506) suggesting that terrestrial BVOC emissions have a large impact on downwind VOC and aerosol concentrations over the remote ocean, particularly in the South Atlantic due to the relatively long lifetime of BVOCs coming from the Amazon basin. Is it fair to say that terrestrial BVOC emissions do not exert significant influence over the remote ocean?**

   Response:

   Thank you for the suggestion. Here we have learnt from the work of Palmer (2022) accordingly. The typical lifetime of isoprene is ~1 hour, while its lifetime will be extended to ~3-5 hours when converted to isoprene peroxyl radicals and ~20-30 hours when converted to isoprene epoxydiols (IEPOX) (Palmer et al., 2022). Based on their lifetime, terrestrial isoprene is hardly long-distance transported to influence remote marine atmosphere, only if isoprene is oxidized to form low-volatile components (LVOCs) or secondary organic aerosols (SOA).The BVOCs in the Amazon probably has influence in the marine atmosphere due to its large emissions. We will couple this marine isoprene emission module with Earth System Model to evaluate the contribution of isoprene from Amazon compared to the marine emission in the future study.

   We removed the paragraph discussed BVOCs in Line 31 and replace with the introduction part of isoprene: *"Among all the non-methane BVOCs species, isoprene exhibits a lifetime of approximately 10-100 days in seawater (Booge et al., 2018). Once released into the atmosphere, it rapidly reacts with OH radicals, resulting in a short atmospheric lifetime of about one hour (Kameyama et al., 2014). Within the marine boundary layer (MBL), isoprene can undergo oxidation, leading to the formation of semi-volatile organic compounds (SVOCs) and low-volatility organic compounds (LVOCs) such as methacrolein and methacrylic acid. These compounds actively participate in the generation of marine secondary organic aerosols (SOAs) (Claeys et al., 2004; Kim et al.,*

*2017). Due to its significant emissions and capacity to contribute to SOA formation, marine isoprene plays a crucial role in aerosol generation and growth within the MBL. The estimation of marine isoprene emission is essential and serves as a fundamental aspect for future studies on marine SOAs and their climate effects (Carslaw et al., 2010).*

2. **Line 50: Here you use the terms "BIO" and "SML" without first defining them in the main text; they are explained in the abstract, but you did not explain them in the introduction.**

   Response:

   We added the explanations of these abbreviations in Line 48: *"Over the past decades, numerous studies have provided estimates of phytoplankton-generated biological emissions (BIO emissions) and photochemistry-generated emissions in the sea surface microlayer (SML emissions) over the global ocean."*

3. **Line 60: I am not sure what this sentence means. I interpret this to mean that biochemical losses of BVOCs are parameterized based on laboratory or field observations. But it is unclear what "dynamic euphotic zone" means in this context. Are you referring to Equation (5), where you calculate the depth of the plankton euphotic zone based on surface downwelling radiation?**

   Response:

   Thank you for the question. The "biochemical loss" (which is substituted by *"biological and chemical consumption"*) is based on the laboratory result from Shaw (2003) and Simo (2022). The "dynamic euphotic zone" means that the module diagnoses the maximum depth of the euphotic zone ($H_{max}$) every hour, which is determined by the diffuse attenuation coefficient at 490nm ($k_{490}$) and hourly surface solar downward radiation ($I_0$) (Eq. 5).

   We revised the description to: *"Several enhancements and refinements have been incorporated into the calculation of BIO emissions. These updates include the diagnosis*

*of the maximum depth of the euphotic zone each hour using the diffuse attenuation coefficient at 490nm ($k_{490}$) and hourly surface solar downward radiation ($I_0$)."* In Line 58.

4. **Line 70: Which two emission pathways are you referring to here? Are you talking about the BIO and SML sources, or are you talking about photochemical and windspeed-driven processes in the surface microlayer? I assume you mean BIO and SML based on the rest of the article, but you could avoid the ambiguity by clearly stating which pathways you're referring to.**

Response:

We revised the statement here to *"BIO and SML emissions"* in Line 68.

5. **Line 80-88: This is good background information, but it feels out of place in this paragraph. Consider moving this to the first or second paragraph instead. This information helps motivate why marine isoprene emissions are important, so I feel it should be introduced before you start describing emission estimate methodologies and uncertainties.**

Response:

We moved this paragraph to Line 31.

6. **Line 90: This is unclear to me. It would be easier to understand if you succinctly explained how you calculated the BIO and SML sources and how your approach addresses some of the uncertainties you outlined in the previous paragraph (data availability, unclear mechanisms, lack of satellite observations at high latitudes during winter, estimates of chlorophyll vertical distribution, and relations between isoprene and marine/meteorological factors). The way it is currently written, I have no way of knowing how your study plans to approach these issues until I have finished reading the paper.**

Response:

Thank you for the question. We revised this paragraph and add the information of the brief introduction of the methods and the other parts of the whole manuscript to address the uncertainty stated in the introduction section in Line 81: *"Two distinct types of emissions, BIO emissions and SML emissions, were calculated using satellite-derived monthly ocean chlorophyll concentration data from MODIS and ERA5 hourly meteorological reanalysis separately. The BIO emission is derived by the correlations between isoprene production and marine chlorophyll concentration, while the SML emission is determined by the surfactant in the sea micro-layer and windspeed. The availability and uncertainty of the dataset are discussed through the comparations with observed isoprene concentration and a series of sensitivity experiments. Our dataset can be used as input data for climate or atmospheric chemistry models. The module also can be coupled with the earth system model to calculate marine isoprene emissions online."*

7. **Line 81: The sentence "Two distinct types of emissions are separately calculated..." doesn't really make it clear what you're doing, or how you're using the MODIS chlorophyll and ECMWF reanalysis data. More broadly, I find that your introduction provides motivation for studying marine isoprene emissions and addresses some uncertainties in previous approaches, but it does not clearly explain how your new dataset was developed or how it addresses the uncertainties you mentioned. I understand that the introduction needs to be brief, and you describe these methods in detail later. But right now your introduction does not give enough information for me to tell what you actually did. After reading your introduction I should know what to expect from the rest of the paper, and right now that isn't the case.**

Response:

We revised the introduction and related sentences in Line 81: *"Two distinct types of emissions, BIO emissions and SML emissions, are calculated using satellite-derived monthly ocean chlorophyll concentration data from MODIS and ERA5 hourly*

*meteorological reanalysis separately. The BIO emission is derived by the correlations between isoprene production and marine chlorophyll concentration, while the SML emission is determined by the surfactant in the sea micro-layer and windspeed. The availability and uncertainty of the dataset are discussed through the comparations with observed isoprene concentration and a series of sensitivity tests. Our dataset can be used as input data for climate or atmospheric chemistry models. The module also can be coupled with earth system model to simulate marine isoprene emissions online".*

8. **Line 88: This paragraph seems to be incorrect. I think you have swapped the descriptions of Sect 4 and Sect 5. You said "Sect. 4 provides information on our dataset and data availability", but Sect. 4 in the text is simply titled "Results" and is focused on spatial-temporal variability of emissions and correlations with climate modes of variability. Similarly for Sect 5, you said it describes the "characteristics of marine isoprene emission", but in the text Sect 5 is just a data availability statement.**

Response:

This paragraph has been corrected with the sequence of brief introduction for each sections:

*"The subsequent section (Sect. 2) elucidates the methods and factors employed in our estimation of marine isoprene emissions. Our results are compared with previous isoprene emission inventories and some field observations in Sect. 3. The characteristics of the marine isoprene emission are analysed in the Sect. 4. Sect. 5 provides information on our dataset and data availability. Sect. 6 is the conclusions and discussions."*

9. **Line 94 : What is the spatial resolution of the downwelling radiative flux diffuse attenuation coefficient data? Is it also at 9km?**

Response:

The spatial resolution of the downwelling radiative flux diffuse attenuation coefficient data is 9km, we have revised the description to: *"Twenty years (2001-2020) monthly average*

*chlorophyll concentration data at 9 km resolution and 490 nm downwelling radiative flux*

*diffuse attenuation coefficient data with the same spatial resolution were obtained from…".*

**10. Line 99 : These meteorological variables (u-wind and v-wind, T2M, SST, and surface downwelling shortwave flux) are all from ERA-5, right?**

Response:

The meteorological variables use here are all from ERA-5. We revised the sentences with a clearer statement: *"The ERA-5 hourly average 10-meter u-wind and v-wind component, 2-meter temperature, sea surface temperature and surface downwelling shortwave flux were applied in the module."*

**11. Line 103: Is the monthly normalized water-leaving radiance at 410 nm also at 0.25x0.25 degree spatial resolution?**

Response:

The monthly normalized water-leaving radiance at 410 nm is at 4km spatial resolution. We have clarified the spatial information of the data: *"The resolution of original data is at 4 km, which is interpolated into a resolution of 0.25°×0.25°."*

**12. Line 140: Please clarify how you can apply this plankton distribution dataset over the entire twenty-year period. I understand that you use MODIS chlorophyll and NOAA water-leaving radiance at 410nm to obtain a plankton type distribution from 2012 - 2020. But it is unclear how you can use an 8-year plankton type distribution to estimate emissions over a twenty-year period.**

Response:

Thank you for the question. This phytoplankton type determined by the data of 2012-2020 was applied as the input type for all twenty years in our module with the assumption that the dominant phytoplankton type in each grid and month during twenty years are constant. The variation in the global spatial distribution of phytoplankton type is dominated by the

seasonal variation of radiation, temperature and the ocean trophic level, while the interannual variation in each month is of less importance. (Dandonneau et al., 2004, Uitz et al., 2010, Brewin et al., 2012). This issue is discussed together with other validation in Sect. 3.4: *"Our module used the dominant phytoplankton type for each month without hourly and daily variations due to the restriction of temporal resolution of measured chlorophyll-a and water leaving radiance data. We simply diagnosed the monthly phytoplankton types during period of 2012-2020. The phytoplankton types in 51 % of global grid cells are same in the all nine-year period, while the types in 89% of the grid cells are same for more than five years .As a result, we believe it is reliable to apply the monthly dominant phytoplankton type in each grid during 2012-2020 in the estimation during all twenty years (2001-2020)."*

13. **Lines 113 - 115: You assumed the concentration of isoprene in the ocean is static. Is this steady state assumption valid? What is the justification? Is it based on observations of marine isoprene concentrations, or is it based on theoretical considerations (e.g., ocean chemistry modelling)? And over what time period could we expect this assumption to be valid (Days? Weeks? Months?)? I appreciate that this assumption is very useful so that isoprene flux is equal to net isoprene production, but some more explanation / justification should be included.**

**& Line 292: I mentioned this in an earlier section, but please explain the assumption that marine isoprene concentrations are constant. It seems to me that you are assuming concentrations are constant to estimate the fluxes, then using those fluxes to estimate the concentrations. The logic seems circular. We know the concentrations are not constant because you show large variability in both observed and modelled concentrations in Figure 4. I don't doubt that you can use some sort of steady-state approximation in order to relate isoprene production to fluxes, but this needs to be clearly explained.**

Response:

Thank you for the question. The steady state assumption of marine isoprene concentration comes from the measurement of isoprene in an iron-fertilization experiment in the Northeast Pacific by Moore and Wang (2006), which suggests no isoprene accumulation in weeks. This result was further cited by Palmer (2005) and derived as the assumption of monthly steady isoprene concentration. Besides, the diurnal changes of isoprene concentration are found in field observations, which suggests the peak isoprene concentration shows in the mid of the day and decreases in the night (Milne et al., 1995, Sinha et al., 2007). We assumed the concentration is steady in each hour, and it changed discontinuously, which is dominated by hourly radiance. We revised the manuscript for the detail of the assumption in Line 108: *"The phytoplankton-generated emission module was developed based on the assumption that the concentration of isoprene in the ocean remains static in each hour. As a result, the net isoprene production (isoprene production minus biological and chemical consumptions) of global ocean approximately equal to the isoprene flux from the ocean to the MBL in each hour."*

14. **Line 110: How do we know isoprene will be oxidized immediately once it enters the marine boundary layer? While isoprene typically has a very short lifetime against OH oxidation, non-negligible isoprene and other BVOC mixing ratios have been measured in the marine boundary layer (e.g., Warneke et al., 2004), particularly around day-to-night transitions when OH concentrations are lower. If I understand correctly, you are assuming MBL isoprene concentrations are negligible so that you can neglect isoprene fluxes from air-to-sea, and instead focus exclusively on fluxes from sea-to-air. Can you provide some more context (i.e., why are you assuming it's negligible?) and justification (i.e., how do we know that isoprene is oxidized immediately in the MBL? What is its typical lifetime in the marine atmosphere?)?**

Response:

Thank you for the question. The typical lifetime of isoprene in the marine atmosphere is about an hour based on former laboratory and field studies (Bonsang et al., 1992, Shaw et

al., 2010, Booge et al., 2018), while the lifetime of isoprene in the marine boundary layer (MBL) is about 10 hours in the nighttime when the OH concentration is very low and influenced by the upwind terrestrial isoprene (Warneke, 2004). As a result, isoprene in the MBL possibly accumulates over one hour only under some specific conditions. Here we revised the manuscript in Line 107 for a clearer explanation in Line 112: *"Typically, the mixing ratio of isoprene in the MBL is very small (0~50 ppt, Yu et al., 2021) with lifetime of about an hour. In our sensitivity experiments, the isoprene mixing ratio is fixed in the MBL for 1 ppt and 20 ppt as the remote and coastal ocean conditions. The result turns out that the mixing ratio of isoprene is hardly able to affect the marine isoprene emission, especially in the remote ocean (mixing ratio = ~1 ppt). Hence it is reasonable to neglect the air-to-sea flux of isoprene against the sea-to air emission flux."*

15. **Line 120: Could you please clarify what you mean by "biochemical costs of isoprene is seawater"? Does this refer to the consumption of isoprene by biological processes, or are you talking about something else?**

Response:

Thank you for the question. We revised the "biochemical costs" in the manuscript to *"biological and chemical consumption"*, which is the consumption process of isoprene. *"Biological consumption is marine isoprene loss due to the degradation by isoprene-degrading bacteria and other microbials. Chemical consumption is caused by the photochemical processes in the surface ocean, which is calculated from reaction rate constant."* The explanation of biological and chemical consumption is added in Line 120.

16. **Line 122: What kinds of observations did Simo et al 2022 use to calculate alpha? Was this relationship observed in different regions of the ocean, or did they use a small set of observations? In other words, do we think this relationship is robust enough to be applied to the global ocean?**

Response:

Thank you for the question. First, we updated the factor α to $\alpha = 0.1 \times C_{chl} + 0.05$ since the Eq. 2 in the manuscript is an incorrect citation. Instead, we actually used the equation 1 in Simó et al., (2022) to calculate biological and chemical consumption term in our BIO emission module. This relationship is derived from a series of observations, including eleven coastal and open sea sites in tropic Pacific, Mediterranean, Atlantic and circum-Antarctic, which covered a wide ocean area (Simo et al., 2022). Therefore, we think the relationship used in our module is sufficiently robust when applied to the global ocean. We revised the manuscript in Line 126: *"This relationship is derived from a series of observations, including eleven coastal and open sea sites in tropic Pacific, Mediterranean, Atlantic and circum-Antarctic (Simo et al., 2022)."*

**17. Lines 133 - 135: Can you clearly state that radiation is given by I and the plankton type coefficient is given by Tc in this sentence? Otherwise Equation (3) is unclear until after the next paragraph.**

Response:

Thank you for the question. The isoprene production rate $P$ is calculated using Eq. (3):

$$P = I \cdot C_{chl} \cdot T_c \ . \qquad\qquad (3)$$

We revised the description in Line 136: *"The isoprene production rate P, is determined by a linear relationship between chlorophyll concentration, radiation, and the diffuse attenuation coefficient at 490 nm, as well as the classification of phytoplankton types. The radiation is used to determine the term I, which is calculated as the total radiance in the euphotic layer. $T_c$ represents the ability of isoprene production for different phytoplankton types. Four distinct types of phytoplankton (i.e. haptophytes, Prochlorococcus, Synechococcus-like cyanobacteria and diatoms) are involved, each with a different isoprene production rate defined below. These coefficients were determined in previous studies, which will be discussed in the next section. $C_{chl}$ (mg·m⁻³) represents the sea surface chlorophyll concentration, which is considered as a parameter within the mixed layer of each grid cell."*

**18. Lines 156 - 158: I understand that 0.433 Tg yr-1 is only accounting for the BIO source, but you include your total estimate (BIO + SML) in Table 1. Are the other emissions estimates in Table 1 total emissions? This is what I assumed, but the Brüggemenn et al study is listed as "Sea Surface Microlayer" so now I am not sure. Please specify in the table whether these are TOTAL, BIO, or SML emission estimates so that it is easier to compare the different studies.**

**Also, what is the uncertainty on your BIO estimate? Do you have an idea of how this estimate might change based on errors in the input data (e.g., MODIS chlorophyll or ERA-5 meteorological data) or model parameters?**

Response:

Thank you for the question. We revised the Table 2 according to the comment:

| Compounds | Emissions
Tg·yr$^{-1}$ | | Reference |
|---|---|---|---|
| | 0.11 | (BIO emissions) | (Palmer and Shaw, 2005) |
| | 1.36 | (BIO emissions) | (Sinha et al., 2007) |
| | 0.79 | (BIO emissions) | (Gantt et al., 2009) |
| | 0.31 | (BIO emissions) | (Arnold et al., 2009) |
| | 1.90 | (Top-down) | (Arnold et al., 2009) |
| | 0.99 | (BIO emissions) | (Myriokefalitakis et al., 2010) |
| | 0.36 | (BIO emissions) | (Luo and Yu, 2010) |
| Isoprene | 13.15 | (Top-down) | (Luo and Yu, 2010) |
| | 0.24 | (BIO emissions) | (Booge et al., 2016) |
| | 0.65 | (BIO emissions) | (Kim et al., 2017) |
| | 1.11 | (SML emissions) | (Brüggemann et al., 2018) |
| | 0.75 | (Total emissions) | (Conte et al., 2020) |
| | 0.96 | (BIO emissions) | (Li et al., 2020) |
| | 1.10 | (Total emissions) | This study |

In the updated table, we added the emissions types considered by previous studies.

Besides, the uncertainties of the BIO emission in our dataset are discussed in the Sect. 3.4. We add the sensitivity experiments for different input factors and parameters in our emission module including wind, chlorophyll concentration, radiation phytoplankton type, and parameterizations factors. In addition, the Sect. 3.4 is attached in the end of this Response Letter.

Another meteorological reanalysis of National Centers for Environmental Prediction (NCEP) FNL is used for comparation. We added content in Line 446: *"Another input meteorological dataset is used in our module to valid the robustness of our module. the data from National Center for Environmental Prediction (NCEP) Global Data Assimilation System (GDAS)/FNL (final) 0.25 Degree Global Tropospheric Analyses and Forecast Grids were applied for this sensitivity experiment. We derived the radiation on the ground and water surface level and wind speed at 10m in 2020 from NCEP reanalysis instead of ERA5 reanalysis as input data for isoprene emission calculations. The total emission using NCEP reanalysis is of 7.8% larger then former result from ERA5 reanalysis with the BIO emission and SML emission larger by 6.7% and 9.0% separately."*

19. **Line 161: You say these factors (temp, salinity, etc.) will lead to different isoprene production rates, but you only accounted for radiation and photic zone depth (Hmax) in equations 1 - 5. How did your account for the impact of temperature, salinity, and nutrients? Are these impacts small / negligible, or are they implicitly accounted for by the chlorophyll concentration term in Equation 2 and Equation 3?**

Response:

Thank you for the question. The sea surface temperature (SST) data is used as input data to calculate isoprene air-to-sea flux in the scenario where the mixing ratio of isoprene in marine boundary layer (MBL) is considered. This discussion was added in Sect. 3.4 as the explanation and justification of the assumption treating the isoprene mixing ratio in the MBL equals to zero. In our module, we assumed that the influences from the factors

including SST and salinity results in the variation of chlorophyll concentrations. We revised related paragraph to: *"Along with various oceanological conditions of different oceans on the global scale, various dominant phytoplankton types would produce isoprene in different rates through their photosynthesis and metabolic process (Booge et al., 2018; Dani and Loreto, 2017)."* Additionally, radiation and chlorophyll are primary input in our module, while the windspeed is only used in the SML emissions calculation. The normalized water leaving radiation is used to determine the phytoplankton types and diffuse attenuation at 490nm is used to calculated the maximum depth of the euphoric layer. Besides, we added a series of sensitivity experiments in the Sect. 3.4 to discuss the impact of temperature, windspeed and chlorophyll concentrations on the estimation of isoprene emission in Line 458.

**20. Line 175: Why was the value of 0.028 chosen? Do we expect haptophytes to dominate in oligotrophic regions of the ocean, or is there another reason you chose this value?**

Response:

Thank you for the question. The haptophyte is a wide-spread marine producer, which dominates Chla-normalized phytoplankton standing stock in modern oceans (Liu et al., 2009). The haptophyte contributes to more than 70% of the global marine dominant phytoplankton types in spring and summer months and over 50% in winter months (Alvain et al., 2005). Based on the dominance of haptophytes in the global ocean all year-long, especially these species with small cell dominate the oligotrophic ocean, we decided to use the coefficient of 0.028 for haptophyte in the oligotrophic waters where Chla concentration lower 0.04 mg·m$^{-3}$ and area with missing value as suggested in Alvain (2005). Related revise is added in Line 172: *"A other type with a coefficient of 0.028 was assigned in areas with chlorophyll concentrations below 0.04 mg·m$^{-3}$, including oligotrophic regions and grids with missing values. This other type is with a same coefficient for type haptophytes, which is a wide-spread phytoplankton with over 50% contribution in the global ocean all year round (Alvain et al., 2005)."*

**21. Lines 180-186: Is this for coastal regions everywhere, or did these studies focus on specific regions?**

Response:

Thank you for your question. The phytoplankton type used in the coastal region is based on previous research in the East China Sea (Guo et al., 2014; Li et al., 2018; Liu et al., 2016). We added this information in the manuscript: *"Based on previous observational studies in the East China Sea, which is a typical coastal region, it was determined that the dominant phytoplankton type is a combination of 50 % diatoms and 50 % haptophytes in the grids with chlorophyll concentrations greater than 3 mg·m⁻³ (Guo et al., 2014; Li et al., 2018; Liu et al., 2016)."*

**22. Lines 195 - 197: What about the large areas of undefined types in tropical and subtropical regions? In particular, Figure 1 a), b), and d) show large "undefined" areas in the southern Subtropical Pacific and western tropical pacific. Another hotspot seems to be the Arabian Sea and Bay of Bengal in Figure 1 c). Is the use of an undefined plankton type a large source of uncertainty in these regions?**

Response:

Thank you for the question. As you found the "other type" mainly occurs in the subtropical ocean and Arabian Sea and Bay of Bengal in boreal summer. However, they were determined by different ways. "Other type" in the subtropical ocean are generally due to the low nutrient level there resulting in the chlorophyll concentrations lower than 0.04 mg·m⁻³. Our module cannot determine the specific phytoplankton type in the area with low chlorophyll concentrations, but the emissions in these areas were still included in our estimation with emission factor of 0.028. Although the empirical factor used may lead to some uncertainty in the emission estimation, the emission strength is small in the area with low chlorophyll concentration which would not lead to large uncertainty. We added following sentences to illustrate this problem in Line 198: *"It is also found that the other*

*type appears in the subtropic ocean, which is generally due to the low nutrient level there resulting in the chlorophyll concentrations lower than 0.04 mg·m⁻³. For the oligotrophic ocean, our module cannot determine the specific phytoplankton type, but the emissions in these areas were still included in our estimation with emission factor of 0.028 according to the dominance of the type haptophytes in the global ocean (Alvain et al., 2005)."* Large extent of "other type" in the regions of Arabian Sea and Bay of Bengal is mainly exhibited in the boreal summer months, which is caused by the missing value of satellite-based input data, including chlorophyll concentration and water-leaving radiance. We applied the interpolation method for each grid cells in these regions for the boreal summer emissions. We revised the manuscript in Line 202: *"The phytoplankton type in another noticeable ocean area of Arabian Sea and Bay of Bengal is decided to the other type due to the missing satellite data there in the summer months. We applied the interpolation method for each grid cells in these regions for the boreal summer emissions. The details of the interpolation method and the improvement are discussed in Sect. 2.5."*

23. **Line 230: I understand that you are using chlorophyll observations as a proxy for nutrient levels and surfactant concentrations, but I do not understand where the expressions for Csurf and Cmax come from. Can you explain this? Perhaps a brief explanation of the methodology of Wurl et al 2011, or at least state the rationale behind Equation (8) and the expressions for Csurf and Cmax. Right now this method seems very opaque without having read Wurl et al 2011.**

Response:

Thank you for the question. We revised Line 231 in the manuscript to: *"In the Eq. (8), $F_{surf}$ accounts for a logarithmic decay of isoprene SML emissions with the decreasing surfactant concentration (Brüggemann et al., 2018). The two surfactant concentration terms, $c_{surf}$ and $c_{max}$, are determined with a simplified method based on previous research, using the field-observation-based surfactant concentration equivalents of Triton X as the surfactant concentration in SML (Wurl et al., 2011). Here the nutrient level of the*

*ocean is determined by the concentration of chlorophyll $C_{chl}$ (mg·m$^{-3}$). The surfactant concentration reaches its maximum at $c_{max}$ = 663 μg·Teq·L$^{-1}$ , which is the mean concentration in the eutrophic waters ($C_{chl} \geq 0.4$ mg·m$^{-3}$) in Wurl (2011)'s experiment. A linear relationship was established to determine the surfactant concentration in the oligotrophic ocean with $C_{chl}$ < 0.4 mg·m$^{-3}$, which is $c_{surf}$ = 857·$C_{chl}$+320 μg·Teq·L$^{-1}$. The $c_{surf}$ approaches to 320 μg·Teq·L$^{-1}$ when chlorophyll concentration in a low level."*

24. **Lines 245: Is this due to the destruction of the surface micro-layer at high windspeeds? Why is your wind speed threshold (13ms-1) different from the one mentioned in the introduction (10ms-1)?**

   **What is the mean SML emission rate? At the end of Section 2.2 you gave a mean BIO estimate, so it would be nice to see the same for SML here. And just like with Section 2.2, you should address the uncertainties in this estimate (or at least explain if you will address them in a later section). Your estimate of SML emissions relies on several empirical parameters (e.g., Flab) and meteorological input variables. All of these quantities have uncertainties, which will propagate into your emission estimate. Some sort of error analysis or sensitivity experiment would be extremely valuable.**

   Response:

   Thank you for the question. We calculated the SML emission only when the wind speed is smaller than the threshold because the surface micro-layer will destruct at high windspeeds over 13 m·s$^{-1}$ (Bruggemann et al., 2018). The "10-meter wind speed" mentioned in the introduction is windspeed at the height of 10 meter, which is not the threshold of 10 m s$^{-1}$.

   Besides, we added the sentence for the mean SML emission rate in Line 246: *"The average annual SML emission was calculated to be 0.616 Tg·yr$^{-1}$ for the period 2001-2020, which is about 30% larger than the BIO emission."*

   In addition, The discussion of uncertainty was added in Sect. 3.4.

**25. Line 265: How do these changes compare to the uncertainties on BIO and SML emissions? Is your interpolation a major source of uncertainty in your emission estimate, or are these changes significantly smaller than the other sources of error?**

Response:

Thank you for the question. The interpolation accounts for 2.4% of the annual total isoprene emissions. Compared to the result of the sensitivity experiments, the change caused by interpolation method is smaller than the uncertainty from most of other factors in their range of values. Additionally, The change of interpolated result is close to the range of the parameter which determines surfactant concentration. This range of the sensitivity experiment accounts for -2.1%-3.0% of the annual total isoprene emissions. We added the sentences in Line 266: *Compared to the result of the sensitivity experiments, The change caused by interpolation method is smaller than the uncertainty from most of other factors in their range of values.*

**26. Lines 289: How do these differences compare to the uncertainty of your estimate? For the comparison with observations to be meaningful, we need to know what kinds of errors are present in your emission estimate.**

Response:

Thank you for the question. The mean deviation between observed and simulated isoprene emission flux is 34% (Figure 8), which is smaller than most of our sensitivity results for input data and another reanalysis from NCAR (Table 5). We examined the uncertainty through a series of sensitivity experiments. The detailed uncertainty analysis was added in Section 3.4, which was also attached in the end of this response letter.

**Lines 295 - 301: Is sea surface temperature also coming from ERA-5? Would you get very different results for Equations 10 - 12 if you used a different reanalysis product?**

Response:

Thank you for the question. The sea surface temperature (SST) used in our module is from ERA-5 reanalysis. Here we present the result by using another SST reanalysis product from National Oceanic and Atmospheric Administration (NOAA) OI SST V2 High Resolution Dataset (https://psl.noaa.gov/data/gridded/data.noaa.oisst.v2.highres.html), which is monthly SST data with the same spatial resolution of 0.25°×0.25° as ERA-5 reanalysis.

We compared these two SST datasets, according to the SST term in the Eq. 10-12, which has a simple mathematical relation to the isoprene concentrations. The annual mean difference between the two SST dataset is about 0.03%, which is neglectable compare to other possible uncertainties.

27. **Lines 309 - 313: Similar to my previous comments on BIO and SML estimates, can you do a sensitivity experiment to at least get some idea about the uncertainties? You mention that Equation (11) may introduce uncertainties which could partly explain model-observation discrepancies, but you don't quantify how big these biases might be. Even if getting a precise error estimate is difficult, it should at least be easy to figure out the impact of errors in sea surface temperature and 10-metre wind speed.**

**You also mention that Eq. 11 is only valid in the range of w = 4 - 15ms-1, so ideally you should eliminate this source of error by excluding locations and times where w falls outside of this range. Do you filter for wind speed in your simulation, or are you using all data points even if they fall outside the range of 4-15ms-1? If you are using all data points even though Eq. 11 is not valid, how big of an error might this introduce?**

Response:

Thank you for the question. The largest uncertainty from Eq. 11 is comes from the exchange velocity term $k_{ex}$. In the same study of Wanninkhof (2014), An estimated uncertainty of 20% was determined for the exchange velocity $k_{ex}$ by Wanninkhof (2014). This 20% uncertainty in velocity contribute to the calculation of isoprene concentration in

the seawater, which possibly explain part of the deviation between estimation and observations. We added sentences in the manuscript (Line 329): *"The uncertainty in the method to simulate the sea-to-air exchange process will further affect the results of marine isoprene concentrations. The uncertainty is caused by about 20% (Wanninkhof, 2014)."* The valid windspeed range of 4-15 m·s⁻¹ for Eq. 11 is used in the further calculation of the marine isoprene concentrations. We had filtered the windspeed in our simulation and calculated the isoprene concentrations with the data in the valid range.

**28. Lines 346: These ranges are small, so I would expect that the uncertainties on these estimates are probably much larger than the reported ranges. So it would be very useful to include an error estimate here.**

Response:

Thank you for the question and advise. Here the range of *"1.075 to 1.112 Tg·yr⁻¹"* is the emission range of twenty year 2001-2020, which only shows the range of annual emission change. First we added the standardized deviation of the twenty-year annual emission in Line 348: *"The standard deviation of the twenty-year period annual marine isoprene total emission is 0.0095 Tg, which is about 0.8% of the annual total emissions."*

Besides, We quoted the uncertainty range in Line 345: *"The range of the annual global BIO emission is 0.443 to 0.664 Tg·yr⁻¹,while SML emission is 0.583 to 0.655 Tg·yr⁻¹."* The detail of the uncertainty discussion is in Sect. 3.4. In addition, the Sect. 3.4 is attached in the end of this Response Letter.

**29. Lines 345 - 360: What is the benefit of using the new parameterization in Eq. 2, and is this significantly better than the previous approach you described in the introduction where a linear relationship between chlorophyll concentration and isoprene emission was used?**

**In general, I agree that there are benefits to using a higher spatial and temporal resolution, but I am not completely convinced that the updates used to calculate BIO reduce uncertainties. Your estimate depends on various satellite**

**and ERA-5 input variables as well as laboratory-derived empirical parameters. All these quantities have their own uncertainties which will affect your BIO estimate. I think it is essential that you try to quantify this uncertainty.**

Response:

Thank you for the question. The new parameterization in Eq. 2 is used for a latest biological and chemical consumption term, which is derived from field observations including eleven sample sites in Pacific, Atlantic, Mediterranean and Southern Ocean. We think this new parameterization is relatively representative compared with other laboratory and field results (Palmer et al., 2005, Conte et al., 2019, Simo et al., 2022). To assess the update of the new parameterization, we presented comparisons of calculated results with observed data in Sect. 3.1. Besides, the uncertainties are discussed in Sect. 3.4, with a series of sensitivity experiments to stress the different sensitivity of each input factors and parameters, as well as the uncertainty range caused by these assumption and parameters. In addition, the Sect. 3.4 is attached in the end of this Response Letter. We tried to determine the uncertainty for BIO emission and SML emission separately, by the means of different assumptions and parameters in Line 419: *"The uncertainty of BIO emission is mainly caused by the phytoplankton types. We calculated the uncertainty in BIO emission using these two specific types. The parameter 0.042 in Table 6 was same as the parameter for diatom, while the minimum parameter is used in that used to calculate the uncertainty. The uncertainty of SML emission is related to marine productivity. The surfactant concentration is determined by chlorophyll-a concentration and is split into three bins. The uncertainty in the phytoplankton types was investigated using the maximum and the minimum concentration."*

**30. Line 364: Is this the main benefit of your SML approach compared to previous methods?**

Response:

Thank you for the question. We think the way the input parameters are used here is the simplification for our module. This is not a major benefit of our method.

**31. Line 365-367: I don't think this needs to be re-stated here.**

Response:

Thank you for the question. We removed the related sentences.

**32. Line 445: General comment that applies to all multi-panel plots, especially the world maps: Please include labels for the different subplots and for the colour bars. The plots themselves are clear, but it is tedious to keep going back and forth between the figure caption and the plot to make sense of what I am looking at.**

Response:

Thank you for the question. We have adjusted figures in the manuscript by adding labels and captions according to the figure contents.

**33. Lines 375 - 380: Do these statistics account for the difference in ocean surface area between the two hemispheres? (i.e., does an "average" Southern Hemisphere grid cell emit more isoprene than an "average" Northern Hemisphere grid cell, or can the difference partly be explained by the fact that there is less ocean in the Northern Hemisphere?). It's not easy to tell using the maps in Figure 6.**

Response:

Thank you for the question. We revised the paragraph and added the average grid cell emission in the both hemisphere in Line 515: *"In the twenty-year period, the average annual emissions in the Northern Hemisphere amounted to approximately 44.9 %, whereas the Southern Hemisphere accounted for 55.1 % of the total emissions. However, the emission per unit area in NH (3.3 mg·m$^{-2}$·yr$^{-1}$) is 6.5% larger than that in SH (3.1 mg·m$^{-2}$·yr$^{-1}$) due to the larger and better nutritional status of coastal ocean areas in NH. The difference in the total emissions between two hemispheres is largest in boreal winter (Fig.*

*5). The emission in the boreal winter of the Southern Hemisphere contributed 17.7 % of annual global emissions in average, while the emission in the same season of the Northern Hemisphere accounted for only 8.7 %." Meanwhile, the emission per unit area in NH (0.70 mg·m⁻²) was still smaller than that in SH (0.85 mg·m⁻²) in the boreal winter. Radiation and duration of day dominate the seasonal variations of total emissions directly and indirectly through their influences on the distribution of chlorophyll concentration."*

34. **Line 440: This vocabulary is a bit unclear to me. Are you saying that BVOC emissions in the tropical ocean are primarily governed by local / small scale atmosphere-ocean interactions (e.g., small scale weather systems)? Or am I misinterpreting something? What do you mean by "local air-sea system"?**

    **Line 441: This is why I am confused about "air-sea system". In the previous sentence you say emissions are determined by local air-sea system (local weather conditions?), and here you say large scale variability is important. Please clarify what you mean by air-sea system and what you mean by "local" versus "large-scale" air-sea system. Are we talking about weather systems or large-scale climate variability?**

    **Line 449: Again, it is unclear what you mean by air-sea system. In this section it seems that you are describing modes of climate variability like ENSO, but in the previous section it sounded like you were describing local weather systems.**

    **Line 449: Please clarify what "identify the target area" means. Is it the region of the ocean that is affected by a particular mode of climate variability (e.g., ENSO)?**

    Response:

    Thank you for the questions. We changed the phase "local air-sea system" to *"local atmosphere and ocean conditions"* in the manuscript, which means a climate (or weather) pattern with a specific spatial scale and a concrete air-sea interactions. The "large-scale air-sea system" here leads the further discussion about El niño-Southern Oscillation

(ENSO) and Indian Ocean Dipole (IOD), which may affect the isoprene emission through the correspond meteorological factors. According to the analysis of the correlation between emissions and meteorological factors, these air-sea systems have the potential to affect the marine isoprene emission potentially. We will conduct a further study to discuss these mechanisms and processes in future. In the later paragraph, the "identify the target area" means to find a specific area which is affected by a concrete climate or weather pattern. We revised the statement in Line 625: *"In order to locate and investigate the potential impact of air-sea systems on isoprene emissions, the multiple variables empirical orthogonal function (MVEOF) was employed to examine the spatial pattern of temporal variation in BIO and SML isoprene emissions (Fig. 9)."*

35. **Line 482: I appreciate that this is an exploratory analysis. It would be interesting if you could briefly speculate on how to verify these relationships. Are there other analyses you could do with your dataset? If not, do you know what other kinds of data / observations would help determine whether these relationships are robust? The correlations you have shown between your emission dataset and different modes of climate variability are interesting, and I think it would be beneficial if you could provide some ideas for how to use this information in follow-up studies.**

Response:

Thank you for the question. Several methods are used to investigate potential characteristics, including MVEOF, wavelet and correlation analysis in this manuscript. As a dataset description paper, the manuscript focused on the task to state the basic status, method and conclusions of the dataset comprehensively. Thanks for your interest in our dataset and its further applications. The background air-sea system plays a leading role in the marine isoprene emission through the factors and their variations. For which the mechanisms of the impact from these factors are needed further investigations. We are pleased to use and integrate new analysis method into our future follow-up works. As the paragraph discusses the potential relationship between MJO and isoprene emission, it is

derived from the period information, which is conducted by the wavelet method. Similar to the MJO, other periodic weather or climate pattern may also have potential influence, like ENSO, IOD in this manuscript or other patterns like North Pacific Gyre Oscillation and Atlantic Niño. These pattern all have interactions between atmosphere and ocean. We added the following sentences in Line 617: *"The air-sea system plays a leading role in the marine isoprene emission. The air-sea system such as MJO, ENSO and IOD may have potential influence on the marine isoprene emissions."*

36. **Line:484 and figure 10: I have a couple problems with this figure. You didn't include a legend, so it is unclear what the different colours contours represent, what the solid black lines represent, or what the dashed red lines represent.**

**Also, the subplots need to be labelled so that it is obvious which emissions (BIO or SML) and which region (Mid-latitude Pacific or Tropical Indian Ocean & Pacific) we are looking at. I found myself frequently jumping back and forth between the figure, the body text, and the figure caption to make sense of the results.**

**The patterns you described in the text are reasonably clear (e.g., the 0.25 year signal in panel C) when you know what to look for, but it would be much easier to interpret the figure if you included a legend and labels for the subplots.**

Response:

[Figure]

Thank you for the question. We revised the figure 10 and updated the caption for detail information: *"The black irregular closed contours in the left column represent periods which the significance level is greater than 95%. The symmetrical black solid curve in the left column is cone of influence. Period signals above this curve is available. Red dash lines in the right column represents the 95% significance level. The peaks of the black curves in the right row over the red dash lines is of 95% significance in the twenty-year period average."*

37. **Lines 487 - 505: This section needs to be expanded. The discussion here is good, but you also need to address uncertainties due to the parameterizations as well as the ERA-5 and satellite input data. Some effort needs to be made to quantify these uncertainties for other researchers to make use of these data. My concern is that if other researchers try to apply your method and get wildly different results, they won't know whether it's due to an error in their methodology or if it's an expected error due to uncertainties in the model parameters and inputs.**

Response:

Thank you for the question. We moved the Sect 4.4 to Sect 3.4 and expanded with a series of sensitivity experiments and uncertainty discussion. We conducted relative discussion for the usage of different input data in Line 450: *"Another input meteorological dataset is used in our module to valid the robustness of our module. the data from National Center for Environmental Prediction (NCEP) Global Data Assimilation System (GDAS)/FNL (final) 0.25 Degree Global Tropospheric Analyses and Forecast Grids were applied for this sensitivity experiment. We derived the radiation on the ground and water surface level and wind speed at 10m in 2020 from NCEP reanalysis instead of ERA5 reanalysis as input data for isoprene emission calculations. The total emission using NCEP reanalysis is of 7.8% larger then former result from ERA5 reanalysis with the BIO emission and SML emission larger by 6.7% and 9.0% separately.* In addition, the Sect. 3.4 is attached in the end of this Response Letter.

38. **Line 506: I see the hourly dataset at 0.25x0.25 degrees is available online. I was able to connect to the FTP server and download the files. I only downloaded a small subset due to the large size of the dataset (2.65 TB), but the files I looked at were formatter properly and I was able to make some plots with them using Python's netCDF libraries. However, I don't see any way to access your emission module. You state on lines 93-94 that the module can be used to calculate emissions online in an Earth System Model. Are you planning to release the code for your module, or is it expected that other researchers wishing to use your method would implement it themselves based on the equations you provided? Please clarify, because the introduction made it seem like it would be possible to download the source code.**

Response:

Thank you for the advice. We have upload our module code together with our dataset.

39. **Line 522: You are only talking about Winter, right? In Section 4.1 you said**

**NH emissions were 44% and SH emissions were 56%. Here it sounds like you are claiming SH emissions are twice as large as NH emissions.**

Response:

Thank you for the advice. Here we talk about the emission of two hemisphere for the boreal winter. The emission in the Southern Hemisphere (SH) are about twice as large as the emission in the Northern Hemisphere (NH) emissions. As you mentioned, the comparison in Section 4.1 of "The emission in the NH contributes 44%, while the emission in the SH contribute 56%" (which the percentage is now revised as 55% and 45%) is for the annual total emission.

40. **Lines 525 - 527: Can you explicitly connect the observed trends in Section 4.1 with the correlations observed in Section 4.2 and air-sea systems observed in Section 4.3? It would be useful to provide some more context for the reported trends.**

Response:

Thank you for the advice. In the Section 4.3 we mainly discussed the air-sea system in the tropical ocean, including ENSO, IOD and MJO. These systems directly influence the meteorological factors like windspeed, sea surface temperature and radiance. These three factors perform differently on their correlation with isoprene emission in the tropical regions. The windspeed and radiance both shows a positive correlation with emission while the sea surface temperature shows negative correlations with emissions. According to the Section 4.1, the low-latitude Ocean (30N-30S) is with a decreasing trend for the annual BIO and SML emissions in the period of 2001-2020. We added the following sentences in Line 567: *"This trend is controlled by the tropical air-sea system potentially. Our former investigation suggest that the ENSO influence the tropical Pacific isoprene emission significantly when the ENSO is at its strong positive or negative phase. In the strong positive phase, the tropical west wind is strengthened, which leads to the warm water accumulate in the tropical Pacific. This process makes the increase of sea surface*

*temperature in the tropic Pacific, which further weakens the isoprene emission in this area."*

This is a possible mechanism for the air-sea system connected with isoprene emission.

**41. Lines 531 - 533: It's not entirely clear how these relationships will improve the accuracy of isoprene emission estimates. The connections are certainly interesting, but it is not clear what you can do with this information to improve emissions or reconcile discrepancies between observations and models.**

Response:

Thank you for the question. We revised the related sentences for a more specific statement in Line 693: *"These quasi-periodic patterns and their relationships with emissions provide valuable insights for refining existing methods. They also help bridge the gap between observations and model calculations."* The quasi-periodic pattern in the tropical and subtropical ocean is a signal and instruction for further determining the relationship between isoprene emission and periodical changes of meteorological and marine variables in tropical and subtropical ocean. These variables are often potentially dominated by air-sea system patterns, like ENSO, MJO, etc. Ideally, once the relationship is firmly decided, we can use the index of these patterns to get the general status of the marine isoprene emissions. This makes using simple index to reflect the marine BVOC emission possible. Further works are needed to improve the relationships and reduce the discrepancies. We discussed the possible future work content in Line 606: *"
[revised manuscript text omitted]

Bonsang, B., Gros, V., Peeken, I., Yassaa, N., Bluhm, K., Zoellner, E., Sarda-Esteve, R., & Williams, J. (2010). Isoprene emission from phytoplankton monocultures: the relationship with chlorophyll-a, cell volume and carbon content. Environmental Chemistry, 7(6), 554-563. https://doi.org/10.1071/En09156

Bonsang, B., Polle, C., & Lambert, G. (1992). Evidence for marine production of isoprene. Geophysical Research Letters, 19(11), 1129-1132. https://doi.org/https://doi.org/10.1029/92GL00083

Booge, D., Marandino, C. A., Schlundt, C., Palmer, P. I., Schlundt, M., Atlas, E. L., Bracher, A., Saltzman, E. S., & Wallace, D. W. R. (2016). Can simple models predict large-scale surface ocean isoprene concentrations? Atmospheric Chemistry and Physics, 16(18), 11807-11821. https://doi.org/10.5194/acp-16-11807-2016

Booge, D., Schlundt, C., Bracher, A., Endres, S., Zancker, B., & Marandino, C. A. (2018). Marine isoprene production and consumption in the mixed layer of the surface ocean - a field study over two oceanic regions. Biogeosciences, 15(2), 649-667. https://doi.org/10.5194/bg-15-649-2018

Brewin, R. J. W., Hirata, T., Hardman-Mountford, N. J., Lavender, S. J., Sathyendranath, S., & Barlow, R. (2012). The influence of the Indian Ocean Dipole on interannual variations in phytoplankton size structure as revealed by Earth Observation. Deep-Sea Research Part Ii-Topical Studies in Oceanography, 77-80, 117-127. https://doi.org/10.1016/j.dsr2.2012.04.009

Bruggemann, M., Hayeck, N., Bonnineau, C., Pesce, S., Alpert, P. A., Perrier, S., Zuth, C., Hoffmann, T., Chen, J. M., & George, C. (2017). Interfacial photochemistry of biogenic surfactants: a major source of abiotic volatile organic compounds. Faraday Discussions, 200, 59-74. https://doi.org/10.1039/c7fd00022g

Bruggemann, M., Hayeck, N., & George, C. (2018). Interfacial photochemistry at the ocean surface is a global source of organic vapors and aerosols. Nature Communications, 9. https://doi.org/ARTN 2101

10.1038/s41467-018-04528-7

Carslaw, K. S., Boucher, O., Spracklen, D. V., Mann, G. W., Rae, J. G. L., Woodward, S., & Kulmala, M. (2010). A review of natural aerosol interactions and feedbacks within the Earth system. Atmospheric Chemistry and Physics, 10(4), 1701-1737. <Go to ISI>://WOS:000274851500015

Ciuraru, R., Fine, L., van Pinxteren, M., D'Anna, B., Herrmann, H., & George, C. (2015). Unravelling New Processes at Interfaces: Photochemical Isoprene Production at the Sea Surface. Environmental Science & Technology, 49(22), 13199-13205. https://doi.org/10.1021/acs.est.5b02388

Ciuraru, R., Fine, L., van Pinxteren, M., D'Anna, B., Herrmann, H., & George, C. (2015). Photosensitized production of functionalized and unsaturated organic compounds at the air-sea interface. Scientific Reports, 5. https://doi.org/ARTN 12741

10.1038/srep12741

Claeys, M., Wang, W., Ion, A. C., Kourtchev, I., Gelencser, A., & Maenhaut, W. (2004). Formation of secondary organic aerosols from isoprene and its gas-phase oxidation products through reaction with hydrogen peroxide. Atmospheric Environment, 38(25), 4093-4098. <Go to ISI>://WOS:000222706000001

Conte, L., Szopa, S., Aumont, O., Gros, V., & Bopp, L. (2020). Sources and Sinks of Isoprene in the Global Open Ocean: Simulated Patterns and Emissions to the Atmosphere. Journal of Geophysical Research-Oceans, 125(9). https://doi.org/ARTN e2019JC015946 10.1029/2019JC015946

Dandonneau, Y., Deschamps, P. Y., Nicolas, J. M., Loisel, H., Blanchot, J., Montel, Y., Thieuleux, F., & Becu, G. (2004). Seasonal and interannual variability of ocean color and composition of phytoplankton communities in the North Atlantic, equatorial Pacific and South Pacific. Deep-Sea Research Part Ii-Topical Studies in Oceanography, 51(1-3), 303-318. https://doi.org/10.1016/j.dsr2.2003.07.018

Dani, K. G. S., & Loreto, F. (2017). Trade-Off Between Dimethyl Sulfide and Isoprene Emissions from Marine Phytoplankton. Trends in Plant Science, 22(5), 361-372. https://doi.org/10.1016/j.tplants.2017.01.006

Friedland, K. D., Leaf, R. T., Kane, J., Tommasi, D., Asch, R. G., Rebuck, N., Ji, R., Large, S. I., Stock, C., & Saba, V. S. (2015). Spring bloom dynamics and zooplankton biomass response on the US Northeast Continental Shelf. Continental Shelf Research, 102, 47-61. https://doi.org/10.1016/j.csr.2015.04.005

Gantt, B., Meskhidze, N., & Kamykowski, D. (2009). A new physically-based quantification of marine isoprene and primary organic aerosol emissions. Atmospheric Chemistry and Physics, 9(14), 4915-4927. https://doi.org/DOI 10.5194/acp-9-4915-2009

Groetsch, P. M. M., Simis, S. G. H., Eleveld, M. A., & Peters, S. W. M. (2016). Spring blooms in the Baltic Sea have weakened but lengthened from 2000 to 2014. Biogeosciences, 13(17), 4959-4973. https://doi.org/10.5194/bg-13-4959-2016

Guo, S. J., Feng, Y. Y., Wang, L., Dai, M. H., Liu, Z. L., Bai, Y., & Sun, J. (2014). Seasonal variation in the phytoplankton community of a continental-shelf sea: the East China Sea. Marine Ecology Progress Series, 516, 103-126. https://doi.org/10.3354/meps10952

Kameyama, S., Yoshida, S., Tanimoto, H., Inomata, S., Suzuki, K., & Yoshikawa-Inoue, H. (2014). High-resolution observations of dissolved isoprene in surface seawater in the Southern Ocean during austral summer 2010-2011. Journal of Oceanography, 70(3), 225-239. https://doi.org/10.1007/s10872-014-0226-8

Kim, M. J., Novak, G. A., Zoerb, M. C., Yang, M. X., Blomquist, B. W., Huebert, B. J., Cappa, C. D., & Bertram, T. H. (2017). Air-Sea exchange of biogenic volatile organic compounds and the impact on aerosol particle size distributions. Geophysical Research Letters, 44(8), 3887-3896. https://doi.org/10.1002/2017gl072975

Liu, H., Probert, I., Uitz, J., Claustre, H., Aris-Brosou, S., Frada, M., Not, F., & de Vargas, C. (2009). Extreme diversity in noncalcifying haptophytes explains a major pigment paradox in open oceans. Proceedings of the National Academy of Sciences of the United States of America, 106(31), 12803-12808. https://doi.org/10.1073/pnas.0905841106

Liu, X., Xiao, W. P., Landry, M. R., Chiang, K. P., Wang, L., & Huang, B. Q. (2016). Responses of Phytoplankton Communities to Environmental Variability in the East China Sea. Ecosystems, 19(5), 832-849. https://doi.org/10.1007/s10021-016-9970-5

Milne, P. J., Riemer, D. D., Zika, R. G., & Brand, L. E. (1995). Measurement of vertical distribution of isoprene in surface seawater, its chemical fate, and its emission from several phytoplankton monocultures. Marine Chemistry, 48(3), 237-244. https://doi.org/https://doi.org/10.1016/0304-4203(94)00059-M

Moore, R. M., & Wang, L. (2006). The influence of iron fertilization on the fluxes of methyl halides and isoprene from ocean to atmosphere in the SERIES experiment. Deep-Sea Research Part Ii-Topical Studies in Oceanography, 53(20-22), 2398-2409. https://doi.org/10.1016/j.dsr2.2006.05.025

Palmer, P. I., Marvin, M. R., Siddans, R., Kerridge, B. J., & Moore, D. P. (2022). Nocturnal survival of isoprene linked to formation of upper tropospheric organic aerosol. Science, 375(6580), 562-+. https://doi.org/10.1126/science.abg4506

Palmer, P. I., & Shaw, S. L. (2005). Quantifying global marine isoprene fluxes using MODIS chlorophyll observations. Geophysical Research Letters, 32(9). https://doi.org/Artn L09805 10.1029/2005gl022592

Shaw, S. L., Gantt, B., & Meskhidze, N. (2010). Production and Emissions of Marine Isoprene and Monoterpenes: A Review. Advances in Meteorology, 2010. https://doi.org/Artn 408696 10.1155/2010/408696

Simo, R., Cortes-Greus, P., Rodriguez-Ros, P., & Masdeu-Navarro, M. (2022). Substantial loss of isoprene in the surface ocean due to chemical and biological consumption. Communications Earth & Environment, 3(1). https://doi.org/ARTN 20

10.1038/s43247-022-00352-6

Sinha, V., Williams, J., Meyerhofer, M., Riebesell, U., Paulino, A. I., & Larsen, A. (2007). Air-sea fluxes of methanol, acetone, acetaldehyde, isoprene and DMS from a Norwegian fjord following a phytoplankton bloom in a mesocosm experiment. Atmospheric Chemistry and Physics, 7, 739-755. https://doi.org/DOI 10.5194/acp-7-739-2007

Uitz, J., Claustre, H., Gentili, B., & Stramski, D. (2010). Phytoplankton class-specific primary production in the world's oceans: Seasonal and interannual variability from satellite observations. Global Biogeochemical Cycles, 24. https://doi.org/Artn Gb3016

10.1029/2009gb003680

Wanninkhof, R. (2014). Relationship between wind speed and gas exchange over the ocean revisited. Limnology and Oceanography-Methods, 12, 351-362. https://doi.org/10.4319/lom.2014.12.351

Warneke, C., de Gouw, J. A., Goldan, P. D., Kuster, W. C., Williams, E. J., Lerner, B. M., Jakoubek, R., Brown, S. S., Stark, H., Aldener, M., Ravishankara, A. R., Roberts, J. M., Marchewka, M., Bertman, S., Sueper, D. T., McKeen, S. A., Meagher, J. F., & Fehsenfeld, F. C. (2004). Comparison of daytime and nighttime oxidation of biogenic and anthropogenic VOCs along the New England coast in summer during New England Air Quality Study 2002. Journal of Geophysical Research-Atmospheres, 109(D10). https://doi.org/Artn D10309

10.1029/2003jd004424

Wohl, C., Jones, A. E., Sturges, W. T., Nightingale, P. D., Else, B., Butterworth, B. J., & Yang, M. X. (2022). Sea ice concentration impacts dissolved organic gases in the Canadian Arctic. Biogeosciences, 19(4), 1021-1045. https://doi.org/10.5194/bg-19-1021-2022

Wohl, C., Li, Q. Y., Cuevas, C. A., Fernandez, R. P., Yang, M. X., Saiz-Lopez, A., & Simo, R. (2023). Marine biogenic emissions of benzene and toluene and their contribution to secondary organic aerosols over the polar oceans. Science Advances, 9(4). https://doi.org/ARTN eadd9031

10.1126/sciadv.add9031

Wurl, O., Wurl, E., Miller, L., Johnson, K., & Vagle, S. (2011). Formation and global distribution of sea-surface microlayers. Biogeosciences, 8(1), 121-135. https://doi.org/10.5194/bg-8-121-2011

Xu, L., Cameron-Smith, P., Russell, L. M., Ghan, S. J., Liu, Y., Elliott, S., Yang, Y., Lou, S., Lamjiri, M. A., & Manizza, M. (2016). DMS role in ENSO cycle in the tropics. Journal of Geophysical Research-Atmospheres, 121(22), 13537-13558. https://doi.org/10.1002/2016jd025333

Yamada, K., & Ishizaka, J. (2006). Estimation of interdecadal change of spring bloom timing, in the case of the Japan Sea. Geophysical Research Letters, 33(2). https://doi.org/Artn L02608 10.1029/2005gl024792

Yu, Z. J., & Li, Y. (2021). Marine volatile organic compounds and their impacts on marine aerosol-A review. Science of the Total Environment, 768. https://doi.org/ARTN 145054 10.1016/j.scitotenv.2021.145054

---

## Author Response (AR2)

**Response to the referees for comments on "Enhanced dataset of global marine isoprene emission from biogenic and photochemical processes for the period 2001-2020"**

Lehui Cui[1], Yunting Xiao[1], Wei Hu[1], Lei Song[2], Yujue Wang[3], Chao Zhang[3], Pingqing Fu[1], Jialei Zhu[1]

[1]Institute of Surface-Earth System Science, School of Earth System Science, Tianjin University, Tianjin, 300072, China

[2]Center for Monsoon System Research, Institute of Atmospheric Physics, Chinese Academy of Sciences, Beijing, 100029, China

[3]Frontiers Science Center for Deep Ocean Multispheres and Earth System, and Key Laboratory of Marine Environment and Ecology, Ministry of Education, Ocean University of China, Qingdao 266100, China

*Correspondence to*: J. Zhu (zhujialei@tju.edu.cn)

**Response for comments and suggestions from referee#1**

**General comments:**

The revised manuscript "Enhanced dataset of global marine isoprene emission from biogenic and photochemical processes for the period 2001-2020" by Cui et al. has been improved in content and structure by addressing the comments of the reviewers.

However, I have a few comments and questions which should addressed before publication.

Biological consumption term ($\alpha$): In the response letter, the authors explain their recalculation of $\alpha$ using equation 1 in Simó et al., (2022) and the resulting changes in global isoprene emissions.

First. I still do not understand why $\alpha$=0.373, when [chla]=5.77 using $\alpha$=0.1 x [chla] +0.05. In this case 0.373=1- $\alpha$. Why do the authors calculate 1- $\alpha$ instead of $\alpha$ in ll. 124?

Second. Line 116 explains the calculation of the flux. However, multiplying $\alpha$ (h-1) with the production (g m-2 h-1) will not result in g m-2 h-1? Please check the provided equation, it's units and the calculations in the model.

Comparison BIO and SML module: I am still a little confused be the definition of the "BIO emissions" compared to the "SML emissions". For the SML emissions the authors state that the wind speed highly influences the emissions due to the cubic wind speed dependency - I totally agree. However, when it comes to the BIO emissions (assuming steady state isoprene concentrations?!) the authors state that those emissions are not influenced by wind speed as only the BIO production is accounted for. The authors say, due to the static assumption, production equals emission to the atmosphere. I am still struggling, if this assumption is valid, especially when talking about the wind speed. Why is the sea-to-air transfer coefficient k considered when calculating SML emissions but not when calculation BIO emissions? At high winds the SML emission sharply decrease due to a non-existing SML, however, at the same time, isoprene emissions should be very high due to high wind speed, but this is, to my

understanding not accounted for (as only production is taken into account). Perhaps the authors could give some info in the manuscript and state, why their assumption is valid enough in terms of error estimations. Furthermore, the sensitivity runs, using a pre-defined atmospheric mixing ratio of isoprene, show influence on the output of the BIO emission module. How is this even possible, when no gas exchange term is used (as discussed above) and the BIO emissions only depend on isoprene production and biological and chemical isoprene consumption?

Response to **General comments:**

Thank you for your latest comments and suggestions. We are grateful for your further concern of our response letter and updated manuscript. All the additions and corrections in the main text of our revised manuscript using red letters, while these updated paragraphs are quoted in *blue italics* in this response letter.

First, the form of Eq. 2 for the BIO emission model have been transformed with extended description of each term in case of misunderstanding. We updated the Eq. 2 and its description as following in Line 122: *"The value of $\alpha$ is calculated by the following equation based on previous observational study (Simo et al., 2022):*

$$\alpha = (0.0042 \times C_{chl} + 0.0021) \qquad \text{(When } C_{chl} < 5.77 \, mg \, m^{-3})$$

$$\alpha = (0.0042 \times 5.77 + 0.0021) = 0.026 \qquad \text{(When } C_{chl} \geq 5.77 \, mg \, m^{-3}) \qquad (2)$$

*The term $0.1 \times C_{chl}$ represents the degradation and utilization of isoprene by heterotrophic bacteria (Simo et al., 2022). It accounts for the observed correlation between bacterial activity and chlorophyll concentrations in the mixed layer. The second term 0.0021 is empirical rate of chemical consumption of isoprene per hour in the ocean (Palmer and Shaw, 2005; Booge et al., 2018). It is important to note that when the chlorophyll concentration in the seawater exceeds 5.77 mg·m⁻³, $\alpha$ is set to a constant value of 0.026 as a maximum stable biological and chemical consumption per hour. This approach was derived from observations when the chlorophyll concentration in the seawater was up to 5.77 mg·m⁻³ (Simo et al., 2022). Therefore, the specific value of 0.026 is determined to account for biological and chemical consumption in nutrient-rich environments."*

Second, the Eq. 1 was updated to make the units more reasonable according to the comments in Line 115: *"The BIO model can be expressed by the following equations:*

$$F_b = (1 - \alpha) \cdot P \cdot S \qquad (1)$$

*where $F_b$ (g·grid⁻¹·h⁻¹) represents the isoprene emission flux from the air-sea interface to the MBL, $P$ (g·m⁻²·h⁻¹) is the isoprene production rate generated by phytoplankton. $S$ (m²·grid⁻¹) is the grid cell area and $\alpha$ is chlorophyll-based rate constant to determine the biological and*

*chemical consumption of isoprene per hour."* In addition, the Eq. 1 only considers the biological and chemical loss of the incremental part by isoprene production.

Next, windspeed affects the BIO and SML emission process in totally different ways. The SML emission comes from photochemical processes in the sea micro-layer based on surfactant concentration there. Windspeed over the sea surface is applied to quantify the air-sea exchange velocity of isoprene from SML emission. The BIO emission is calculated according to biogenic production with the assumption that the net production rate equals the isoprene BIO emission flux since we assumed the isoprene mixing ratio is negligible in the marine boundary layer. As a result, windspeed was not used in the calculation of BIO emission as shown in Eq. 1. Actually, the isoprene in the marine boundary layer would suppress the air-sea exchange of isoprene from BIO emission, where air-sea exchange velocity is determined by windspeed and temperature (Eq. 11). To describe this suppress process, Eq. 13 has been added in Sect. 3.4, which was applied to test the sensitivity of isoprene in the MBL with various mixing ratios" (in Line 487). The discussion is added as follows: *"The BIO module is based on the assumption that isoprene in the MBL is of very short lifetime, as well as its low mixing ratio in most remote ocean areas. The isoprene presence in the MBL will inhibit the emission of marine isoprene to the MBL. Considering the atmospheric concentration of isoprene in the MBL ($C_{air}$), an emission suppression term is added into the Eq. (1):*

$$F_b = (1 - \alpha) \cdot P \cdot S - k_{ex} \cdot H \cdot C_{air} \tag{13}$$

*In Eq. (13), the air-sea exchange velocity $k_{ex}$ (m·h$^{-1}$) is determined by Eq. (11). H is a dimensionless Henry's law constant, which is calculated by Mochalski et al. (2011):*

$$H = exp\left(-17.85 + \frac{4130}{T+273.16}\right) \tag{14}$$

*Here T is water temperature in Celsius degree."*

The last term in Eq. 13 accounts for the influence of atmospheric isoprene concentration on BIO emissions with the contribution of windspeed. Due to the lack of simulated isoprene concentration in the offline BIO emission calculation, we omitted this term in Eq. 1. It is very possible to calculate BIO emission using Eq. 13 instead of Eq. 1 in the future work when the BIO module is coupled into earth system model which is able to simulate the isoprene

concentration in the MBL.

Finally, the SML emission at high windspeed is strictly limited by the threshold of 13m·s$^{-1}$, when the SML no-longer exists. Previous study has pointed out that under the limit of 13m·s$^{-1}$, almost all ocean surface are covered by SML with different equivalent surfactant concentrations (Brüggemann et al., 2018). Moreover, we checked all windspeed date and found that windspeed larger than 13m·s$^{-1}$ only occurred in about 3% grid cells with daily windspeed. Therefore, we think this windspeed limit is almost reasonable and makes very small uncertainty in the estimation of global SML emission.

Response to **Specific comments**

**1.   Introduction: The last sentence of the abstract should be moved to the paragraph "data availability".**

Response:

Thank you for your comments. It seems the DOI link at the end of the abstract is a common custom for this journal.

**2.   Line 56: "Several enhancements…" This new sentence in the revised version does not belong to the introduction if the authors are talking about their own refinements. If this content reflects work from other people they should be cited here.**

Response:

Thank you for your comment. Related citation *(Gantt et al., 2009)* have been added in Line 59.

**3.   Line 315: Could the authors give some insights (references) why they included the SML emissions when calculating the oceanic isoprene concentrations in order to compare with observations? Is it likely that gases which are produced in the SML will diffuse to the underlying water instead of being emitted to the atmosphere? Observations, the authors compare their model results to, are normally made at ~5m. In most of the comparison studies (Figure 4) the simulated isoprene concentrations are at the higher end or even significantly higher than the range of observations. Might this be due to the fact that SML emissions are included in the calculation of the simulated oceanic isoprene concentrations?**

Response:

Thank you for your comments and useful information. Indeed, the sampling depth of ~5m is not considered in our comparison of observed marine isoprene concentration and our dataset. As the referee commented, including SML emission may cause an overestimation of isoprene concentration in the sea water. However, we cannot quantify the isoprene diffused into seawater from SML emission in recent version module. As a result, we actually got a maximum isoprene

concentration in the seawater with assumption that all isoprene from SML emission enters the underlying seawater. We added related sentences in Line 310 to remind this tips: *"Note that here both BIO and SML emission are considered to have effects on the marine isoprene concentration with assumption that all isoprene from SML emission enters the underlying seawater. This may cause an overestimation of the isoprene concentration in the seawater compared to the actual situation."*

**4. Line 418: "The uncertainty…". It is not clear, which uncertainty is described here. Is it based on interannual variability? Is it 1sigma standard deviation? Or is this uncertainty only based on the phytoplankton sensitivity tests?**

Response:

Thank you for your comments. The uncertainty mentioned here is based on the sensitivity tests. For BIO emission, the uncertainty here accounts for test of phytoplankton types. For SML emission, the uncertainty here accounts for the surfactant concentration test. We revised the related sentences for a more specific description in Line 412: *"From a series of sensitivity tests, the range of annual global BIO emission is 0.443 to 0.664 Tg·yr$^{-1}$ and 0.583 to 0.655 Tg·yr$^{-1}$ for SML emission."*

References:

Brüggemann, M., Hayeck, N., and George, C.: Interfacial photochemistry at the ocean surface is a global source of organic vapors and aerosols, Nat Commun, 9, ARTN 2101 https://doi.org/10.1038/s41467-018-04528-7, 2018.

Conte, L., Szopa, S., Aumont, O., Gros, V., and Bopp, L.: Sources and Sinks of Isoprene in the Global Open Ocean: Simulated Patterns and Emissions to the Atmosphere, J Geophys Res-Oceans, 125, ARTN e2019JC015946 https://doi.org/10.1029/2019JC015946, 2020.

Gantt, B., Meskhidze, N., and Kamykowski, D.: A new physically-based quantification of marine isoprene and primary organic aerosol emissions, Atmos Chem Phys, 9, 4915-4927, https://doi.org/10.5194/acp-9-4915-2009, 2009.

Mochalski, P., King, J., Kupferthaler, A., Unterkofler, K., Hinterhuber, H., & Amann, A. (2011). Measurement of isoprene solubility in water, human blood and plasma by multiple headspace

extraction gas chromatography coupled with solid phase microextraction. Journal of Breath Research, 5(4), 046010. https://doi.org/10.1088/1752-7155/5/4/046010

Palmer, P. I. and Shaw, S. L.: Quantifying global marine isoprene fluxes using MODIS chlorophyll observations, Geophys Res Lett, 32, https://doi.org/10.1029/2005gl022592, 2005.

Simo, R., Cortes-Greus, P., Rodriguez-Ros, P., and Masdeu-Navarro, M.: Substantial loss of isoprene in the surface ocean due to chemical and biological consumption, Commun Earth Environ, 3, ARTN 20 https://doi.org/10.1038/s43247-022-00352-6, 2022.

**Response for comments and suggestions from referee#2**

**General comments:**

This revised manuscript presents a global marine isoprene emission dataset at high spatial and temporal resolution spanning the period 2001 - 2020. Emissions are calculated using a combination of satellite chlorophyll and radiance measurements and meteorological reanalysis data with empirical parameterizations. Uncertainties in the emission estimate are quantified using a sensitivity analysis which shows how the emission estimate responds to changes in the input meteorology and satellite data as well as plankton distributions and empirical parameters. The estimated emissions are compared with a variety of observational records, and correlations with meteorological driving variables as well as climate modes of variability are explored. This manuscript has been significantly improved by the authors' revisions. All of my major concerns from the previous version have been addressed.

Specifically, the new sensitivity analysis described in Section 3.4 and summarized in Table 5 provides strong evidence that the emissions estimates are quite robust. This is extremely important for potential data users, and it also makes the comparisons with observations and previous emission estimates (Sections 3.1 and 3.2) more meaningful. Key assumptions are justified based on previous observations or modelling results, and these justifications are now clearly explained in the text. The multi-panel plots are much easier to interpret with the addition of descriptive labels, legends, and captions. Confusing terminology has also been largely fixed. The authors added their module code to the online database, which will be helpful for future modelling studies. I also appreciate that the authors have also expanded the discussion of the trends and variability of the data in Section 4, which makes the paper more interesting than a simple data description.

I believe this dataset will be highly valuable for the global atmospheric modelling community,

especially for anyone working on chemistry-climate interactions in the marine environment. I think the scientific content and overall structure of the revised manuscript is sound and does not need to be changed. The phrasing and grammar of the manuscript could be improved for clarity, but the paper is reasonably easy to understand as is.

I recommend accepting this manuscript for publication as is.

Response to **General comments:**

Thank you for your latest comments. We are pleased to have your suggestions and support during the review process of this work.